



# In situ observations of CH$_2$Cl$_2$ and CHCl$_3$ show efficient transport pathways for very short-lived species into the lower stratosphere via the Asian and North American summer monsoons

Valentin Lauther[1], Bärbel Vogel[2], Johannes Wintel[1,a], Andrea Rau[1], Peter Hoor[3], Vera Bense[3], Rolf Müller[2], and C. Michael Volk[1]

[1]Institute for Atmospheric and Environmental Research, University of Wuppertal, Wuppertal, Germany
[2]Forschungszentrum Jülich, Institute of Energy and Climate Research – Stratosphere (IEK-7), Jülich, Germany
[3]Institute for Atmospheric Physics, Johannes Gutenberg University, Mainz, Germany
[a]now at: Elementar GmbH, Langenselbold, Germany

**Correspondence:** Valentin Lauther (lauther@uni-wuppertal.de)

**Abstract.**

Efficient transport pathways for ozone depleting very short-lived substances (VSLS) from their source regions into the stratosphere are a matter of current scientific debate, however they have yet to be fully identified on an observational basis. Understanding the increasing impact of chlorine containing VSLS (Cl-VSLS) on stratospheric ozone depletion is important in order to validate and improve model simulations and future predictions. We report on the first transport study using airborne in situ measurements of the Cl-VSLS dichloromethane (CH$_2$Cl$_2$) and trichloromethane (chloroform, CHCl$_3$) to derive a detailed description of the two most efficient and fast transport pathways from (sub-)tropical source regions into the extratropical lower stratosphere (Ex-LS) in northern hemisphere (NH) late summer. The Cl-VSLS measurements were obtained in the upper troposphere and lower stratosphere (UTLS) above Western Europe and the mid latitude Atlantic Ocean in the frame of the WISE (Wave-driven ISentropic Exchange) aircraft campaign in autumn 2017 and are combined with the results from a three-dimensional simulation of a Lagrangian transport model as well as back-trajectory calculations. Compared to background measurements of similar age we find up to $150\,\%$ enhanced CH$_2$Cl$_2$ and up to $100\,\%$ enhanced CHCl$_3$ mixing ratios in the Ex-LS. We link the measurements of enhanced mixing ratios to emissions in the region of southern and eastern Asia. Transport from this area to the Ex-LS at potential temperatures in the range of $370 - 400\,\mathrm{K}$ takes about $5 - 10$ weeks via the Asian summer monsoon anticyclone (ASMA). Our measurements suggest anthropogenic sources to be the cause of these strongly elevated Cl-VSLS concentrations observed at the top of the lowermost stratosphere (LMS). A faster transport pathway into the Ex-LS is derived from particularly low CH$_2$Cl$_2$ and CHCl$_3$ mixing ratios in the UTLS. These low mixing ratios reflect weak emission sources and a local seasonal minimum of both species in the boundary layer of Central America and the tropical Atlantic. We show that air masses uplifted by hurricanes, the North American monsoon, and general convection above Central America into the tropical tropopause layer to potential temperatures of about $360 - 370\,\mathrm{K}$ are transported isentropically within $1 - 5$ weeks into the Ex-LS. This transport pathway linked to the North American monsoon mainly impacts the middle and lower part of the LMS with particularly low CH$_2$Cl$_2$ and CHCl$_3$ mixing ratios. In a case study, we specifically analyze air samples directly





linked to the uplift by the category 5 hurricane Maria that occurred during October 2017 above the Atlantic Ocean. Regionally differing CHCl$_3$:CH$_2$Cl$_2$ emission ratios derived from our UTLS measurements suggest a clear similarity between CHCl$_3$ and

CH$_2$Cl$_2$ when emitted by anthropogenic sources and differences between the two species mainly caused by additional, likely biogenic, CHCl$_3$ sources. Overall, the transport of strongly enhanced CH$_2$Cl$_2$ and CHCl$_3$ mixing ratios from southern and eastern Asia via the ASMA is the main factor for increasing the chlorine loading from the analyzed VSLS in the Ex-LS during NH late summer. Thus, further increases in Asian CH$_2$Cl$_2$ and CHCl$_3$ emissions, as frequently reported in recent years, will further increase the impact of Cl-VSLS on stratospheric ozone depletion.

## 1 Introduction

Within the last two decades emissions of the chlorine containing very short-lived substances (Cl-VSLS) dichloromethane (CH$_2$Cl$_2$) and trichloromethane (chloroform, CHCl$_3$) increased significantly by about 8 %/year (Hossaini et al., 2015) and 3.5 %/year (Fang et al., 2018), respectively. With both Cl-VSLS not being regulated by the Montreal Protocol on Substances

that Deplete the Ozone Layer and its amendments and adjustments their influence on stratospheric ozone depletion is currently an important topic of investigation. Due to the sparseness of Cl-VSLS measurements in the stratosphere the impact of changes in Cl-VSLS surface emissions on their distribution in the stratosphere has yet to be fully characterized on an observational basis. Particularly important is the identification of rapid and efficient transport pathways for Cl-VSLS from their source regions into the stratosphere. The present study is the first to use airborne in situ measurements of CH$_2$Cl$_2$ and CHCl$_3$ in the extratropical

lower stratosphere (Ex-LS) to analyze the impact of different source regions onto the stratospheric chemical composition and to identify transport pathways of CH$_2$Cl$_2$ and CHCl$_3$ into the stratosphere.

CH$_2$Cl$_2$ is almost exclusively emitted by anthropogenic sources with only about 10 % of its emission being of natural origin (Engel et al., 2018). Thereby CH$_2$Cl$_2$ mixing ratios in the troposphere at northern hemisphere (NH) mid latitudes are by a factor of three larger than those in the southern hemisphere (Hossaini et al., 2017). Global CH$_2$Cl$_2$ emissions in 2017 are estimated to

be about 1 (Tg Cl)/year (Claxton et al., 2020) which is even higher than emissions of CFC-11 at its emission peak in 1986 with about 0.3 (Tg Cl)/year (Daniel et al., 2007, however, CFC-11 is practically inert in the troposphere, which is not the case for CH$_2$Cl$_2$). Almost 90 % of the global CH$_2$Cl$_2$ emission sources are located in Asia (Claxton et al., 2020). Other, more localized studies, estimate that about 10 % of global CH$_2$Cl$_2$ emissions originate in India (Say et al., 2019) and that $25 - 37$ % (Feng et al., 2018) or even 45 % (Oram et al., 2017) of global CH$_2$Cl$_2$ emissions originate in China. Collected air samples from

IAGOS-CARIBIC confirm particularly high emissions in the broad region of southern and eastern Asia (Leedham-Elvidge et al., 2015) as similarly shown for the north Indian subcontinent from air sampled during the StratoClim aircraft campaign in summer 2017 (Adcock et al., 2021). European and American CH$_2$Cl$_2$ sources in 2017 were estimated to contribute less than 10 % to the global CH$_2$Cl$_2$ emissions (Claxton et al., 2020).





Based on AGAGE ground-based measurements Engel et al. (2018) estimate the global $CHCl_3$ emissions in 2017 to be about
0.29 (Tg Cl)/year. Compared to $CH_2Cl_2$ the distribution of $CHCl_3$ emission sources is rather unclear. On global average Engel
et al. (2018) estimate $CHCl_3$ emissions from anthropogenic sources to be equally high as from biogenic sources. However,
emission estimates of anthropogenic $CHCl_3$ sources range between 60 % (Trudinger et al., 2004), 30 % (Worton et al., 2006),
and 10 % (McCulloch, 2003) of the total emissions. The increase in global $CHCl_3$ emissions during the last decade was traced
back entirely to an increase in eastern Chinese $CHCl_3$ emissions of most likely anthropogenic origin (Fang et al., 2018). In
addition, Chinese $CHCl_3$ emissions amount to almost 90 % of all East Asian $CHCl_3$ emissions (Fang et al., 2018).

Anthropogenic emissions of $CH_2Cl_2$ and $CHCl_3$ partly arise from the non-chemical industry like pulp and paper manufacture
and water treatment, as well as from the chemical and pharmaceutical industry (Aucott et al., 1999; McCulloch, 2003). Both
$CH_2Cl_2$ and $CHCl_3$ are the main co-products needed for any manufacturing process of chloromethanes. $CH_2Cl_2$ is also strongly
used in the production of the hydrofluorocarbon HFC-32 (Chipperfield et al., 2020). The fraction of produced $CH_2Cl_2$ and
$CHCl_3$ used as feedstock to manufacture other halogens is not further released into the atmosphere. This fraction has been
estimated for $CHCl_3$ to be over 95 % (McCulloch, 2003; Chipperfield et al., 2020) but only 15 % for $CH_2Cl_2$ (Chipperfield
et al., 2020). The remaining fraction not used as feedstock for the production of other halogens (about 5 % of $CHCl_3$; about
% of $CH_2Cl_2$) is emitted into the atmosphere in form of chemical solvent, paint stripper, and degreasing agent (McCulloch
and Midgley, 1996; Montzka et al., 2011) as well as through foam blowing and agricultural fumigation (Oram et al., 2017).
While $CH_2Cl_2$ is believed to have no significant oceanic sources and is only temporarily taken up by the oceans to be re-
released to the atmosphere later, a process that is not yet fully understood (Moore, 2004), $CHCl_3$ is estimated to have about
50 % of its biogenic emission sources located in offshore seawater (Laturnus et al., 2002; McCulloch, 2003). Biomass burning
was reported to play a significant role in $CH_2Cl_2$ emissions (Rudolph et al., 1995) but more recent studies could not confirm
that finding (Simmonds et al., 2006; Simpson et al., 2011; Leedham-Elvidge et al., 2015). In contrast, biomass burning was
observed to be a small but significant $CHCl_3$ emission source (Rudolph et al., 1995; Scheeren et al., 2002; Bourtsoukidis et al.,
2017).

The main atmospheric sink of both $CH_2Cl_2$ and $CHCl_3$ is the reaction with hydroxyl radicals (OH) in the troposphere. Time
series of background mixing ratios of both species are anticorrelated to the seasonal cycle of OH (Cox et al., 2003). In the NH
seasonal anthropogenic use of products releasing $CHCl_3$ to the atmosphere (e.g., landfill and chlorination of water) have been
observed to have a small local impact on the seasonality of $CHCl_3$ (Gentner et al., 2010). In addition, the global distribution of
OH shows significant regional differences (Spivakovsky et al., 2000; Hanisco et al., 2001; Lelieveld et al., 2016). Therefore,
also the photochemical lifetimes of $CH_2Cl_2$ and $CHCl_3$ are regionally different. For $CH_2Cl_2$ Hossaini et al. (2019) suggest an
average tropospheric lifetime of 168 days (about 6 months) and a stratospheric lifetime of $1 - 2$ years (outside the poles) was
estimated by Hossaini et al. (2017). Both species have similar reaction rates with OH implying similar photochemical lifetimes
for both Cl-VSLS (Hsu and DeMore, 1994).

In the tropical tropopause layer (TTL) the lifetime of both $CH_2Cl_2$ and $CHCl_3$ is estimated to be about $6 - 10$ months, being
long enough for both Cl-VSLS to enter the stratosphere under normal dynamic conditions (Park et al., 2010). For the level of
zero radiative heating Hossaini et al. (2015) simulated an increase in average $CH_2Cl_2$ mixing ratios of about 83 % between





2005 and 2013. Hossaini et al. (2019) estimate an increase of total stratospheric chlorine from Cl-VSLS from about 69 ppt in
2000 to about 111 ppt in 2017 of which > 80 % enter the stratosphere as source gases and the rest as product gases of Cl-VSLS.
Hossaini et al. (2019) further state that $CH_2Cl_2$ and $CHCl_3$ contribute with about 68 % and 19 % to this increase, respectively.
However, due to high Asian emissions and an efficient transport into the stratosphere via the Asian summer monsoon (ASM),
the estimation of stratospheric chlorine from Cl-VSLS could even be underestimated by 8 − 26 % (Adcock et al., 2021).

Between June and September the ASM is a wide spread convective system located above the Indian subcontinent, East
and Southeast Asia (e.g., Yihui and Chan, 2005). The ASM provides fast vertical transport of surface air into the large scale
anticyclone (ASMA) above, which spans from the upper troposphere at about 360 K potential temperature to the lower strato-
sphere at about 450 K potential temperature (e.g., Park et al., 2007, 2009; Bergman et al., 2013; Vogel et al., 2019). Within
the ASMA air masses are somewhat confined and separated from the surrounding UTLS air by a strong gradient of potential
vorticity (e.g., Ploeger et al., 2015). Several studies have shown that these air masses are transported further vertically into the
tropical pipe or break out of the ASMA to enter the extratropical LMS quasi-horizontally by Rossby wave breaking events
(e.g., Popovic and Plumb, 2001; Garny and Randel, 2016; Vogel et al., 2014, 2016). Thus, the ASM has a strong impact on the
chemical composition of the stratosphere in boreal summer (e.g., Randel et al., 2010; Randel and Jensen, 2013; Vogel et al.,
2015; Santee et al., 2017).

The specific transport pathways of Cl-VSLS from their source regions into the stratosphere have not been identified on an
observational basis. The most efficient transport pathway for Cl-VSLS into the stratosphere is suggested to be via the ASMA.
This is why Cl-VSLS emissions from the region of continental Asia are suggested to have the highest ozone depletion potential
(ODP) compared to emissions from other source regions (Claxton et al., 2019). Their regionally dependent ODP is estimated
to be in the range of 0.0097 − 0.0208 ($CH_2Cl_2$) and 0.0143 − 0.0264 ($CHCl_3$) (for comparison here the ODPs of some other
chlorocarbons: CFC-11: 1; $CCl_4$: 0.87; HCFC-22: 0.034; $CH_3Cl$: 0.015; Carpenter et al., 2018; Claxton et al., 2019).

Projecting different past $CH_2Cl_2$ emission rates Hossaini et al. (2017) predict a possibly significant delay of the recovery
date of stratospheric ozone ranging from a few years up to no recovery at all compared to estimations including only long-lived
chlorinated species. They also found a doubling of southern hemispheric spring time ozone loss caused by $CH_2Cl_2$ between
2010 and 2016. Their study emphasizes the importance of studying Cl-VSLS on a regular observational basis. However, the
estimated impact of Cl-VSLS on stratospheric ozone trends is small compared to that of long-lived chlorinated species or even
the impact of meteorology or the 11-year solar cycle (Chipperfield et al., 2018). Nevertheless, with the expected decrease of
long-lived chlorinated trace gases during the next decades due to the Montreal Protocol and its amendments and adjustments
the relative importance of Cl-VSLS on stratospheric ozone depletion will further increase.

Observational evidence for Cl-VSLS being transported into the stratosphere is extremely rare and their main transport path-
ways into the stratosphere have not been described on an observational basis. In the present paper we use in situ measurements
of $CH_2Cl_2$ and $CHCl_3$ to identify two efficient transport pathways for Cl-VSLS from the boundary layer into the Ex-LS. In
addition we provide observational evidence for different impacts on the stratospheric chemical composition depending on the
transport pathway the two Cl-VSLS take to enter the Ex-LS in NH late summer.





## 2 Airborne observations and model simulations

### 2.1 The WISE campaign 2017

All measurements presented in this study were obtained in the frame of the WISE (Wave-driven Isentropic Exchange) campaign (Riese et al., 2017, last accessed: 2021-01-14) which took place in September and October 2017. A total of 15 scientific flights were carried out with the German HALO (High Altitude and LOng range) research aircraft mainly from Shannon (Ireland) and from Oberpfaffenhofen (Germany) probing a wide area above the Atlantic Ocean and Western Europe. Among other goals, the WISE campaign aimed at investigating transport and mixing processes in the extratropical tropopause layer and the Ex-LS, the

impact of the Asian monsoon system on the chemical composition of the extratropical lowermost stratosphere (LMS), as well as the role of halogenated VSLS for ozone depletion and radiative forcing in the upper troposphere/lower stratosphere (UTLS) region. In this study we present UTLS measurements up to a potential temperature of $404\,\mathrm{K}$ of the last ten WISE flights, i.e., from 28 September to 21 October 2017 (Figure 1).

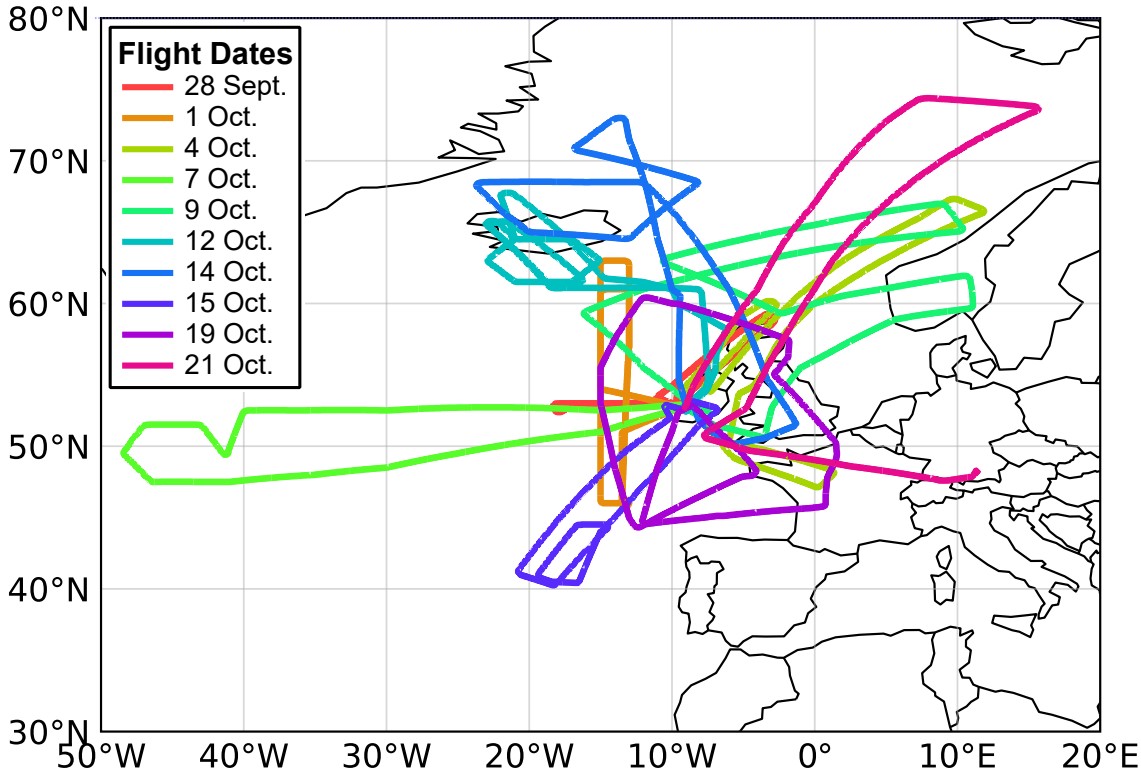

**Figure 1.** Map of ten flight tracks carried out with the German HALO (High Altitude and LOng range) research aircraft from and to Shannon (Ireland) with one flight from Shannon to Oberpfaffenhofen (Germany). The flights were conducted from 28 September to 21 October in 2017 in the frame of the WISE campaign (details see text).



## 2.2 In situ trace gas measurements

Our analysis is mainly based on airborne in situ observations of the trace gas instruments HAGAR-V ($CH_2Cl_2$ and $CHCl_3$) and UMAQS ($N_2O$) (as described below). The corresponding avionic data is provided by the Basic HALO Measurement and Sensor System (BAHAMAS) (Krautstrunk and Giez, 2012; Giez et al., 2017). The different measurement frequencies of the instruments were matched to that of HAGAR-V's MS module of $1/180$ Hz. Exceptions are the flights on 28 September and 1 October where the MS measurement frequency is $1/240$ Hz. Each data point is the average of a time interval of $40$ s, except

for the flights on 28 September, 1 October, and 4 October where it is $60$ s, corresponding to a spatial resolution at maximum cruising speed of $10$ km and $15$ km along the flight path, respectively. The used time and location of a data point is the respective center of the averaged time interval.

### 2.2.1 High Altitude Gas AnalyzeR - 5 channel version (HAGAR-V)

HAGAR-V is a novel airborne in situ instrument. It is a modernized and largely extended version of the airborne in situ instru-

ment HAGAR (Werner et al., 2010) and is mounted in a HALO standard rack (R-G550SM). Similar to HAGAR, HAGAR-V comprises a two-channel gas chromatograph with electron capture detection (GC/ECD) as well as a non dispersive infrared absorption module for the detection of $CO_2$ (LI-COR 7000). In contrast to HAGAR, HAGAR-V additionally comprises a mass spectrometer (MS) coupled to two GC channels by a two-position valve which allows to switch between the two channels. This novel MS module can thus be used either for the detection of a wide range of atmospheric trace gases (different target species

on each channel) or to double the measurement frequency (same target species on both channels). However, during WISE only one of the two GC/MS channels was used, measuring nine different species ($CH_2Cl_2$, $CHCl_3$, $CH_3Cl$, CFC-11, CFC-113, HFC-125, HFC-134a, iso-, and n-Pentane). In this study, the focus is on $CH_2Cl_2$ and $CHCl_3$ measurements by HAGAR-V's novel MS module, thus the instrumental description is confined only to the GC/MS part of the instrument. A more detailed description of HAGAR-V is given by Lauther (2020).

The general MS sampling process during WISE was as follows: Ambient air is drawn from outside the aircraft to the instrument and is further compressed to $3$ bar (a) by two diaphragm pumps (KNF 813.5 and 814) connected in series. The sample passes through a preconcentration tube packed with about $70$ mg Carboxen 572 (Supelco) at $20\,°C$ to adsorb the target species. At a usual adsorption time of $40$ s the preconcentrated sample volume is about $130$ ml. Afterwards the sample is desorbed by flash heating the trap to about $270\,°C$ and injected on to the separation columns by applying a helium carrier gas

flow. The sampled species are separated within two $0.25$ mm J&W Scientific $Al_2O_3/Na_2SO_4$ PLOT capillary columns of $4$ m and $5$ m length (pre- and main-column, respectively). Both columns are temperature controlled changing from initial $35\,°C$ to final $160\,°C$ in $20$ s (pre-column) and $35$ s (main-column), providing two sample refocusing steps in the process. The sample is detected by a quadrupole MS detector (5975C, Agilent Technologies) using electron ionization (EI) mode.

Fast GC/MS measurements are mandatory when operating from aboard an aircraft. To achieve a sample frequency of

$1/180$ Hz per MS channel particularly the heating and cooling rates of the preconcentration traps and the columns were



optimized during the MS module development process. Both units are self built, keeping the design and the application as adaptable as possible.

The cooling of the preconcentration traps is realized by a Stirling cooler (Twinbird, SC-UD08) and each trap is heated by a self regulating Ni heating wire (which is also used as temperature sensor) convoluted around the trap tube. To our knowledge,
HAGAR-V is the only state-of-the-art airborne GC/MS instrument using indirect trap heating and our thermodesorption design provides consistent heating and cooling rates of $80\,°C\,s^{-1}$ and $-25\,°C\,s^{-1}$ (from $270\,°C$ down to $20\,°C$) inside the trap tube. In addition, our thermodesorption concept avoids large variable currents at relatively low voltages (peak current $< 7\,A$ at $48\,V$ for $< 2\,s$, then $< 2\,A$) and is thus well suited to be used aboard an aircraft with stringent constraints regarding electromagnetic compatibility.

The self built separation column ovens are conceptually comparable to the principles of regular modern Low Thermal Mass capillary column systems (e.g., Luong et al., 2006). In parallel to the column a heating wire and a temperature sensor wire are coiled to a torus of $7\,cm$ in diameter acquiring fast and homogenous heat application to the column (up to $20\,°C\,s^{-1}$) and temperature read-out. Deactivated capillary columns emerge from the torus functioning as connecting lines and particle traps to enhance the measurement stability. The cooling of the columns is realized by fans providing a setback from final to initial
temperatures within $60\,s$.

Following the compression by the inlet pumps, usually the air sample is dehydrated because water vapor can strongly affect the reproducibility of MS measurements. However, during WISE the dehydration system of HAGAR-V could not be used. Consequently, the MS module measured only at low ambient water vapor levels (mainly at $H_2O < 100\,ppm$; median: $5.6\,ppm$), i.e., in the UTLS region. With this solution the MS module measured during about 90% of a typical WISE flight's duration
(i.e., about $7.6\,h$ per flight).

HAGAR-V uses two different working standards for in-flight calibration to enhance the accuracy in case of non-linear system responses. Both working standards consist of compressed clean ambient air; one of them is additionally diluted with about $25\,\%$ synthetic air. The main bottles of the working standards were calibrated by the University of Frankfurt against a calibration gas that was calibrated in second generation against an AGAGE standard on the SIO-14 ($CH_2Cl_2$) and SIO-98 ($CHCl_3$) scale.
Every second or third flight the in-flight calibration gas bottles were refilled from the main bottles after a calibration between main and flight bottles. Considering possible differences between main and flight bottles, uncertainties of the mixing ratios within the main bottles, as well as potential influence from HAGAR-V's inlet pump system, the MS relative accuracy was estimated to be $2.0\,\%$ and $4.4\,\%$ for $CH_2Cl_2$ and $CHCl_3$, respectively.

Measurement precisions were optimized during data processing, using a strongly adapted version of the IGOR Pro analysis
package called NOAHChrom, originally developed by NOAA, USA. Exponentially modified Gaussian (EMG) functions were fitted to the MS signal peaks within individual time windows. Thereby peak tailing could be accurately treated and neighboring peaks were included in the background fit. In addition, the MS data were corrected for small system contamination and an occasional systematic measurement bias of one calibration gas. The measurement precision was derived for each flight from the standard deviation of one of the two in-flight calibration gases relative to its mixing ratio. The median precisions during
WISE were $1.7\,\%$ ($1\,ppt$) and $2.7\,\%$ ($0.4\,ppt$) for $CH_2Cl_2$ and $CHCl_3$, respectively.



### 2.2.2 University of Mainz Airborne Quantum cascade Laser Spectrometer (UMAQS)

UMAQS simultaneously measures CO and $N_2O$ from aboard HALO. The instrument uses the principle of direct absorption spectroscopy of a continuous-wave quantum cascade laser operating at a sweep rate of $2\,kHz$ (Müller et al., 2015). In this study we use UMAQS' measurements of $N_2O$ with a total drift-corrected uncertainty of $0.18\,ppb$ (Kunkel et al., 2019). Note, for

this study the $N_2O$ measurements are averaged over $40 - 60\,s$ to fit the integration times of HAGAR-V's MS module, thereby smoothing out instrumental noise which most likely further improves the $N_2O$ precision. The instrument is calibrated regularly in-flight using a secondary standard which is calibrated against a NOAA standard before and after the campaign. The accuracy of the used $N_2O$ mixing ratios is $0.39\,ppb$.

### 2.3 CLaMS simulations

To support the interpretation of airborne measurements we use global three-dimensional simulations of the Chemical Lagrangian Model of the Stratosphere (CLaMS; McKenna, 2002a,b; Pommrich et al., 2014) as well as pure CLaMS back-trajectory calculations. Pure CLaMS trajectory calculations consider only the advective (reversible) transport, neglecting (irreversible) mixing processes entirely (e.g., Vogel et al., 2019; Hanumanthu et al., 2020). However, in this study back-trajectories are useful to trace back the detailed transport pathway and transport time of an air parcel in the UTLS to possible source

regions in the boundary layer and therefore provide added value compared to three-dimensional CLaMS simulations. Both three-dimensional CLaMS simulations and back-trajectory calculations are driven by ECMWF ERA-Interim reanalysis data with a horizontal resolution of $1° \times 1°$ (Dee et al., 2011). The irreversible part of transport was set to discrete mixing steps every $24\,h$ in the three-dimensional CLaMS simulation. CLaMS employs a hybrid vertical coordinate ($\zeta$) which, in this study, transforms from a strictly isentropic coordinate ($\Theta$) to a pressure-based orography-following coordinate system ($\sigma$ coordinates)

below a threshold of $300\,hPa$. More detailed information about CLaMS is given by Pommrich et al. (2014) and references therein. Equivalent latitudes and the location of the thermal tropopause (lapsrate, according to WMO) along the flight path was calculated from ERA-Interim reanalysis data.

### 2.3.1 Artificial tracers of air mass origin

In this study CLaMS simulations of artificial tracers of air mass origin (also referred to as surface origin tracers, $\Omega_i$; Vogel et al.,

2015, 2016, 2019) are used to identify the location of the origin of air masses whose impact can be seen in the concentration data gathered during WISE. The surface origin tracers are released within 24 defined regions in the boundary layer ($\zeta = 120\,K \sim 2 - 3\,km$ above ground, including orography) as shown in Figure 2, top panel. The different surface origin tracers are continuously released (every $24\,h$) at the model boundary and are subsequently transported (advected and mixed) to the free atmosphere during the course of the simulation. The simulation was initialized with the meteorological data from 1 May, 2017,

implying that all air parcels residing in the free troposphere and the stratosphere at this date are not marked with the surface origin tracers. As a consequence, the fraction of all surface origin tracers ($\Omega = \sum_{i=1}^{n=24} \Omega_i$) of an air parcel can be $< 100\,\%$ during the course of the simulation because also air masses older than 1 May 2017 can contribute to the composition of an




air parcel. In this study we examine short-lived species measured in October 2017 with the focus on relatively fast transport, therefore a simulation period of approximately $5 - 6$ months is chosen here. The used model simulation is spatially constrained

from the surface to $\Theta = 900\,\mathrm{K}$ (about $37\,\mathrm{km}$ altitude) with a horizontal resolution of $100\,\mathrm{km}$ and a maximum vertical resolution of about $400\,\mathrm{m}$ (at the tropopause).

**Figure 2.** World map depicting the boundaries of CLaMS's 24 surface origin tracers (top) and three surface origin tracers combining several tracers from regions of significant (> 90 %) impact on the WISE measurements (bottom). The tracer names corresponding to their abbreviations are listed next to the maps. Also included in the list are the tracers of combined regions (cf. Section 3.1.2).





### 2.3.2 Back-trajectory calculations

In order to investigate the transport pathways corresponding to the WISE measurements analyzed here, the trajectory module of CLaMS was used to calculate back-trajectories. Other than in the three-dimensional CLaMS simulation, trajectories are calculated using only the advective part of transport without mixing. The advantage is that a single pathway for each air parcel can be calculated and analyzed with the additional information of its transport time. The back-trajectories are initialized at the time and location of the center of the respective MS sample integration time window and end at the first contact with the model boundary layer (below $2 - 3$ km above surface). In general, the maximum length of a trajectory is confined to 120 days.

The spatial uncertainty of calculated back-trajectories increases with time because mixing processes occurring during transport are neglected. However, the back-trajectory analysis is used here in a statistical way (ensembles of about 100 to 200 trajectories) and not to consider single trajectories. In addition, the maximum trajectory length of 120 days was chosen to match a large part of the time frame of the three-dimensional CLaMS simulation but the average length of the used back-trajectories is 50 days. We will show (in sections 3.1.2 and 3.1.3) that the results of the three-dimensional CLaMS simulation in which mixing of air parcels is included agrees very well with the results of the back-trajectory analysis.

## 3 Results

### 3.1 CH$_2$Cl$_2$-N$_2$O correlation during WISE

The analysis presented in this paper is mainly based on the CH$_2$Cl$_2$-N$_2$O correlation observed during WISE (Figure 3). With a photochemical lifetime of 123 years (Ko et al., 2013) N$_2$O has a much longer lifetime than CH$_2$Cl$_2$. As expected, the correlation is relatively compact for data points with low N$_2$O mixing ratios (i.e., N$_2$O $< 325$ ppb, relatively old, mixed and processed air). Towards younger air masses (N$_2$O $> 325$ ppb) there is a distinct split of the correlation into two branches. In the stratosphere, the upper branch of the CH$_2$Cl$_2$-N$_2$O correlation shows up to $150\%$ enhanced CH$_2$Cl$_2$ mixing ratios compared to data of the lower branch at the same N$_2$O mixing ratios. For N$_2$O $> 328.5$ ppb, data points with low CH$_2$Cl$_2$ mixing ratios even anticorrelate with N$_2$O (Figure 3, inlay). In general, the majority of measurements was obtained in the stratosphere above the thermal tropopause (TP) with an increasing number of observations below the thermal TP for increasing N$_2$O mixing ratios. Thereby mainly air parcels of the lower CH$_2$Cl$_2$-N$_2$O correlation branch are from below the thermal TP.

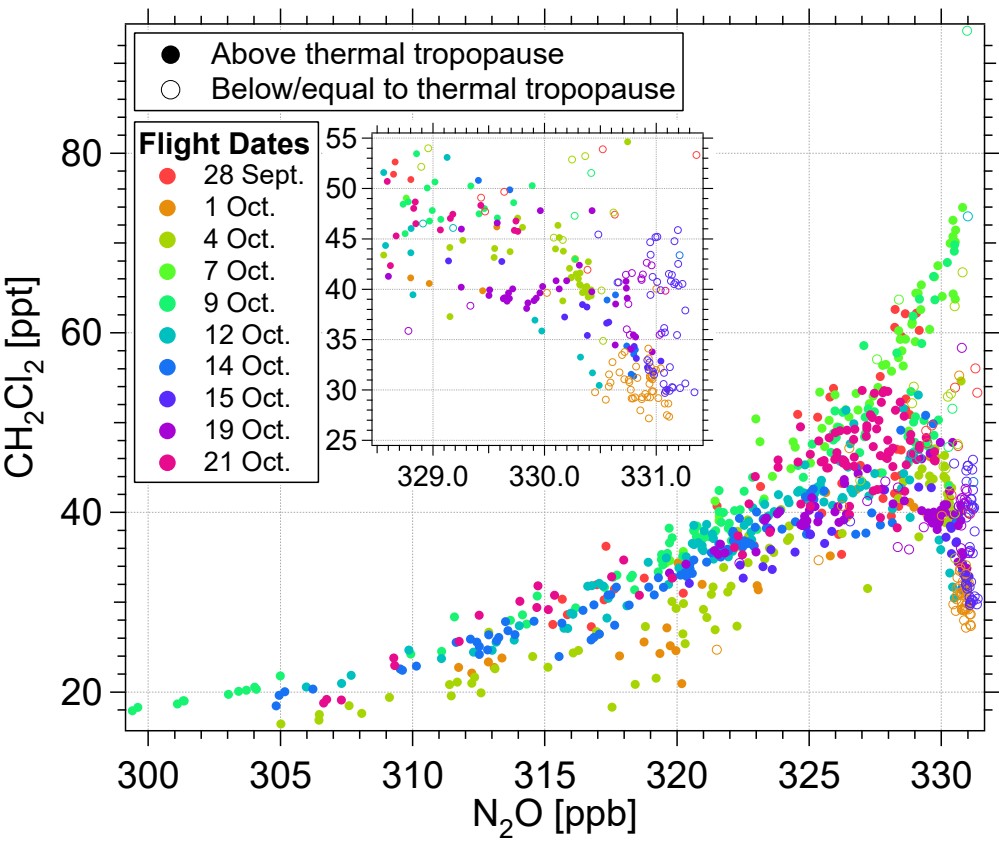

**Figure 3.** $CH_2Cl_2$-$N_2O$ correlation color coded by flight date. The embedded figure shows a detailed magnification of the anticorrelation within the lower branch. Air parcels below the thermal tropopause are marked as open circles, air parcels above by closed circles.

The most frequent convection to potential temperatures on the order of 360 K is expected to originate in the tropics. There-fore, tropical monthly averaged ground-based $CH_2Cl_2$ measurements of AGAGE at Ragged Point, Barbados (Figure 4; Prinn et al., 2018) were analyzed. These AGAGE observations suggest an explanation for the $CH_2Cl_2$-$N_2O$ anticorrelation observed during WISE. The mainly OH induced $CH_2Cl_2$ seasonality results in minimum tropical $CH_2Cl_2$ surface mixing ratios in

September 2017. This September minimum is comparable to WISE data of low $CH_2Cl_2$ mixing ratios in the UTLS region in October (Figure 4, at $N_2O \approx 330.9$ ppb) assuming a transport time from Earth's surface to the UTLS region of a few weeks. The observed decrease of low $CH_2Cl_2$ mixing ratios for increasing $N_2O$ mixing ratios (from older to younger air) agrees well with the decreasing tropical monthly averaged $CH_2Cl_2$ mixing ratios from about July to September 2017, as observed by AGAGE. Extratropical NH ground-based AGAGE observations yield significantly higher $CH_2Cl_2$ mixing ratios than those in

the tropics. It is thus very likely that the lower branch of the $CH_2Cl_2$-$N_2O$ correlation is caused by the tropical Atlantic $CH_2Cl_2$ surface seasonality.



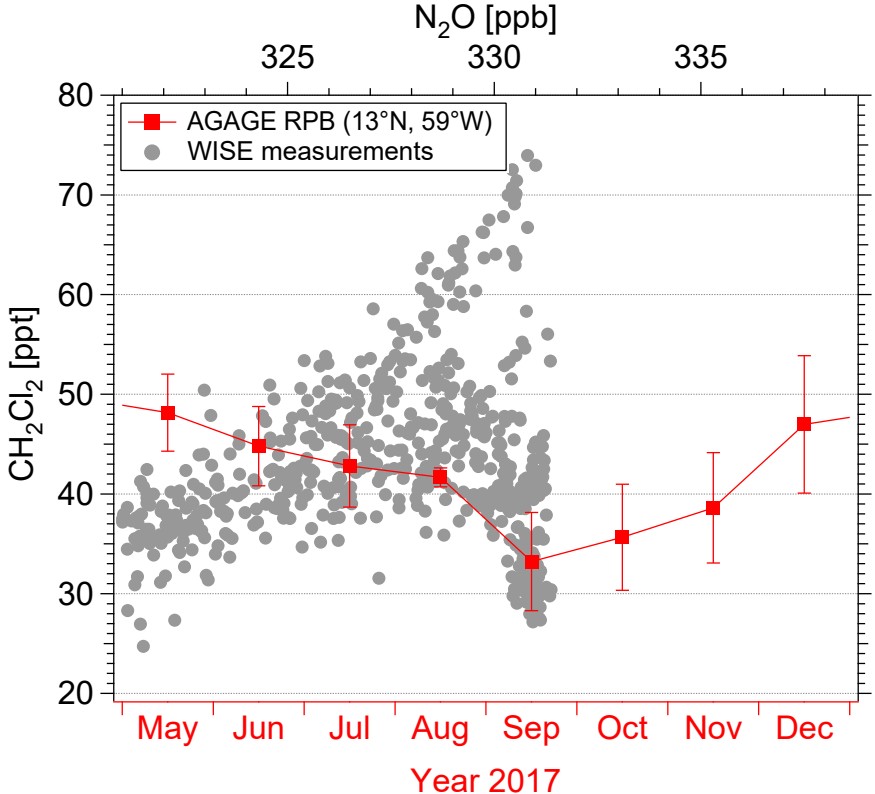

**Figure 4.** Monthly averaged ground-based $CH_2Cl_2$ measurements of AGAGE at Ragged Point, Barbados ($13°$ N Prinn et al., 2018) overlayed by a detailed plot of the $CH_2Cl_2$-$N_2O$ correlation observed during WISE. The AGAGE $CH_2Cl_2$ time series (shown in red) shows $CH_2Cl_2$'s seasonality and is overlayed by the correlation such that the $CH_2Cl_2$ minimum at $N_2O \geq 325$ ppb measured during WISE in October 2017 matches the surface $CH_2Cl_2$ minimum at Barbados in September 2017. Further, the axis of the time series is adjusted so that 1 month corresponds to a change of $\approx 2$ ppb $N_2O$, which is the typical $N_2O$ change per month of age in the UTLS (Andrews et al., 2001). The gradient of the $CH_2Cl_2$-$N_2O$ correlation's lower branch qualitatively fits the temporal variation of the ground-based $CH_2Cl_2$ measurements. The graph illustrates the congruence of ground-based tropical AGAGE $CH_2Cl_2$ measurements and airborne extratropical WISE $CH_2Cl_2$ measurements when assuming that the variation of stratospheric $CH_2Cl_2$ with age (here expressed in terms of $N_2O$ mixing ratio) arises from the temporal variation of $CH_2Cl_2$ at the ground ("tape recorder effect"). Although this simplified view ignores the impact of mixing processes and chemical reduction of $CH_2Cl_2$ it qualitatively explains the lower branch of the correlation curve for air parcels younger than a few months.

### 3.1.1 Correlation filter

In order to separately analyze the $CH_2Cl_2$-$N_2O$ correlation's distinct features, the measurements are filtered relative to a "mean correlation". The "mean correlation" is derived from a quadratic fit applied to the $CH_2Cl_2$-$N_2O$ correlation for $N_2O < 325$ ppb, i.e., where there is no visible split of the correlation (Figure 5, left). In order to identify chemically contrasting air masses of potentially different origin, we focus on the most extreme differences in the chemical composition: Measurements more





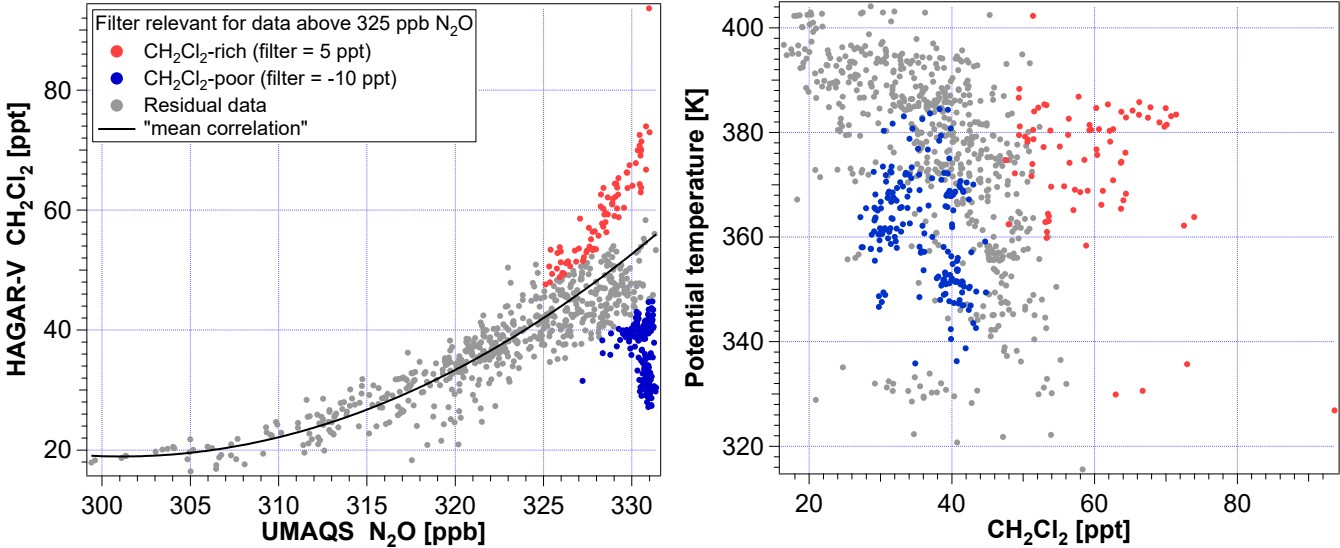

**Figure 5.** Left panel: $CH_2Cl_2$-$N_2O$ correlation color coded with the used definition of the data filter. Red data points are considered measurements of $CH_2Cl_2$-rich air, consisting of measurements more than 5 ppt higher than the "mean correlation" and $N_2O \geq 325$ ppb. Blue data points are considered measurements of $CH_2Cl_2$-poor air, consisting of measurements more than 10 ppt lower than the "mean correlation" and $N_2O \geq 325$ ppb. The "mean correlation" is derived from a quadratic fit to the correlation for $N_2O < 325$ ppb. Right panel: Scatter plot of $CH_2Cl_2$ as a function of the potential temperature and color coded to highlight $CH_2Cl_2$-rich (red) and $CH_2Cl_2$-poor (blue) air. On average the $CH_2Cl_2$-rich air is found at higher potential temperatures than the $CH_2Cl_2$-poor air.

than 5 ppt higher than the "mean correlation" are considered $CH_2Cl_2$-rich air; measurements more than 10 ppt lower than the "mean correlation" are considered $CH_2Cl_2$-poor air. In addition, only measurements with $N_2O > 325$ ppb (corresponding to $\Theta < 390$ K) are considered. The choice of these filter conditions allows the $CH_2Cl_2$-rich and -poor air masses to be clearly

discriminated. It will further be shown below that this filter definition yields a good correspondence with the impact of different air mass origins on the $CH_2Cl_2$-$N_2O$ correlation.

The thus defined measurements of $CH_2Cl_2$-rich air contain on average 66 % higher mixing ratios than those of $CH_2Cl_2$-poor air $((59\pm8)$ ppt vs $(36\pm5)$ ppt, respectively). In addition, the median potential temperature of measurements of $CH_2Cl_2$-rich air is 16.2 K higher than that of $CH_2Cl_2$-poor air (377.8 K vs 361.6 K, respectively). These findings not only indicate tropospheric

intrusions of air from two different source regions into the stratosphere, the different levels of potential temperature also suggest two different transport mechanisms. One is transporting $CH_2Cl_2$-rich air mainly to the top of the LMS ($\Theta \approx 380$ K) and the other is transporting $CH_2Cl_2$-poor air mainly to the middle and lower part of the LMS ($\Theta \approx 350 - 370$ K; Figure 5, right).

### 3.1.2 Impact of different air mass origin on the extratropical UTLS

In order to investigate the impact of different air mass origin on the WISE trace gas measurements, tracers of air mass origin

simulated with CLaMS are analyzed. To focus on fast transport into the LMS in the range of approximately 6 months reflecting



**Table 1.** Median fractions of different surface origin tracers from CLaMS in measurements of $CH_2Cl_2$-rich and $CH_2Cl_2$-poor air parcels, and the respective ratios of the median fractions. The last row shows the median fraction of $\Omega$. $\Omega$ is the sum of all (non-normalized) surface origin tracers of the respective air parcels ($\Omega = \sum_{i=1}^{n=24} \Omega_i$; cf. Section 2.3.1) which is the fraction of an air parcel actually considered in the tracer analysis of $CH_2Cl_2$-rich and -poor air. The fraction $(100 - \Omega)\%$ is the part of an air parcel that has already been in the free atmosphere on 1 May, 2017. The geographical location of each surface origin tracer is given in Figure 2.

| Surface origin tracer | | $CH_2Cl_2$-rich [%] | $CH_2Cl_2$-poor [%] | rich/poor | poor/rich |
|---|---|---|---|---|---|
| W-ITCZ | CAM | 7.3 | 24.9 | 0.29 | 3.40 |
| | TAO | 2.3 | 6.9 | 0.34 | 2.98 |
| | NAF | 3.8 | 6.9 | 0.54 | 1.90 |
| | LSH | 1.9 | 2.3 | 0.83 | 1.20 |
| E-ITCZ | TEP | 9.8 | 10.7 | 0.92 | 1.09 |
| | Neast | 4.3 | 4.1 | 1.04 | 0.96 |
| | Wpool | 7.7 | 5.7 | 1.35 | 0.74 |
| | NWP | 5.0 | 3.4 | 1.49 | 0.67 |
| | TWP | 6.7 | 4.5 | 1.49 | 0.67 |
| | INO | 6.7 | 4.1 | 1.64 | 0.61 |
| SaEA | SEA | 10.8 | 5.8 | 1.87 | 0.54 |
| | NIN | 4.9 | 2.6 | 1.90 | 0.53 |
| | BoB | 7.1 | 3.7 | 1.92 | 0.52 |
| | ECH | 6.0 | 3.1 | 1.94 | 0.52 |
| | IND | 5.2 | 2.6 | 1.97 | 0.51 |
| | TIB | 6.5 | 3.3 | 2.00 | 0.50 |
| $\Omega$ | | 63.2 | 81.4 | 0.78 | 1.29 |

the mean tropospheric lifetime of $CH_2Cl_2$ and $CHCl_3$ (see Section 1), only the fraction of air parcels released from the boundary layer since 1 May is considered. Therefore, in every air parcel each surface origin tracer fraction ($\Omega_i$) is normalized to the sum of all fractions of surface origin tracers ($\Omega = \sum_{i=1}^{n=24} \Omega_i \leq 100\%$) in the air parcel, thus neglecting the fraction of air that was in the free atmosphere at the initialization date of the CLaMS simulation at 1 May, 2017 (i.e., air older than 6 months). The start

time of our simulations on 1 May, 2017, is further chosen to be before the onset of the Asian summer monsoon (pre-monsoon) in order to include all transport processes into the LS impacted by the Asian monsoon circulation. In the following, all analyzed surface origin tracers are normalized as described above, if not stated otherwise.

Further, to work out differences of air mass origin between $CH_2Cl_2$-rich and -poor air, for each surface origin tracer the median fraction in $CH_2Cl_2$-rich air parcels is compared to that in $CH_2Cl_2$-poor air parcels. Surface origin tracers with particularly

high relative median fractions in either $CH_2Cl_2$-rich or -poor air are combined following these two criteria: (1) considered are only surface origin tracers with median fractions $\geq 1\%$ in $CH_2Cl_2$-rich or -poor air parcels, and (2) the median ratio of $CH_2Cl_2$-rich/$CH_2Cl_2$-poor air respectively $CH_2Cl_2$-poor/$CH_2Cl_2$-rich air must be $> 1.8$. With this definition, regions of air





mass origin – as defined for the model simulation – of significantly enhanced influence on measurements of $CH_2Cl_2$-rich (-poor) air relative to $CH_2Cl_2$-poor (-rich) air are combined. Table 1 lists the median fractions in $CH_2Cl_2$-rich and -poor air of
each surface origin tracer fulfilling criterion (1).

The surface origin tracers also fulfilling criterion (2) for $CH_2Cl_2$-rich air are all located in the region of southern and eastern Asia (SaEA) including India, China, and Southeast Asia (cf. Figure 2). The source region of this SaEA tracer is mostly land-based and located in the core region of the Asian summer monsoon (ASM) from where the highest $CH_2Cl_2$ emissions globally are expected (Claxton et al., 2020). The median fraction of the SaEA surface origin tracer in $CH_2Cl_2$-rich air is about twice
that in $CH_2Cl_2$-poor air (41.5 % vs 20.7 %, respectively).

The surface origin tracers fulfilling criterion (1) and (2) for $CH_2Cl_2$-poor air are all located in the tropics along the mostly western part of the Intertropical Convergence Zone (ITCZ) from $120°\,W$ to about $45°\,E$ (W-ITCZ; cf. Figure 2, bottom). The source region of this W-ITCZ tracer includes a large maritime region and is not known for significant $CH_2Cl_2$ emissions. The median fraction of the W-ITCZ surface origin tracer in $CH_2Cl_2$-poor air is about three times higher than in $CH_2Cl_2$-rich air
(40.6 % vs 13.5 %, respectively) with a particularly high contribution from the region of Central America (CAM).

The surface origin tracers fulfilling criterion (1) but not (2) are all geographically connected. To focus on NH regions of air mass origin and because its fraction in both $CH_2Cl_2$-rich and -poor air is very low ($< 2.5\,\%$), the surface origin tracer for the lower southern hemisphere (LSH, Figure 2, top) will not be considered in the following analysis. Without LSH, the third major region of air mass origin significantly influencing the WISE measurements by relatively fast transport mainly includes
an extended region of the summertime ITCZ mostly in the eastern hemisphere and the Pacific Ocean (E-ITCZ), excluding the regions of W-ITCZ and SaEA. The region of this E-ITCZ tracer combines a vast maritime region and areas adjacent to the core region of the ASM. The fractions of the E-ITCZ surface origin tracer in $CH_2Cl_2$-rich and -poor air parcels do not strongly favor either over the other.

With mainly fractions above 40 % the SaEA tracer dominates the $CH_2Cl_2$-$N_2O$ correlation both below 325 ppb $N_2O$ and the
upper branch above 325 ppb $N_2O$ including $CH_2Cl_2$-rich air (Figure 6, bottom left). Towards $CH_2Cl_2$-poor air, the SaEA tracer gradually decreases while the W-ITCZ tracer increases up to fractions above 50 % (Figure 6, bottom right). In fact, both surface origin tracers, SaEA and W-ITCZ, show significant correlations with all WISE $CH_2Cl_2$ measurements at $N_2O > 325$ ppb. Thereby the Spearman's correlation coefficients $R_{SaEA} = 0.7$ and $R_{W\text{-}ITCZ} = -0.72$ indicate a significant monotone but not necessarily linear positive and negative correlation, respectively, with fractions of the SaEA tracer ranging from 8.5 % to 48 %
and those of the W-ITCZ tracer ranging from 9.3 % to 70.8 %.

On the one hand, of all measured air masses entering the LS in the course of NH summer a large fraction originated in southern and eastern Asia. In addition, these air masses are preferably composed of $CH_2Cl_2$-rich air and thus strongly contribute to steepening the $CH_2Cl_2$-$N_2O$ correlation slope (upper branch). On the other hand, young air from the region of the central and western part of the ITZC strongly influences the UTLS with $CH_2Cl_2$-poor air (lower branch). Further, measurements in
between $CH_2Cl_2$-rich and -poor air in the $CH_2Cl_2$-$N_2O$ correlation contain moderate fractions (in the range of $20-40\,\%$) from both regions of air mass origin.



**Figure 6.** CH$_2$Cl$_2$-N$_2$O correlation color coded with the sum of all (non-normalized) surface origin tracers ($\Omega$, top left), the SaEA (bottom left), the W-ITCZ (bottom right) and the E-ITCZ (top right) surface origin tracer. The SaEA, W-ITCZ, and E-ITCZ surface origin tracers are each normalized to the sum of all surface origin tracers (i.e., of each air parcel only the fraction of the sum of all surface origin tracers is considered), thereby neglecting the fraction of older air that has been above the model boundary layer on the simulation's initialization date (1 May, 2017; cf. Section 2.3.1). The CH$_2$Cl$_2$-N$_2$O correlation color coded with the absolute fraction of SaEA and W-ITCZ is shown in Figure A1 in Appendix A.





It has to be noted that the ground-based measurements of AGAGE (Section 3.1) were obtained in the CAM surface origin tracer region, which is included in the W-ITCZ tracer. The extraordinarily high impact of the CAM tracer (24.9 %) on the measurements of $CH_2Cl_2$-poor air strongly supports the comparison made in Section 3.1 and underlines our conclusion of
$CH_2Cl_2$'s tropical Atlantic surface seasonality being reflected in the measurements within the UTLS region.

The influence of the E-ITCZ surface origin tracer on the $CH_2Cl_2$-$N_2O$ correlation is about equal in all air parcels with fractions of around 40 % (Figure 6, top right). This region of air mass origin is thus generally important for the composition of young air masses in the LMS without a specifically strong influence on either $CH_2Cl_2$-rich or -poor air.

### 3.1.3    Results of back-trajectory calculations

The back-trajectories calculated for $CH_2Cl_2$-rich and -poor air are analyzed in two steps. First, the location of maximum rate of change in potential temperature (diabatic ascent rate) along each back-trajectory is derived and the transport time from the measurement to this location is calculated. Second, the back-trajectories are considered up to the point where they reach the model boundary layer. General transport pathways are derived for measurements of $CH_2Cl_2$-rich and -poor air. Within the maximum of 120 days the model boundary layer is reached by 59 out of 80 back-trajectories of $CH_2Cl_2$-rich air (74 %) and
170 out of 189 back-trajectories of $CH_2Cl_2$-poor air (90 %), and only these back-trajectories are analyzed in the following. The median time for an air parcel at the boundary layer to reach the location of measurement is 48 days; $CH_2Cl_2$-poor air in general shows shorter transport times (43 days) than $CH_2Cl_2$-rich air (64 days).

**Locations of maximum diabatic ascent rate and transport times**


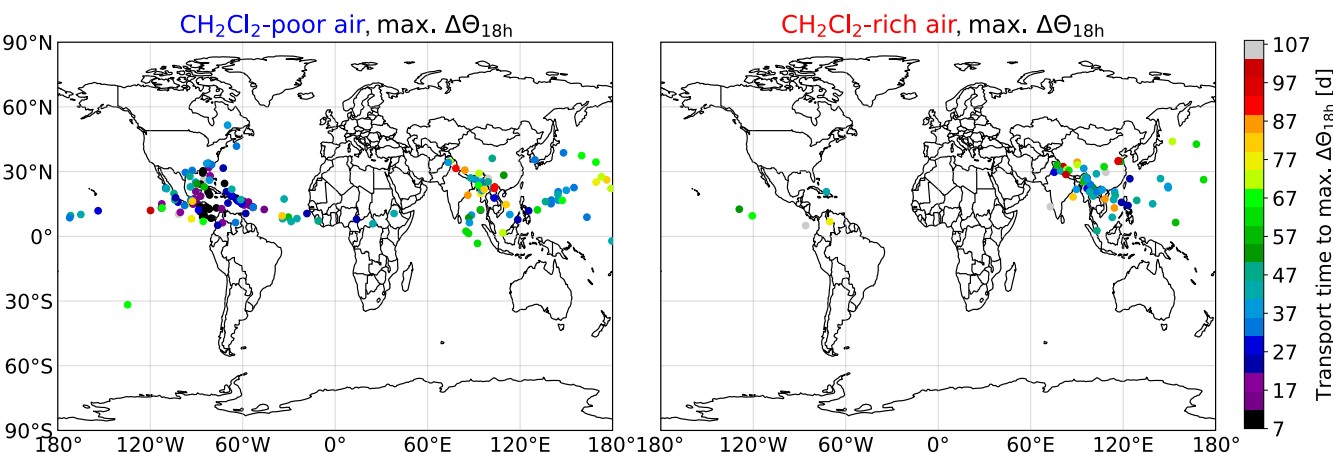

**Figure 7.** Location of maximum change in potential temperature over a time interval of 18 h (max. $\Delta\Theta_{18h}$) along back-trajectories, color coded with the transport time from the location of measurement to the location of max. $\Delta\Theta_{18h}$. Left panel: $CH_2Cl_2$-poor air; right panel: $CH_2Cl_2$-rich air. Shown are the locations of max. $\Delta\Theta_{18h}$ for $CH_2Cl_2$-rich and -poor air from all WISE flights between 28 September and 21 October.





The location of maximum change in potential temperature over a time interval of $18\,\mathrm{h}$ (max. $\Delta\Theta_{18h}$) along each trajectory is used to identify the locations of strong uplift along the trajectories of sampled $CH_2Cl_2$-rich and -poor air. This uplift occurs in the troposphere. Details about the calculation and use of max. $\Delta\Theta_{18h}$ are given by Hanumanthu et al. (2020).

Almost all trajectories of $CH_2Cl_2$-rich air show their max. $\Delta\Theta_{18h}$ above the region of southern and eastern Asia, in particular
above the region of the Tibetan Plateau, northern India, China, and Southeast Asia (Figure 7, right). This uplift mostly occurred $5-10$ weeks prior to the measurement, i.e., in July and August, the peak season of the ASM. This strongly suggests that the measurements of $CH_2Cl_2$-rich air were almost exclusively uplifted within the ASM. There is a clear overlap between the Asian region of concentrated locations of max. $\Delta\Theta_{18h}$ and the region of the SaEA surface origin tracer with the highest relative contribution to air parcels of $CH_2Cl_2$-rich air (cf. Section 3.1.2) suggesting a consistency between trajectory calculations and
the three-dimensional CLaMS simulation.

Of all trajectories related to $CH_2Cl_2$-poor air more than $60\,\%$ exhibit the location of max. $\Delta\Theta_{18h}$ above the region of Central America with the rest being located above southern and eastern Asia and along the ITZC (Figure 7, left). The transport times since the ascent above Central America mainly range between $1-5$ weeks and are much shorter than for those air parcels lifted up above Asia. The main uplift of $CH_2Cl_2$-poor air above Central America thus falls in the time period of late August
and throughout the entire September. This result supports the comparison of $CH_2Cl_2$-poor air with the seasonal minimum $CH_2Cl_2$ mixing ratios observed by AGAGE at Barbados (cf. Figure 4). During the time period of late August and September the region around Central America is influenced by several convective systems: (1) the North American monsoon, (2) the ITCZ, and (3) tropical cyclones, i.e., hurricanes. It is very likely that all of these convective systems contributed to the fast uplift of $CH_2Cl_2$-poor air. The convection systems of the North American monsoon and the ITCZ share many characteristics
and overlap geographically which makes it difficult to distinguish between the two systems (e.g., Siu and Bowman, 2019). The uplift of air parcels by hurricanes can be distinguished and localized more clearly which is analyzed below. The back-trajectory analysis suggests that the most important region for vertical transport of $CH_2Cl_2$-poor air is above Central America, which was also identified as the most significant region of air mass origin of $CH_2Cl_2$-poor air in the three-dimensional CLaMS simulation (cf. Section 3.1.2).


**Case study: convective uplift by hurricane Maria**

In order to investigate the role of tropical cyclones for the transport of Cl-VSLS into the extratropical UTLS region, the locations of max. $\Delta\Theta_{18h}$ were compared with the tracks of several tropical cyclones. Significant matches with the category
5 hurricane Maria (Pasch et al., 2019) were found for back-trajectories of measurements of four WISE flights (on 1, 14, 15, and 19 October). A total of 27 trajectory locations of max. $\Delta\Theta_{18h}$ agreed within a time window of $0.2$ days and a $1°$ radius with the center of hurricane Maria at some point along its track (Figure 8). The $1°$ radius of tolerance was chosen because it corresponds to the spatial resolution of the ERA-Interim reanalysis data used for the trajectory calculation, as well as (roughly) to the hurricane's radius from its core. This analysis directly links 27 WISE measurements to the convection of hurricane Maria
with transport times ranging between one week and one month.





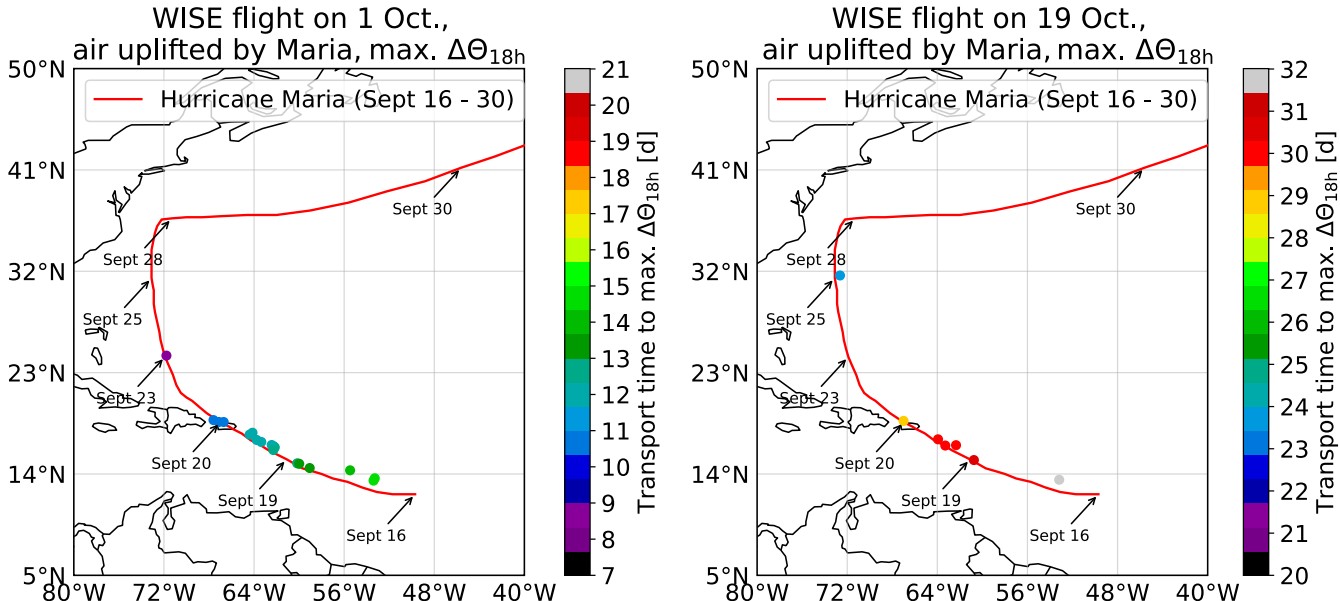

**Figure 8.** Trajectory locations of max. $\Delta\Theta_{18h}$ color coded with the transport time from the location of measurement to the position of max. $\Delta\Theta_{18h}$ along the trajectory. The red line indicates the storm track of the center of hurricane Maria (Pasch et al., 2019) with arrows marking the location at which the hurricane resided at the indicated date. Left panel: Flight on 1 Ocober; right panel: Flight on 19 October.

Interestingly, $CH_2Cl_2$ mixing ratios of measurements linked to hurricane Maria positively correlate with transport time ($R_{Pearson} = 0.85$; Figure 9, top right). Those air samples related to short transport times contain the lowest $CH_2Cl_2$ mixing ratios at $N_2O > 325$ ppb measured during WISE (Figure 9, top left). According to the back-trajectories, most of the air parcels lifted up by hurricane Maria left the model boundary layer above the tropical Atlantic where $CH_2Cl_2$ sources are small (Figure

9, bottom). In that region the seasonal minimum of $CH_2Cl_2$ mixing ratios is in September (cf. Section 3.1). This implies that when air masses lifted up by hurricane Maria mix, they can only increase their $CH_2Cl_2$ mixing ratio, i.e., mixing with air of higher $CH_2Cl_2$ mixing ratio. Air parcels related to longer transport times did not take a direct path to the extratropics after being lifted up by hurricane Maria and rather followed the subtropical jet stream eastwards around the globe, thereby enhancing the chances of mixing with air of higher $CH_2Cl_2$ mixing ratios.

In general, despite being a significant source of convection, hurricane Maria did not contribute to the transport of enhanced $CH_2Cl_2$ mixing ratios into the stratosphere and rather led to the transport of $CH_2Cl_2$-poor air, a consequence of $CH_2Cl_2$'s tropical Atlantic boundary layer seasonality. This result is consistent with the lack of strong $CH_2Cl_2$ sources in the oceanic region of convection. Nevertheless, our analysis shows that large hurricanes can provide a fast transport into the extratropical UTLS. Below in Section 4 it is discussed how tropospheric air masses observed on 1 October are mixed into the LS a few days

after the observation. For instance, this is of particular importance for brominated short-lived substances (e.g., $CH_2Br_2$ and $CHBr_3$) that have a high ODP and some of their largest emission sources located in tropical oceans (e.g., Hepach et al., 2015; Rotermund et al., 2021).



**Figure 9.** Upper left panel: Detailed graph of the $CH_2Cl_2$-$N_2O$ correlation color coded with the transport time since uplift by hurricane Maria. The $CH_2Cl_2$-$N_2O$ correlation in the background is plotted in different shades of gray indicating the measurements of $CH_2Cl_2$-rich air (dark gray) and $CH_2Cl_2$-poor air (light gray). Upper right panel: Correlation of $CH_2Cl_2$ and the transport time since uplift by hurricane Maria, color coded according to the air parcels' location above (blue-green) or below/equal to the thermal tropopause (orange). Lower panel: back-trajectories from the location of measurement to the model boundary layer of air parcels lifted up by hurricane Maria, color coded with the potential temperature of the air parcels at the respective trajectory location.



**Analysis of transport pathways**

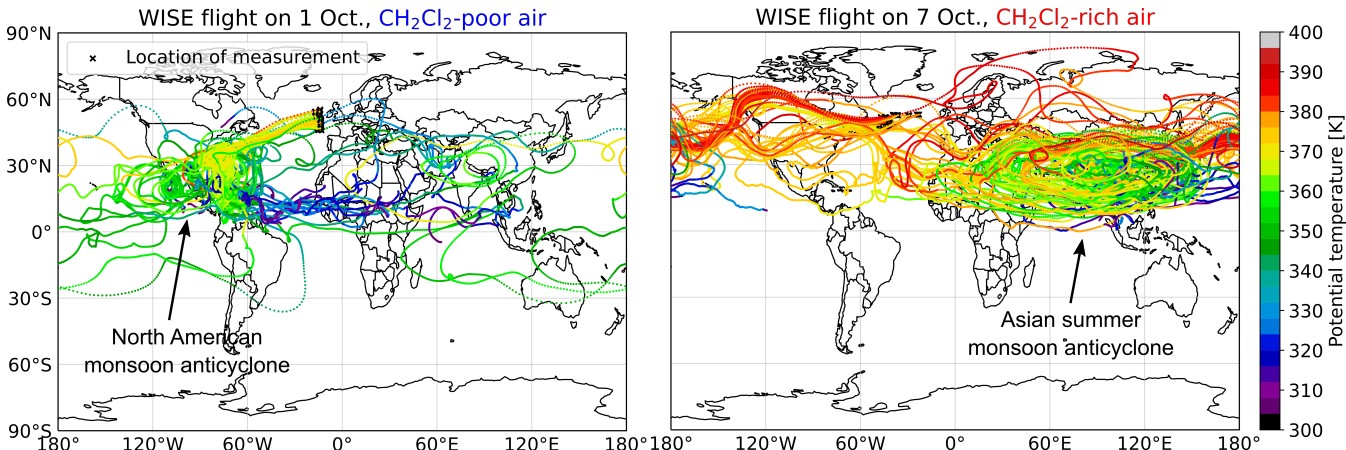

**Figure 10.** Back-trajectories from the location of measurement to the model boundary layer for CH$_2$Cl$_2$-poor air sampled on 1 October (left panel) and for CH$_2$Cl$_2$-rich air sampled on 7 October (right panel) color coded with the potential temperature of the trajectory. The trajectories of CH$_2$Cl$_2$-poor air show an uplift above Central America to about 365 K, an isentropic northward drag towards an anticyclonic system above North America, an eastward breakout, and a direct and isentropic pathway into the extratropics above the Atlantic Ocean. The trajectories of CH$_2$Cl$_2$-rich air show an uplift above southern and eastern Asia up to about 360 K with further upward transport by the ASMA to about 385 K and a breakout eastward following the subtropical jet stream until they quasi-isentropically enter the extratropics above the eastern Pacific or western Atlantic Ocean.

The back-trajectories from the location of measurement to the model boundary layer are analyzed to identify the main transport pathways of CH$_2$Cl$_2$-rich and -poor air into the stratosphere. As representative examples, Figure 10 shows the trajectories of the WISE flights on 1 October (left) and 7 October (right) for CH$_2$Cl$_2$-poor and -rich air, respectively.

     Almost all trajectories of CH$_2$Cl$_2$-rich air show the following general pathway: The air parcels are convectively lifted up above southern and eastern Asia to $\Theta \approx 360$ K. Further ascent of the air parcels occurs in a clockwise upward spiraling
motion (Vogel et al., 2019), following the dynamics of the Asian summer monsoon anticyclone (ASMA) mainly to potential temperatures in the range of $370 - 400$ K. Preferably within this potential temperature range, the air parcels break out of the ASMA eastwards (e.g., Honomichl and Pan, 2020), following the subtropical jet stream at about $40°$ N until, eventually, they quasi-isentropically enter the extratropics above the eastern Pacific or western Atlantic Ocean (e.g., Vogel et al., 2014, 2016). All trajectories suggest that the air parcels arrived from the west to the location of measurement and most trajectories suggest
a slight diabatic descent of up to $-10$ K in the extratropics a few days before the measurement.

     The majority ($> 60\,\%$) of trajectories of CH$_2$Cl$_2$-poor air show a strong uplift above the region of Central America up to potential temperatures mainly in the range of $360 - 370$ K. After convection, the trajectories experience a northward drag towards an anticyclonic structure located above North America, and most of these trajectories further directly enter the extratropics above the Atlantic Ocean or the North American east coast leading to short transport times to the location of measurement.



Some trajectories lifted up above the region of Central America eventually follow the subtropical jet stream eastwards around
the globe before entering the extratropics. This significantly increases the transport time of an air parcel by about 3 weeks and
has the potential to cause it to descend by up to about $-10\,\mathrm{K}$ as indicated by the back-trajectory calculations. However, this
concerns a minority of air parcels and has only a minor effect on the median transport time of all observed air parcels lifted up
above the region of Central America (max. $\Delta\Theta_{18h}$ between $0° - 30°\,\mathrm{N}$ and $50° - 120°\,\mathrm{W}$) which is still 44 days shorter than
that of the observed air parcels lifted up above southern and eastern Asia (max. $\Delta\Theta_{18h}$ between $0° - 40°\,\mathrm{N}$ and $60° - 160°\,\mathrm{E}$;
24 vs 68 days, respectively). Below, in Section 4, the transport pathway from Central America to the Ex-LS is discussed in
more detail.

The analysis of the entire set of back-trajectories shows that for the majority of measurements there are two distinct transport
pathways into the Ex-LS. $CH_2Cl_2$-rich air is transported by the ASMA and $CH_2Cl_2$-poor air mainly by convection above
Central America, which includes the North American monsoon, the ITCZ and hurricanes. In general, air parcels are lifted
up to similar potential temperature levels by the convection of the ASM in Asia and the convection above Central America.
The key difference yielding the observed higher potential temperatures of $CH_2Cl_2$-rich air from Asia compared to those of
$CH_2Cl_2$-poor air from Central America is the additional uplift by the ASMA following the convection within the ASM (e.g.,
Müller et al., 2016; Brunamonti et al., 2018; Vogel et al., 2019; von Hobe et al., 2021). The slow upward spiraling dynamics
within the ASMA also contributes to the longer transport time from the boundary layer to the location of the measurement
of $CH_2Cl_2$-rich air compared to that of $CH_2Cl_2$-poor air. Another aspect adding to the different transport times is the longer
transport pathway from Asia because air masses were always observed to reach the location of measurement from the west.

### 3.2  Comparison of $CH_2Cl_2$ and $CHCl_3$

In this section the results of the $CH_2Cl_2$ analysis are used to investigate $CHCl_3$ data measured during WISE. Figure 11 (left)
shows the $CHCl_3$-$N_2O$ correlation color coded to highlight air parcels of $CH_2Cl_2$-rich (red) and -poor (blue) air (cf. Sec-
tion 3.1.1). In general, the $CHCl_3$-$N_2O$ correlation reveals similar but less clearly pronounced structures as observed for the
$CH_2Cl_2$-$N_2O$ correlation. The $CHCl_3$-$N_2O$ correlation similarly is less compact for higher $N_2O$ mixing ratios. However, a
distinct split of the correlation, as observed for $CH_2Cl_2$, is not clearly visible, but a broad scatter on the $CHCl_3$ axis with
mixing ratios in the stratosphere being enhanced by up to $100\,\%$ compared to the lowest measurements at similar $N_2O$ values.
Measurements of $CH_2Cl_2$-rich air also show clearly enhanced $CHCl_3$ mixing ratios and measurements of $CH_2Cl_2$-poor air
also contain the lowest $CHCl_3$ mixing ratios at given $N_2O$ values. Nevertheless, there are a few significant differences which
will be analyzed in the following.

The seasonal cycle of $CHCl_3$ is less pronounced but in phase with that of $CH_2Cl_2$ (cf. Figure B1 in Appendix B). Based on a
comparison with ground-based AGAGE observations, $CHCl_3$ data for measurements of $CH_2Cl_2$-poor air between $N_2O$ values
of $229.5\,\mathrm{ppb}$ and $331\,\mathrm{ppb}$ reflect $CHCl_3$'s tropical surface seasonality as it similarly was observed for $CH_2Cl_2$.

In our data, high $CH_2Cl_2$ concentrations coincide with high $CHCl_3$ concentrations in many, but not in all cases. There are
examples of high $CHCl_3$ concentrations where $CH_2Cl_2$ concentrations are relatively low. This suggests that air from regions
with relatively stronger $CHCl_3$ than $CH_2Cl_2$ sources was measured. However, air masses of $CH_2Cl_2$-rich air clearly stand out





by their elevated CHCl$_3$ mixing ratios in the region of $\Theta \approx 380\,\text{K}$ (Figure 11, center). Based on the results of Section 3.1.3, we

therefore suggest that the ASMA is the dominant factor also for the transport of enhanced CHCl$_3$ mixing ratios to the Ex-LS at $\Theta \approx 380\,\text{K}$.

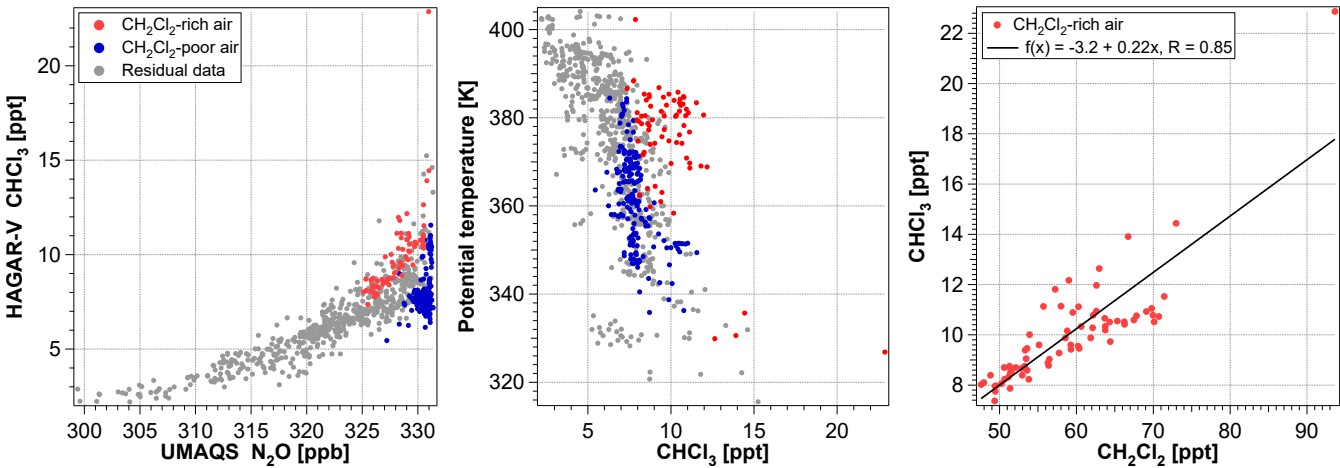

**Figure 11.** CHCl$_3$-N$_2$O correlation (left) and CHCl$_3$ as a function of potential temperature (center) color coded to highlight measurements of CH$_2$Cl$_2$-rich and -poor air; CHCl$_3$-CH$_2$Cl$_2$ correlation of measurements of only CH$_2$Cl$_2$-rich air (right).

Due to their similar photochemical lifetime CH$_2$Cl$_2$ and CHCl$_3$ are expected to linearly correlate in the stratosphere; however varying correlation slopes can arise due to different emission ratios in the source regions defining the respective composition of the air parcel. The measurements of CH$_2$Cl$_2$-rich air show a significant positive linear correlation with CHCl$_3$ (Figure 11, right)

suggesting sources or source regions with similar emission ratios of these species. Due to the strong evidence for CH$_2$Cl$_2$-rich air being significantly affected by anthropogenic sources, the significant positive correlation with CHCl$_3$ suggests that this also holds for CHCl$_3$. The highest anthropogenic emissions of CHCl$_3$ are expected to originate from China (Fang et al., 2018), which is within the region of sources particularly impacting the air masses of CH$_2$Cl$_2$-rich air analyzed here (Section 3.1.2). This suggests a significant anthropogenic impact that clearly enhances CHCl$_3$ concentrations in the upper LMS.

At a closer look, the CHCl$_3$-CH$_2$Cl$_2$ correlation in Figure 11 (right) reveals two correlation lines with different slopes. The nature of the different slopes can be better understood when looking at the CHCl$_3$-CH$_2$Cl$_2$ correlation of all WISE measurements color coded with CLaMS's surface origin tracers (Figure 12; left and center). The CHCl$_3$-CH$_2$Cl$_2$ correlation fans out towards higher mixing ratios giving the impression of several correlation lines with different slopes. The data points forming the steepest correlation slope show the highest W-ITCZ tracer fractions and the lowest SaEA tracer fractions while for

the data points forming the lowest correlation slope the opposite is the case. The CHCl$_3$-CH$_2$Cl$_2$ correlation slope thus flattens with increasing entry of air masses originating from southern and eastern Asia. This suggests larger CHCl$_3$:CH$_2$Cl$_2$ emission ratios in the region of the central and western ITCZ region (with presumably mostly biogenic sources) than in southern and eastern Asia (where anthropogenic sources likely dominate).





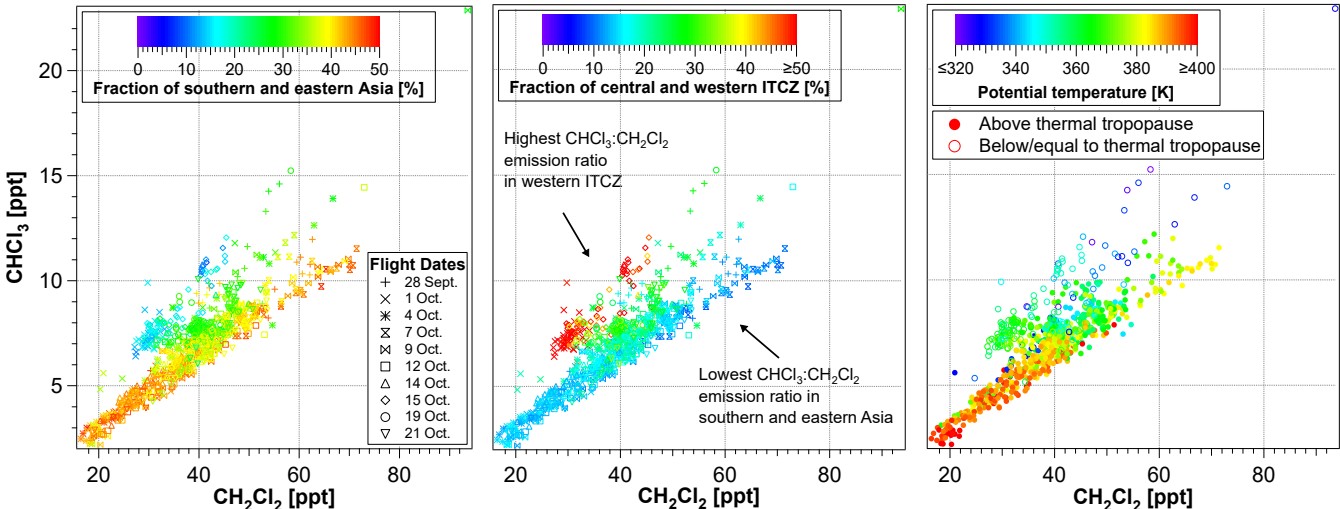

**Figure 12.** $CHCl_3$-$CH_2Cl_2$ correlation color coded with the SaEA (left) and the W-ITCZ (center) surface origin tracer, and the potential temperature (right). The correlation exhibits different correlation slopes clearly depending on the origin of air. A large impact of sources from southern and eastern Asia coincides with a low correlation slope dominating the larger part of the correlation. A large impact of sources from the central and western part of the ITCZ coincides with the steepest correlation slope implying a relatively larger $CHCl_3$:$CH_2Cl_2$ emission ratio in this region compared to the emission ratio in southern and eastern Asia. The highest $CHCl_3$ mixing ratios and the majority of measurements with the steepest correlation slope were observed in the tropopause region and the upper troposphere.

Compared to the lowest correlation line, the wider range of surface origin tracer fractions apparent in the correlation lines
with steeper slopes might be due to the different strengths of seasonality of $CH_2Cl_2$ and $CHCl_3$ possibly affecting the linear relationship between $CHCl_3$ and $CH_2Cl_2$. In addition, the highest mixing ratios pertain to the steeper correlation lines. However, all of those were observed at low potential temperature levels and below the thermal TP (Figure 12, right) where mixing ratios can easily exceed those in the stratosphere above. The evolution of tropospheric air masses at potential temperatures above 350 K as shown in the steeper correlation line is discussed further below in Section 4. In summary the analysis suggests
clear similarities between $CH_2Cl_2$ and $CHCl_3$ when emitted by anthropogenic sources and differences between the two species mainly due to additional (presumably biogenic) $CHCl_3$ sources.

## 4  Discussion

Figure 13 shows measured and simulated WISE tracers as a function of equivalent latitude and potential temperature. It illustrates the air masses of enhanced $CH_2Cl_2$ and $CHCl_3$ that were transported from southern and eastern Asian sources by the
ASMA to potential temperatures of around 380 K and to the Ex-LS. At slightly lower potential temperatures and equivalent latitudes we observed particularly low $CH_2Cl_2$ mixing ratios and partly low $CHCl_3$ mixing ratios. The corresponding air masses were sampled at potential temperatures mainly above 360 K but mostly below the thermal tropopause and were uplifted from Central American as well as tropical Atlantic and Northern African source regions via convection by hurricanes, by the ITCZ




and the North American monsoon and transported further towards the location of measurement at higher latitudes; therefore
these air masses are characterized by low values of equivalent latitude.

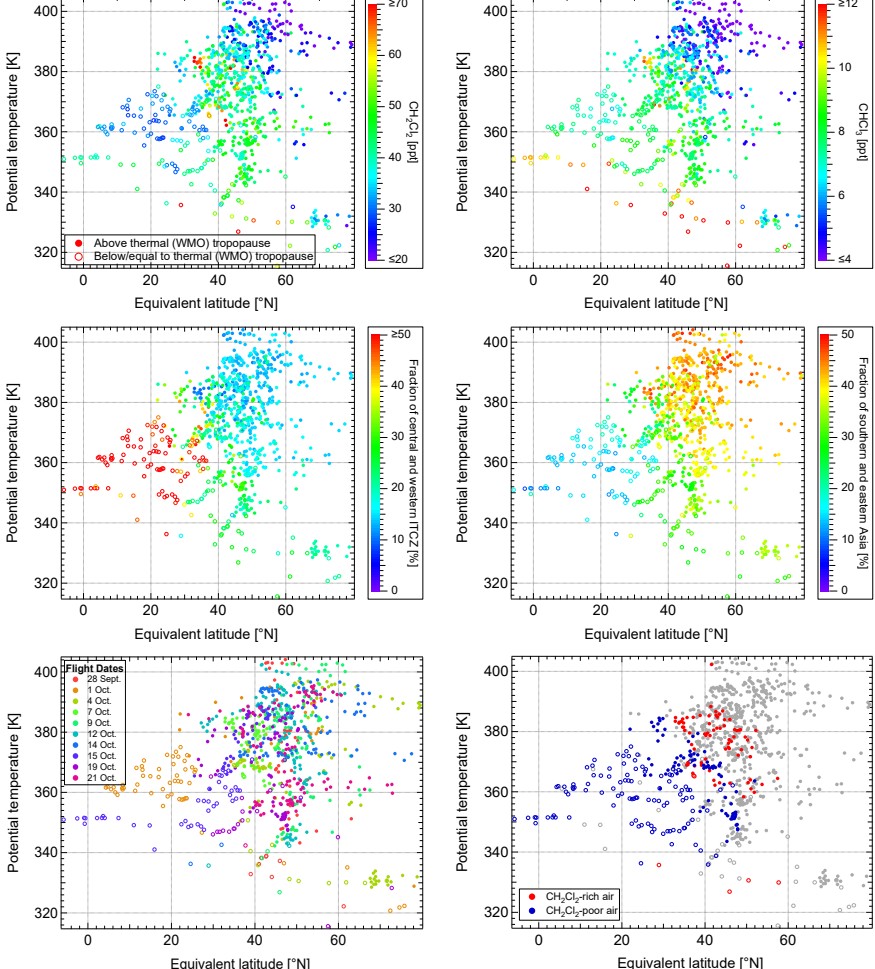

**Figure 13.** Measured (top) and simulated (middle) tracers of the WISE flights (bottom, left) from 28 September to 21 October, 2017, as a function of equivalent latitude and potential temperature. Note that equivalent latitudes $\leq 0°$ N are likely calculated artifacts due to a negative bias induced by convection (Pan et al., 2012). The open symbols indicate a measurement location below or equal to the thermal tropopause (TP) and the location of full symbol data points is above the thermal TP. The coloration of the panel at bottom right correspond to the lower branch (blue) and to the upper branch (red) of the $CH_2Cl_2$-$N_2O$ correlation (referred to as $CH_2Cl_2$-poor and -rich air, respectively; cf. Section 3.1.1). $CH_2Cl_2$-poor air measured at potential temperatures generally up to $375\,\mathrm{K}$ mostly originated from the Central American as well as tropical Atlantic and Northern African boundary layer (central and western ITCZ) and has mostly not (yet) entered the lower stratosphere at the time of measurement. $CH_2Cl_2$-rich air is strongly influenced by air masses from southern and eastern Asia and was measured almost exclusively in the extratropical lower stratosphere. Note that the fraction of surface origin tracers given in the graphs is not an absolute fraction of the whole air parcels but of the air masses younger than 6 months within the air parcels (cf. Section 3.1.2). The two different transport pathways from the boundary layer into the extratropical UTLS region are described in the text.





The presented distribution of air masses from different source regions in the NH UTLS is in good agreement with a similar study by Rotermund et al. (2021) also based on WISE measurements, but using bromine observations. However, the lower stratospheric region of high bromine concentrations from Asian source regions described by Rotermund et al. (2021) is at lower potential temperatures and higher equivalent latitudes than the $CH_2Cl_2$-rich air described in the present paper. This could

be due to relatively stronger (mostly biogenic) bromine emission sources in the adjacent region of the ASM compared to the mostly anthropogenic $CH_2Cl_2$ emission sources mainly located in the core region of the ASM. In addition, in the present paper the first five research flights in September are not analyzed in contrast to the study by Rotermund et al. (2021). Nevertheless, compared to the very short-lived bromine species analyzed by Rotermund et al. (2021), the combination of a longer lifetime, highly significant Asian emission sources, and a strong seasonal cycle clearly benefits the use of $CH_2Cl_2$ observations to derive

details about the different transport mechanisms and pathways from the source region into the NH summertime UTLS.

Further, elevated quantities of peroxyacetyl nitrate (PAN) were measured in the NH LMS during the WISE flight on 13 September 2017 by the GLORIA instrument with main sources in South Asia and Southeast Asia uplifted by the ASMA (Wetzel et al., 2021). Moreover, the transport pathway into the LS via the ASMA derived from back-trajectories of air originating in southern and eastern Asia was also observed for other measurements taken in the NH UTLS over Europe and the Atlantic

Ocean during the HALO TACTS campaign in August and September 2012 (Vogel et al., 2014, 2016; Müller et al., 2016; Rolf et al., 2018). In the present study, we have for the first time directly related this transport pathway to in situ Cl-VSLS measurements in the LS and observed that air masses strongly enhanced in $CH_2Cl_2$ and $CHCl_3$ are rather rapidly transported to the top of the NH LMS at about 380 K by this pathway. This finding supports the modeled results of Claxton et al. (2019) who show that Cl-VSLS sources located in tropical Asia have a higher potential for stratospheric ozone depletion than those from

any other source region. In addition, $CHCl_3$ has significant biogenic sources (Engel et al., 2018). Our study suggests that not only the enhanced $CH_2Cl_2$ mixing ratios but also the enhanced $CHCl_3$ mixing ratios observed at about 380 K are significantly impacted by anthropogenic sources which are expected to be strongest in the region of southern and eastern Asia (Claxton et al., 2020) and eastern China (Fang et al., 2018), respectively.

There are several studies analyzing the transport of air into the stratosphere by convection above Central and North America

and its further distribution by the North American monsoon anticyclone (NAMA) (e.g., Gettelman et al., 2004; Ray, 2004; Pittman et al., 2007; Weinstock et al., 2007; Herman et al., 2017; Wang et al., 2021; Clapp et al., 2021). Studies based on observational data mostly focus on the equatorward transport of air out of the NAMA (e.g., Gettelman et al., 2004; Ray, 2004; Pittman et al., 2007; Weinstock et al., 2007). Mainly model based simulations (e.g., Li, 2005; Ploeger et al., 2013; Nützel et al., 2019) and a study based on satellite observations (Clapp et al., 2021) have addressed north- and northeastward outflow

of the NAMA. Here, we have described a transport pathway from the marine boundary layer in Central America and the tropical Atlantic into the NH midlatitude UTLS based on in situ Cl-VSLS observations. To our knowledge in situ Cl-VSLS measurements have never been used before to study transport from the tropics south of the NAMA to the midlatitude UTLS. The horizontal advection northwards following the convection in the tropics might be related to the NAMA as described in a recent model study by Wang et al. (2021). Further, Clapp et al. (2021) observed the main outflow (68 %) of the NAMA to be

in north-eastward direction between 35° N and 60° N in July and August. In good agreement with both studies, the trajectories



of WISE measurements uplifted above Central America show a northward drag towards a location of circular movement resembling the NAMA with a north-eastward escape from the circulation (cf. Figure 10, left).

Many studies have addressed the topic of tropospheric intrusions into the stratosphere above Central and North America by analyzing direct injections via overshooting convection (e.g., Smith et al., 2017; Anderson et al., 2017; Herman et al., 2017;

Cooney et al., 2018; Clapp et al., 2019, 2021). The results of Wang et al. (2021) suggest that for air uplifted in the region of Central America (15° N to 20° N) overshooting convection is not the main transport pathway into the stratosphere during NH summer. Above this region of Central America the TP usually is at potential temperatures on the order of 380 K and most convection in this region does not uplift air higher than that. Drawn towards the NAMA, the uplifted air gets further transported horizontally to higher latitudes (Wang et al., 2021) where the TP is $1 - 2$ km higher than usual due to the NAMA (Schoeberl

et al., 2020). Further horizontal transport northeastward out of the anticyclone, as shown in our study, eventually causes the tropical air masses (being on high potential temperatures) to isentropically enter the LS.

Particularly during the WISE flight on 1 October when we sampled the largest number of air parcels uplifted by hurricane Maria the measurements were highly impacted by air originating in the region of Central America (cf. Figure 13). With a median transport time of 18 days these measurements agree well with the fast transport pathway into the stratosphere described

by Wang et al. (2021). However, despite the fact that we observed most of these air masses at latitudes around 50° N and potential temperatures in the range of $350 - 370$ K, the majority of measurements were below the thermal TP. Figure 14 shows the meteorological situation of this particular WISE flight on 1 October based on ERA-Interim reanalysis data (Dee et al., 2011). Obviously, the air masses breaking out of the anticyclone above the North American east coast (cf. Figure 10, left) turned into a streamer carrying a local high TP to higher latitudes. A few days after our observation, this streamer became

unstable and mixed into the LS. This implies that tropical air lifted up by hurricanes and other convective systems in the region of Central America can enter the Ex-LS quasi-isentropically during NH autumn even if the convection in the tropics has not transported the air above the TP. This transport pathway, described by Wang et al. (2021) as the fastest and most efficient transport of tropical tropospheric air into the North American LMS region, has been corroborated for the first time on the basis of in situ Cl-VSLS observations by the present study. We have thereby shown that tropical surface mixing ratios of VSLS

from the region of Central America and the Atlantic Ocean can be efficiently transported into the Ex-LS during the late North American monsoon season.

Our results further show a regional dependency of the slope of the NH UTLS $CHCl_3$-$CH_2Cl_2$ correlation. Observations by Say et al. (2019) in the Indian boundary layer suggest a similarly flat $CHCl_3$-$CH_2Cl_2$ correlation slope as observed during WISE for air masses strongly impacted by Asian sources. However, AGAGE measurements (Prinn et al., 2018) from Barbados

in 2017 show seasonally varying $CHCl_3$-$CH_2Cl_2$ correlation slopes not necessarily matching the steep slope observed for air masses strongly impacted by Central American source regions during WISE. The here presented regional dependency of the $CHCl_3$-$CH_2Cl_2$ correlation slope could thus be a seasonal phenomenon depending on transport efficiency and locally varying emissions. Obviously, more in situ observations of $CH_2Cl_2$ and $CHCl_3$ in the UTLS (particularly in different seasons) and ground-based (particularly in Asia) are needed to better understand the correlation behavior of $CH_2Cl_2$ and $CHCl_3$ in the

UTLS.





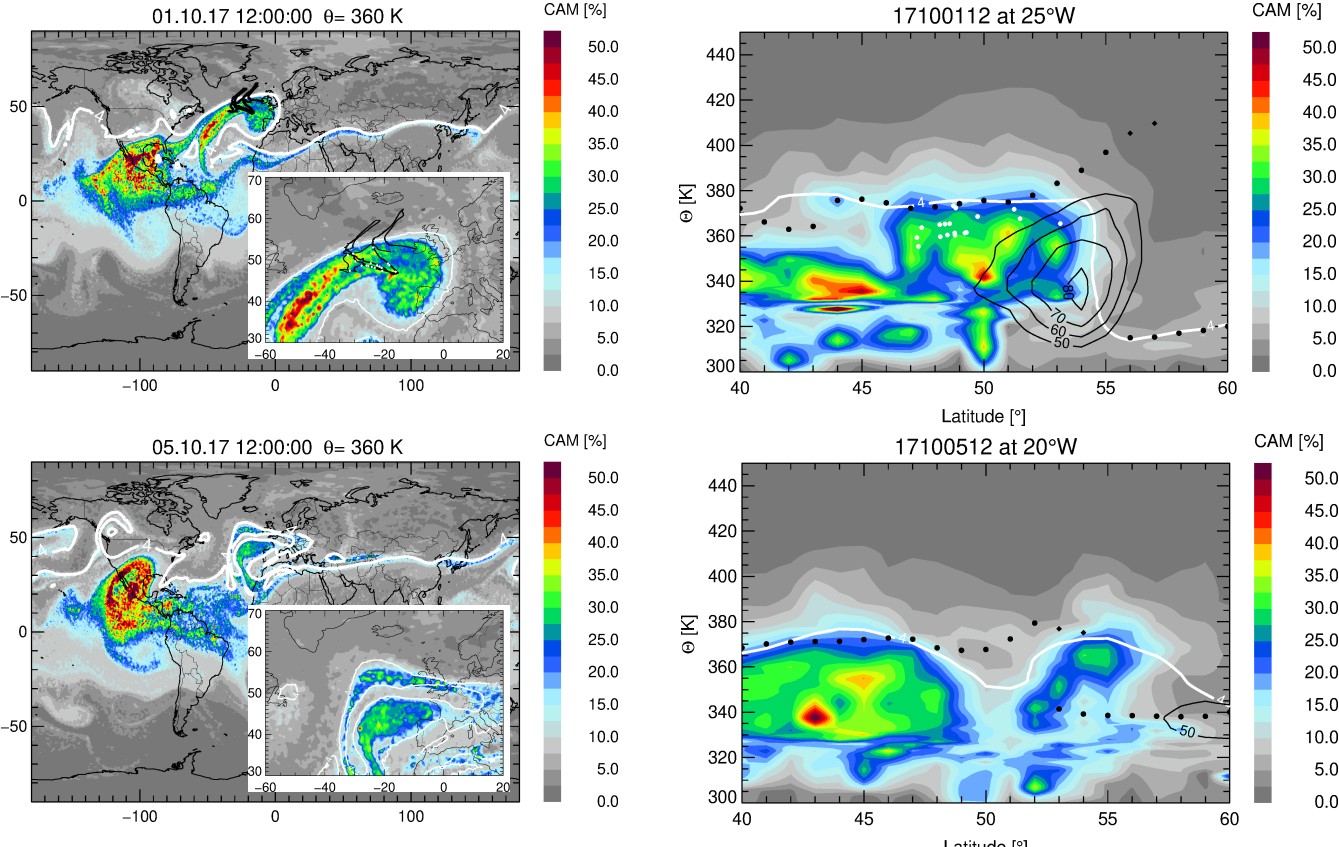

**Figure 14.** Meteorological situation at noon time for the WISE flight on 1 October (top) and four days later (bottom) using ERA-Interim reanalysis data. The colors indicate the surface origin tracer of the region of Central America (CAM) at 360 K of potential temperature (left panels) and as a vertical cross section at 20° W (top, right) and 25° W (bottom, left). The flight track (transferred to noontime) is shown as a black line on the isentropic view (top, left); on the vertical cross sections the black lines indicate zonal wind speed. White dots mark the measurement location of air lifted up by hurricane Maria. Note that these measurement locations, as well as the shown flight path, are not necessarily located exactly at 360 K (top, left) or at 25° W (top, right). Black dots indicate the location of the first thermal tropopause (TP), black diamonds indicate the second thermal TP; the white line shows the 4 PVU surface. The plots show the probing of a high TP streamer of air originating in Central America at midlatitudes on 1 October. Four days later, the streamer became unstable and a large volume mixed into the LS above the thermal TP. This figure illustrates the intrusion of tropical air into the LS: Air within the streamer has been lifted up by a hurricane into the TTL and was further transported to higher latitudes by an upper level anticyclone above North America (cf. Figure 10, left) to be finally mixed into the LS by Rossby wave breaking.

## 5    Conclusions

We have presented the first study on transport of Cl-VSLS into the Ex-LS based on tracer-tracer correlations using in situ Cl-VSLS observations. A schematic of the transport pathways we deduced in this study is shown in Figure 15. Our measurements





in the LS above the midlatitude Atlantic Ocean in autumn 2017 revealed up to $150\%$ enhanced $CH_2Cl_2$ and up to $100\%$
enhanced $CHCl_3$ mixing ratios compared to measurements with similar $N_2O$ mixing ratios, i.e., similarly processed air. In the
stratosphere, the samples of $CH_2Cl_2$-rich air also contained most of the observed $CHCl_3$-rich air and the highest mixing ratios
of both species detected in the stratosphere at $\Theta \approx 380\,K$. In contrast to $CHCl_3$, $CH_2Cl_2$ is almost exclusively of anthropogenic
origin (Engel et al., 2018) and a good correlation of $CH_2Cl_2$-rich air with $CHCl_3$-rich air suggests anthropogenic sources also
impacting the enhanced $CHCl_3$ mixing ratios observed in the region at about $380\,K$ potential temperature. Using a global
three-dimensional Lagrangian model simulation we have shown a particularly strong influence of southern and eastern Asian
sources in these air masses of enhanced $CH_2Cl_2$ and $CHCl_3$ mixing ratios.

Back-trajectory calculations agree well with the global three-dimensional model simulation and reveal a distinct transport
pathway via the Asian summer monsoon for the air masses of enhanced $CH_2Cl_2$ and $CHCl_3$ mixing ratios. This pathway
implies convection over southern and eastern Asia to about $360\,K$ potential temperature $4-10$ weeks prior to the measurement
(i.e., in July and August) and a slow circular upwelling to $370-400\,K$ in the ASMA. The observed air masses broke out of the
anticyclone eastward following the subtropical jet stream before entering the extratropics above the eastern Pacific or western
Atlantic Ocean (horizontal red arrow in Figure 15). This transport pathway was also observed during the HALO TACTS
campaign in 2012 (Vogel et al., 2014, 2016; Müller et al., 2016; Rolf et al., 2018).

Our results provide observational evidence to support the findings of model studies (e.g., Hossaini et al., 2015, 2019; Claxton
et al., 2019) which connect the recent increase in Asian $CH_2Cl_2$ and $CHCl_3$ emissions (e.g., Leedham-Elvidge et al., 2015;
Oram et al., 2017; Feng et al., 2018; Fang et al., 2018; Adcock et al., 2021) with an increase in the contribution to stratospheric
chlorine levels by the two species. Particularly the region at about $380\,K$ potential temperature is dominated by young air
masses from southern and eastern Asia thereby strongly increasing the chlorine loading from VSLS in this layer in NH late
summer. Our results emphasize that further increases in Asian $CHCl_3$ emissions will inevitably lead to similarly clear signatures
of enhanced $CHCl_3$ mixing ratios in the Ex-LS as we already observe for $CH_2Cl_2$.

A faster pathway from the (sub-)tropical boundary layer into the NH Ex-LS with transport times of $1-5$ weeks was derived
from particularly low $CH_2Cl_2$ mixing ratios observed in the UTLS region. The $CH_2Cl_2$-poor air mainly originated from Central
America as well as from the the tropical Atlantic Ocean and Northern Africa (central and western ITCZ) and was uplifted
above Central America during the course of September. Ground-based AGAGE measurements (Prinn et al., 2018) within that
region show minimum background mixing ratios of both $CH_2Cl_2$ and $CHCl_3$ in September. This seasonal minimum is clearly
reflected in our UTLS measurements and allows these air masses to be distinguished from the strongly enhanced mixing ratios
transported via the ASMA.

The transport pathway derived from $CH_2Cl_2$-poor air follows a general pattern: Air masses are convectively uplifted into the
TTL above Central America to about $360-370\,K$ potential temperature (vertical blue arrow in Figure 15). The convection is
induced by the general updraft in the ITCZ region, by the North American monsoon, and by hurricanes. We could directly link
measurements of $CH_2Cl_2$-poor air to the uplift by the category 5 hurricane Maria (Pasch et al., 2019). After the convection, the
air masses were horizontally transported to higher latitudes and drawn towards an anticyclonic structure above North America.
Resolved by back-trajectories, the anticyclone above North America was much smaller than the ASMA and was located mostly





at $35°$ N and $80°$ W above Florida, likely being a remnant of the NAMA which usually declines in late September (e.g., Vera
et al., 2006). Other than observed for the ASMA, the circulating air parcels above North America did not significantly increase
their potential temperature. Further, the air masses broke out of the anticyclone northeastward forming a streamer which carried
a local high TP into higher latitudes. Eventually, these air masses mixed into the LS by Rossby wave breaking and influenced
the chemical composition of the NH Ex-LS $10 - 20$ K below the air masses dominated by transport via the ASMA.

    Our study shows that air masses lifted by convection in the tropical region of Central America do not need to directly cross
the TP or to slowly enter the tropical pipe to be transported into the stratosphere. In the TTL, fast horizontal transport north-
ward on high potential temperature levels provides an efficient and fast pathway for air lifted up in the tropics above Central
America to quasi-isentropically enter the Ex-LS during NH late summer. Air transported along this pathway was observed to be
mostly $CH_2Cl_2$-poor and $CHCl_3$-poor air. However, transport along this pathway may cause other ozone depleting short-lived
substances with stronger sources in the region of the central and western ITCZ (such as tropical maritime and coastal sources,
e.g., $CH_2Br_2$ and $CHBr_3$, Hepach et al., 2015) to be significantly enhanced in the middle and lower part of the LMS (e.g.,
Rotermund et al., 2021).

    Particularly the use of in situ $CH_2Cl_2$ measurements as a very short-lived tracer has clearly revealed the differences be-
tween the two main transport pathways into the NH Ex-LS described in this study. In addition, we have deduced a higher
$CHCl_3$:$CH_2Cl_2$ emission ratio in the central and western ITCZ region compared to southern and eastern Asia. The difference
might be due to additional biogenic $CHCl_3$ sources in the ocean-rich central and western ITCZ region, while the emissions in
southern and eastern Asia are most likely dominated by anthropogenic continental sources. However, more UTLS observations
of $CH_2Cl_2$ and $CHCl_3$ in different seasons as well as more ground-based long-term observations of the two species in Asia are
needed to complete the understanding of the seasonal and inter-annual variability of the transport pathways identified in our
study. Figure 15 shows a schematic drawing of the two reported main transport pathways into the NH Ex-LS in late summer.
The scheme summarizes the main findings of this paper.





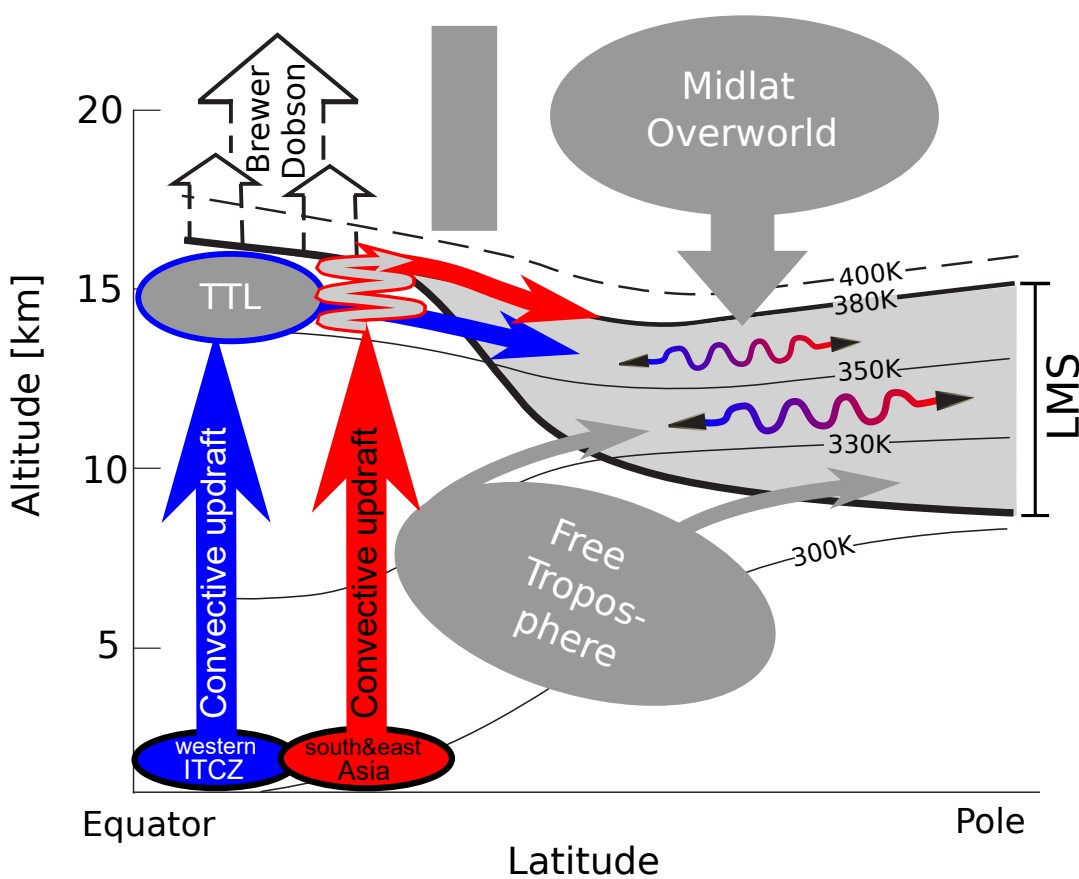

**Figure 15.** Schematic meridional view of the two major transport pathways for $CH_2Cl_2$-rich and $CHCl_3$-rich air (red) and $CH_2Cl_2$-poor and mainly $CHCl_3$-poor air (blue) from the source region into the NH LMS. The pathway from source regions located mostly in the western part of the ITCZ starts with convection into the tropical tropopause layer (TTL) above Central America by general updraft in the ITCZ, by the North American monsoon, and by hurricanes as shown for hurricane Maria in Section 3.1.3. Quasi-isentropic transport to the north and northeast eventually transports the air into the LMS at $\Theta \approx 360\,K$. Air masses from southern and eastern Asia are uplifted by the Asian summer monsoon (ASM) to $\Theta \approx 360\,K$ with subsequent slow upwelling within the monsoon anticyclone to $\Theta \approx 380\,K$. These air masses break out of the anticyclone to follow the subtropical jet stream eastwards before isentropically entering the Ex-LS above the eastern Pacific or western Atlantic Ocean.

*Data availability.* The following measured and simulated WISE data used in this paper is available at the HALO data depository (https://halo-db.pa.op.dlr.de/mission/96): HAGAR-V $CH_2Cl_2$ and $CHCl_3$ (Dataset #5917–#5926); UMAQS $N_2O$ (Dataset #5979–#5988); BAHAMAS aircraft data (Dataset #5618–#5627); CLaMS equivalent latitude and tropopause (Dataset #5455–#5464). Signing a data protocol is mandatory to access the data from the HALO data depository. The CLaMS surface origin tracers and the back-trajectory calculations are available upon request (b.vogel@fz-juelich.de).






## Appendix A: Absolute fractions of surface origin tracers

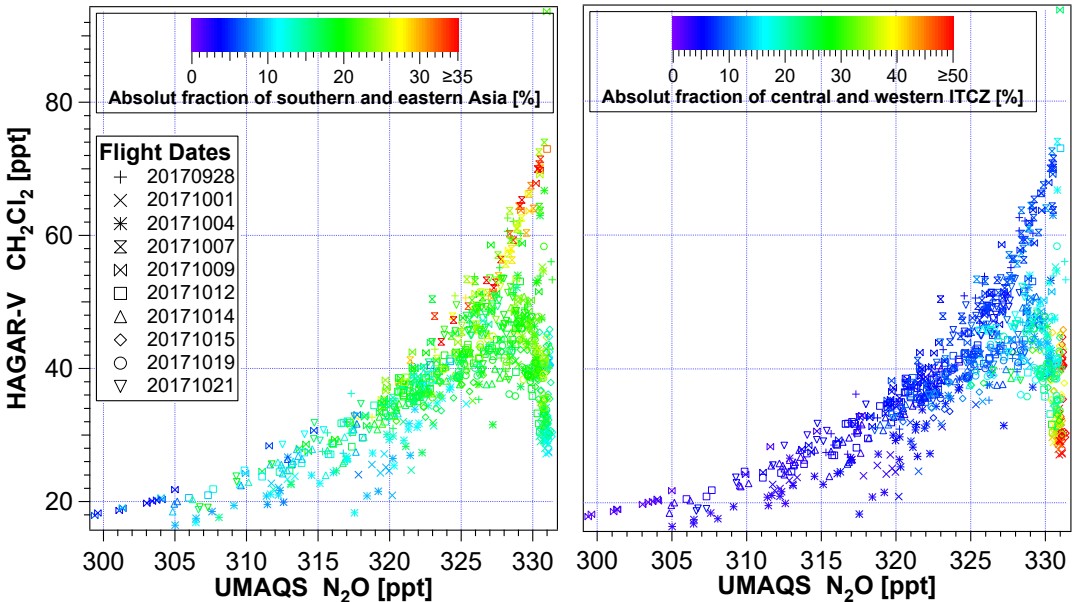

**Figure A1.** $CH_2Cl_2$-$N_2O$ correlation color coded with different surface origin tracers. The shown tracer fractions are relative to the other surface origin tracers including the fraction of air that has already been above the model boundary layer at the initialization of the model simulation on 1 May, 2017.



## Appendix B: CHCl₃-N₂O correlation vs AGAGE measurements

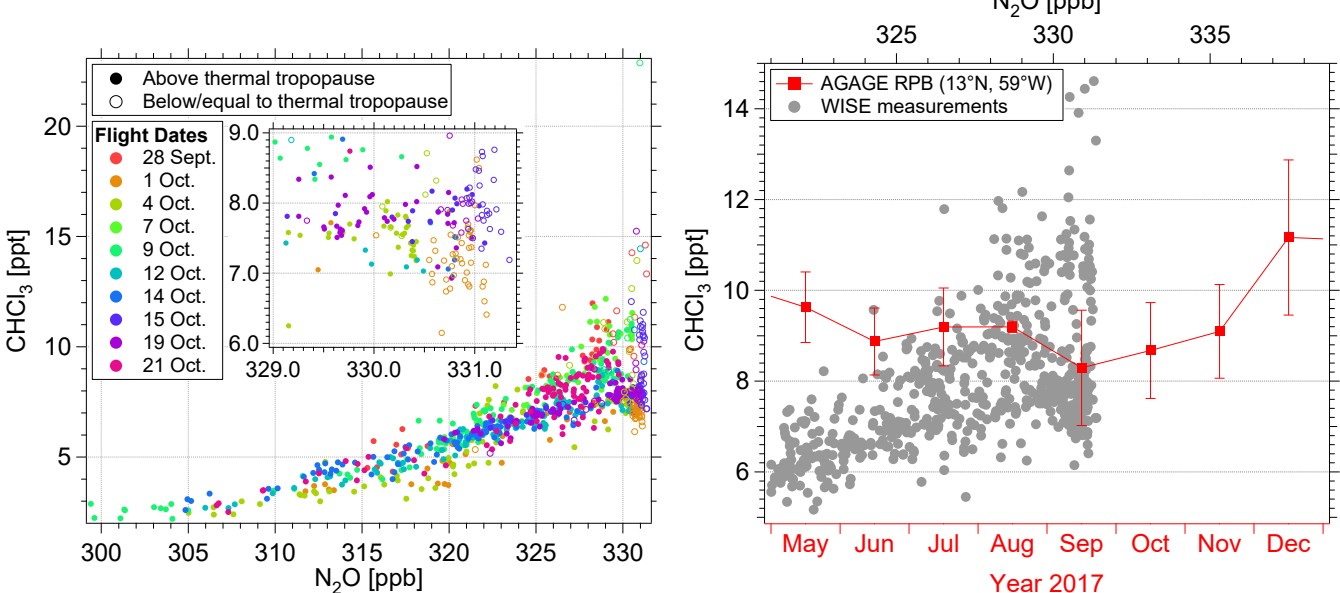

**Figure B1.** CHCl₃-N₂O correlation color coded by flight date (left) and monthly averaged ground-based CHCl₃ measurements of AGAGE at Ragged Point, Barbados (right; Prinn et al., 2018) overlayed by a detailed plot of the CHCl₃-N₂O correlation. The right plot showing the AGAGE CHCl₃ time series is similar to Figure 4 where a detailed description is given.

a



*Author contributions.* VL, JW, AR, and CMV carried out the measurements with the HAGAR-V instrument; PH and VB provided the UMAQS $N_2O$ data; the CLaMS simulations and the back-trajectory calculations were performed by BV. VL processed the HAGAR-V data
and analyzed the measurements and simulations; CMV helped with interpreting the results. The results of the study were discussed by all the co-authors, with particular contributions by BV, RM and CMV. The paper was written by VL, with supporting comments from BV, RM, CMV, JW, and PH.

*Competing interests.* RM is an editor of ACP; otherwise the authors declare that they have no conflict of interest.

*Acknowledgements.* The authors would like to thank Simon O'Doherty and Dickon Young for the use of the AGAGE network ground-based $CH_2Cl_2$ and $CHCl_3$ measurements at Ragged Point obtained from http://www.agage.mit.edu/ (last access: 23 June 2021). Operation of the AGAGE laboratory at Ragged Point is supported by the National Aeronautics and Space Administration (NASA, grant NNX16AC98G to MIT) and the National Oceanic and Atmospheric Administration (NOAA, contract 1305M319CNRMJ0028 to the University of Bristol). We acknowledge the use of ERA-Interim reanalysis data obtained from the European Centre for Medium-Range Weather Forecasts (ECMWF).
We also acknowledge the use of CLaMS calculations regarding the location of the tropopause and the equivalent latitude provided by Jens-Uwe Grooß (Forschungszentrum Jülich). We gratefully like to acknowledge Tanja Schuck (University of Frankfurt) for the calibration of the HAGAR-V laboratory standards. We warmly thank the whole WISE community, in particular the scientific coordinators of the campaign, the pilots, the management and ground support team of the Deutsches Zentrum für Luft- und Raumfahrt Flugexperimente (DLR-FX), and the flight planning team for making this campaign a success. We gratefully acknowledge the extended HAGAR-V team supporting the mea-
surements of the WISE mission particularly Emil Gerhardt (software and field support), Peter Knieling (electronics), and Axel Frohschauer (mechanics). We acknowledge the use of water vapor data from the FISH instrument provided by Christian Rolf and Nicole Spelten (both Forschungszentrum Jülich) as well as the use of avionic data provided by the BAHAMAS team (Mess- und Sensortechnik, DLR-FX). We also want to thank Andreas Petzold (Forschungszentrum Jülich) for helpful remarks on the $CHCl_3$-$CH_2Cl_2$ correlation. We gratefully ac-knowledge the computing time for the CLaMS simulations granted on the supercomputer JURECA at Jülich Supercomputing Centre (JSC)
under the VSR project ID JICG11. Our activities were mainly funded by the German Science Foundation (Deutsche Forschungsgemein-schaft, DFG) Priority Program SPP 1294: VO1530/5-1 (HAGAR-V), VO1276/5-1 (CLaMS), HO4225/7-1 and HO4225/8-1 (UMAQS). VL was partly funded by a HITEC (Helmholtz Interdisciplinary Doctoral Training in Energy and Climate Research) fellowship by the Forschungszentrum Jülich.



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
