# Peer review of "In situ observations of CH2Cl2 and CHCl3 show efficient transport pathways for very short-lived species into the lower stratosphere via the Asian and the North American summer monsoon"

_Atmospheric Chemistry and Physics, 2021_

## Referee Comment (RC1)

Review of "In situ observations of CH2Cl2 and CHCl3 show efficient transport pathways for very short-lived species into the lower stratosphere via the Asian and North American summer monsoons"

This study presents observations of very short-lived substances (VSLS), $CH_2Cl_2$ and $CHCl_3$, collected during the WISE aircraft campaign over the Northern Hemisphere Atlantic Ocean and Western Europe during fall 2017. These observations are analyzed using CLaMS simulations to identify source regions and regions with rapid convection that transport the VSLS to the extratropical lower most stratosphere (LMS) and the upper troposphere. The Asian summer monsoon anticyclone is identified as a significant pathway for transporting chlorinated VSLS into the LMS, including samples with elevated anthropogenic sources of $CH_2Cl_2$. The role of the North American monsoon in the rapid transport of VSLS to the extratropical upper troposphere/lower stratosphere is explored with hurricane Maria as a case study. Further refinement of the text is needed to clearly communicate the importance of this pathway to stratospheric composition.

Overall, the paper is well-written and thoroughly researched. However, at times the language is imprecise, specifically with regards to which observations are representative of stratospheric air and which transport time is being discussed.

**Specific comments:**
1. This distinction between tropospheric and stratospheric air is clear in the figures but does get lost in some of the text. For stratospherically relevant numbers reported, these should be calculated with only observations collected above the thermal tropopause. If that is already being done, it is not clearly communicated in the main text.

   For example, on line 19 in the abstract and in the main text it is not clear if the reported "1-5 weeks into the Ex-LS" was calculated only with observations collected above the thermal tropopause.

2. It is misleading to place N2O and month on the same axis in Figure 4. Measurements of CH2Cl2 at N2O mixing ratios of 325 ppb should not necessarily be compared to surface observations in June; some of the low mixing ratios of CH2Cl2 shown here are likely due to photochemical processing in the stratosphere, not seasonal surface trends.

   It should also be noted that biases between NOAA and Advanced Global Atmospheric Gas Experiment (AGAGE) records of CH2Cl2, particularly in the northern tropic station shown in Figure 4, have been identified, suggesting either calibration errors or longitudinal gradients in mixing ratios of CH2Cl2 (Engel et al., 2018).

   Additionally, the CH2Cl2-poor air is connected to the OH-driven seasonality based on ground-based observations at one AGAGE station without discussing other explanations. However, since CH2Cl2 is not well mixed in the troposphere, a contributing factor could be the uplift of air with less anthropogenic influence than measured at the ground-based station. This option is alluded to on line 402, but it is not clearly introduced as a separate mechanism for low CH2Cl2 mixing ratios.

3. Variations of "first study" statements appear throughout the paper. I would caution against that phrasing since it seems unnecessary for the importance of the paper and can be untrue if one of the qualifications in these statements are removed. From a literature search, at least Adcock et al. (2021) provides back-trajectory analysis and identifies source regions of measurements of CH2Cl2 in the UTLS. Additionally, Rotermund et al. (2021), which is properly referenced in the discussion section, should also be acknowledged in the introduction as a study that employed similar methods to investigate source regions of brominated VSLS during the WISE campaign.

4. It would be helpful to either place the results of the present study in the context of similar studies conducted with observations of other VSLS (e.g., Aschmann et al., 2009; Ashfold et al., 2012; Levine et al., 2007; Liang et al., 2014), or explain in the introduction why the results of those studies have limited applications for studying the transport of Cl-VSLS.

5. "…two most efficient and fast transport pathways from (sub-)tropical source regions into the extratropical lower stratosphere (Ex-LS)…" (Line 7)

   This study has identified the ASMA as an efficient pathway into the extratropical stratosphere and the North American monsoon as a fast transport pathway into the Northern Hemisphere upper troposphere/lower stratosphere. However, from the observations presented in this paper, it is not clear how efficiently Cl-VSLS lofted into the upper troposphere by hurricanes are mixed into the stratosphere. This sentence could be simplified to: "…two transport pathways from (sub-)tropical source regions into the extratropical lower stratosphere and upper troposphere…". Further detail of the difference between the two pathways is provided later in the abstract.

6. Line 132: Please state why earlier flights are not included in this study. Also, altitude and pressure ranges are not given for the measurements. Were any filtering criteria used to limit observations to the UTLS?

7. Section 3.1.3 would read more clearly if the "Analysis of transport pathways" is discussed before "Case study: convective uplift by hurricane Maria". As written, what is meant by fast transport time

8. Line 370: Please explain why the comparison to August and September ground-based observations is justified based on the transport time since maximum $\Delta\Theta_{18h}$ and not transport time since the boundary layer?

   Similarly, is the uplift described on line 373 recent or fast uplift of CH2Cl2-poor air (since max $\Delta\Theta_{18h}$ or since the boundary layer, respectively)?

9. In the discussion section, line 504, the authors state "…clearly benefits the use of CH2Cl2 observations to derive details about the different transport mechanisms and pathways…". It is an interesting and novel way to frame the paper, and if the authors would like to highlight that aspect, it should be introduced at the end of Section 1.

However, in the discussion please note that CH2Cl2 and CH2Br2 have similar atmospheric lifetimes, and regionally varying surface mixing ratios, not just the OH seasonal cycle, can influence the low CH2Cl2 observed in the UTLS. The use of the CHCl3:CH2Cl2 ratio as a diagnostic tracer could also be better highlighted.

10. The discussion of the flight on 1 October (lines 542 – 556) could be moved to the hurricane case study section. Having this text at the end of Section 4 somewhat distracts from the big picture highlights given. Please note on line 553 and in similar text that while NAMA was observed to be a fast transport pathway to the UTLS, a low fraction of stratospheric air originates from central and western ITCZ (Figure 6).

**Technical comments:**
- "Regionally differing … CHCl3 sources" (Lines 23 – 26): As written this sentence is confusing. Please clarify that a larger fraction of CHCl3 than CH2Cl2 is emitted by natural sources, and, consequently, a lower CHCl3:Cl2Cl2 is found in air parcels that originate from regions with significant anthropogenic influences than regions with weaker anthropogenic emissions.

- "Due to the sparseness of Cl-VSLS measurements in the stratosphere" (Line 36): This statement is true, but please reference some past studies that measured Cl-VSLS in the stratosphere, such as (Laube et al., 2008; Schauffler et al., 1993, 2003) in addition to some of the references already present in the paper.

- Paragraph beginning on Line 54: It's important for your analysis of the CHCl3:CH2Cl2 ratio that the reader understands CH2Cl2 is largely emitted by anthropogenic sources, while a larger fraction of CHCl3 is from natural sources. This should be directly stated for the reader here. Overall, the detail given for the different emissions sources for the two VSLS could be more concise and focused so that this point does not get missed.

- Line 77: It may be helpful to state the atmospheric lifetimes of CH2Cl2 and CHCl3 at the beginning of this paragraph so that they are not lost in the discussion.

- Line 104: If there are modeling studies that explored source regions of Cl-VSLS in addition to Claxton et al. (2019), they should be referenced here.

- Lines 255 - 260: Some of the wording in this paragraph is hard to follow. A useful point that is missing here is N2O, due to its long tropospheric lifetime is well mixed in the troposphere, while CH2Cl2 is not.

- "measurements of AGAGE" (Line 262): Here and elsewhere in the paper, please correct to read "ground-based measurements of CH2Cl2 from the AGAGE network"

- Line 299 – 304: The description of the different criteria is hard to follow. Please carefully reword. Also, readers may have to refer back to criteria (1) and (2), it could be helpful to format them as separate bullets.

- Figure 7: Could the transport time to the boundary layer be shown in a similar figure? The lines given in Figure 10 are hard to distinguish.

- Line 354: Locations of and transport times since maximum diabatic ascent rate

- Line 389: Please state how many of these air masses were collected above the thermal tropopause.

- "positively correlate with transport time" (Line 391): What is shown in Figure 8 is the time since max convection, not the overall transport time. Please clarify the discussion in this paragraph.

- Lines 411 and 434: Please clearly state that you are only performing back trajectories for observations collected above the thermal tropopause, otherwise here and elsewhere should read "into the UTLS"

- Line 447: State for the readers that a broader range of surface mixing ratios of CH2Cl2 (~10 – 70 ppt) are reported from surface stations than for CHCl3 (~5 – 15 ppt) (https://agage.mit.edu/data/agage-data).

- Line 476: Remind the reader here or earlier in the section that a larger fraction of CHCl3 emissions than CH2Cl2 are from natural sources and the two compounds have similar atmospheric lifetimes.

- Line 408: The ASMA pathway was also identified by Adcock et al. (2021) for CH2Cl2.

**References**

Adcock, K. E., Fraser, P. J., Hall, B. D., Langenfelds, R. L., Lee, G., Montzka, S. A., et al. (2021). Aircraft-Based Observations of Ozone-Depleting Substances in the Upper Troposphere and Lower Stratosphere in and Above the Asian Summer Monsoon. *Journal of Geophysical Research: Atmospheres*, *126*(1). https://doi.org/10.1029/2020JD033137

Aschmann, J., Sinnhuber, B. M., Atlas, E. L., & Schauffler, S. M. (2009). Modeling the transport of very short-lived substances into the tropical upper troposphere and lower stratosphere. *Atmospheric Chemistry and Physics*, *9*(23), 9237–9247. https://doi.org/10.5194/acp-9-9237-2009

Ashfold, M. J., Harris, N. R. P., Atlas, E. L., Manning, A. J., & Pyle, J. A. (2012). Transport of short-lived species into the Tropical Tropopause Layer. *Atmospheric Chemistry and Physics*, *12*(14), 6309–6322. https://doi.org/10.5194/acp-12-6309-2012

Engel, A., Rigby, M., Burkholder, J. B., Fernandez, R. P., Froidevaux, L., Hall, B. D., et al. (2018). Update on Ozone-Depleting Substances (ODSs) and Other Gases of Interest to the Montreal Protocol, Chapter 1. In *Scientific Assessment of Ozone Depletion: 2018, Global Ozone Research and Monitoring Project – Report No. 58*. Geneva, Switzerland: World Meteorological Organization.

Laube, J. C., Engel, A., Bönisch, H., Möbius, T., Worton, D. R., Sturges, W. T., et al. (2008). Contribution of very short-lived organic substances to stratospheric chlorine and bromine in the tropics – a case study. *Atmospheric Chemistry and Physics*, *8*(23), 7325–7334. https://doi.org/10.5194/acp-8-7325-2008

Levine, J. G., Braesicke, P., Harris, N. R. P., Savage, N. H., & Pyle, J. A. (2007). Pathways and timescales for troposphere-to-stratosphere transport via the tropical tropopause layer and their relevance for very short lived substances. *Journal of Geophysical Research Atmospheres*, *112*(4), 1–15. https://doi.org/10.1029/2005JD006940

Liang, Q., Atlas, E., Blake, D., Dorf, M., Pfeilsticker, K., & Schauffler, S. (2014). Convective transport of very short lived bromocarbons to the stratosphere. *Atmospheric Chemistry and Physics*, *14*(11), 5781–5792. https://doi.org/10.5194/acp-14-5781-2014

Rotermund, M. K., Bense, V., Chipperfield, M. P., Engel, A., Grooß, J.-U., Hoor, P., et al. (2021). Organic and inorganic bromine measurements around the extratropical tropopause and lowermost stratosphere: insights into the transport pathways and total bromine. *Atmospheric Chemistry and Physics*, *21*(20), 15375–15407. https://doi.org/10.5194/acp-21-15375-2021

Schauffler, S. M., Heidt, L. E., Gilpin, T. M., Solomon, S., Lueb, R. A., & Atlas, E. L. (1993). Measurements of Halogenated Organic Compounds Near the Tropical Tropopause. *Geophys. Res. Lett.*, *20*(22), 2567–2570.

Schauffler, S. M., Atlas, E. L., Donnelly, S. G., Andrews, A., Montzka, S. A., Elkins, J. W., et al. (2003). Chlorine budget and partitioning during the Stratospheric Aerosol and Gas Experiment (SAGE) III Ozone Loss and Validation Experiment (SOLVE). *Journal of Geophysical Research*, *108*(D5), 4173. https://doi.org/10.1029/2001JD002040

---

## Author Comment (AC1)

Reply to the comments of Referee #1 on the paper

*In situ observations of $CH_2Cl_2$ and $CHCl_3$ show efficient transport pathways for very short-lived species into the lower stratosphere via the Asian and North American summer monsoons*

by V. Lauther et al.

We thank Referee #1 for the thorough reading of and the very helpful remarks on our manuscript. The comments have been considered carefully while revising the draft and resulting modifications are addressed point-py-point in the following (Referee's comments are cited in bold face):

**Specific comments:**

**1. This distinction between tropospheric and stratospheric air is clear in the figures but does get lost in some of the text. For stratospherically relevant numbers reported, these should be calculated with only observations collected above the thermal tropopause. If that is already being done, it is not clearly communicated in the main text. For example, on line 19 in the abstract and in the main text it is not clear if the reported "1-5 weeks into the Ex-LS" was calculated only with observations collected above the thermal tropopause.**

This is a very good remark! We agree and have now specified the given transport times at the respective locations in the paper. In addition, we have added an overview table with median transport times derived from back-trajectories of different lengths (from the model boundary layer and from the location of maximum convection) for all observations in the UTLS and either only for those in the UT or only for those in the LS. The used data set in the new table (Table 2) is also filtered to include only measurements of $CH_2Cl_2$-poor air linked to the NAM as well as measurements of $CH_2Cl_2$-rich air linked to the ASM. We consistently use median transport times now and provide ranges of transport times derived from the 25. and 75. percentile of the respective data set. For details, we refer to the track change version of the manuscript.

In addition, we noticed that the separation between measurements in the UT and the LS shown in the figures was based on ECMWF's ERA5 reanalysis data (Hersbach et al., 2020). There are but a few differences regarding the discrimination between UT and LS of individual data points when using ERA5 instead of ERA-Interim data. However, to be consistent, we changed the displayed separation between UT and LS in all relevant figures (3, 12 (now 13), 13 (now 14), B1 (now C1); no changes were necessary in Fig. 9 (now 10)) to the calculations based on ERA-Interim.

**2. It is misleading to place N2O and month on the same axis in Figure 4. Measurements of CH2Cl2 at N2O mixing ratios of 325 ppb should not necessarily be compared to surface observations in June; some of the low mixing ratios of CH2Cl2 shown here are likely due to photochemical processing in the stratosphere, not seasonal surface trends.**

This is true and we thought about that while carefully choosing the wording of this section (i.e. "Although this simplified view ignores the impact of mixing processes and chemical reduction of $CH_2Cl_2$ it qualitatively explains the lower branch of the correlation curve for air parcels younger than a few months."). However, we think Figure 4 is an interesting qualitative comparison between two different data sets providing the reader a good first impression of a plausible explanation for one part of the rather complex $CH_2Cl_2$-$N_2O$ relationship. Nevertheless, we agree that Figure 4 should be interpreted with care and not be used out of context. In the caption of Figure 4 we thus refer to the similarities between the $CH_2Cl_2$-$N_2O$ relationship and the ground-based $CH_2Cl_2$ observations from the AGAGE network focusing only on the lower branch of the correlation — implying a comparison of the correlation with AGAGE data from about July to September. This range for a plausible comparison is also stated in the main text ("The observed decrease of low $CH_2Cl_2$ mixing ratios for increasing $N_2O$ mixing ratios (from older to younger air) agrees well with the decreasing tropical monthly averaged $CH_2Cl_2$ mixing ratios from about July to September

45  2017, as observed by AGAGE."). However, we agree that the figure might be misleading to some readers and therefore we have now included vertical lines highlighting the comparison between July and September to clarify the plausible range of the comparison. In addition, we have specified the respective text in the Figure 4 caption as follows:

"The gradient of the $CH_2Cl_2$-$N_2O$ relationship's lower branch (low $CH_2Cl_2$ mixing ratios between vertical lines) qualitatively
50  fits the temporal variation of the ground-based $CH_2Cl_2$ measurements."

**It should also be noted that biases between NOAA and Advanced Global Atmospheric Gas Experiment (AGAGE) records of CH2Cl2, particularly in the northern tropic station shown in Figure 4, have been identified, suggesting either calibration errors or longitudinal gradients in mixing ratios of CH2Cl2 (Engel et al., 2018).**
55

**Additionally, the CH2Cl2-poor air is connected to the OH-driven seasonality based on ground-based observations at one AGAGE station without discussing other explanations. However, since CH2Cl2 is not well mixed in the troposphere, a contributing factor could be the uplift of air with less anthropogenic influence than measured at the ground-based station. This option is alluded to on line 402, but it is not clearly introduced as a separate mechanism for low CH2Cl2**
60  **mixing ratios**

These are good points and we have included them in the text of Section 3.1 as follows:

"It is thus very likely that the lower branch of the $CH_2Cl_2$-$N_2O$ relationship is caused by the tropical Atlantic
65  $CH_2Cl_2$ surface seasonality. However, the low $CH_2Cl_2$ mixing ratios observed during WISE could also be impacted by air uplifted from regions less influenced by $CH_2Cl_2$ sources than the observations from the AGAGE network at Barbados. Ground-based observations of $CH_2Cl_2$ surface mixing ratios from the AGAGE and the NOAA network show strong regional differences particularly in the NH tropics. However, it is unclear if these differences are caused by calibration biases or are of a natural origin (Engel et al., 2018)."
70

**3. Variations of "first study" statements appear throughout the paper. I would caution against that phrasing since it seems unnecessary for the importance of the paper and can be untrue if one of the qualifications in these statements are removed. From a literature search, at least Adcock et al. (2021) provides back-trajectory analysis and identifies source regions of measurements of CH2Cl2 in the UTLS. Additionally, Rotermund et al. (2021), which is properly referenced**
75  **in the discussion section, should also be acknowledged in the introduction as a study that employed similar methods to investigate source regions of brominated VSLS during the WISE campaign.**

We agree and have removed the respective statements of "first study" from the paper. Also Adcock et al. (2021) is discussed in the paper. For details, we refer to the track change version of the manuscript.
80

Rotermund et al. (2021) is now acknowledged in the introduction of the paper in the following way:

A study by Rotermund et al. (2021) employed similar methods to identify source regions and the impact on the Ex-LS of Br-VSLS using measurements from the same aircraft campaign as the measurements used in the present paper and is compared
85  to our results in Section 4.

**4. It would be helpful to either place the results of the present study in the context of similar studies conducted with observations of other VSLS (e.g., Aschmann et al., 2009; Ashfold et al., 2012; Levine et al., 2007; Liang et al., 2014), or explain in the introduction why the results of those studies have limited applications for studying the transport of**
90  **Cl-VSLS.**

The results of similar transport studies focused on other VSLS (usually Br-VSLS with mainly natural, oceanic sources) cannot necessarily be used to describe the transport of Cl-VSLS (mainly anthropogenic, land-based sources) because of their different

source distribution in the troposphere. Therefore it is necessary to describe transport pathways into the stratosphere for VSLS (particularly for those with strongly varying mixing ratios in the troposphere) specifically for the respective VSLS or group of VSLS with similar characteristics (e.g., photochemical lifetime and source distribution). To point this out, we have now added the following explanations to the introduction of the paper:

"Observational evidence for Cl-VSLS being transported into the stratosphere is extremely rare  (e.g., Schauffler et al., 1993; Woodbridge et al., 1995; Schauffler et al., 2003; Laube et al., 2008; Park et al., 2010; Adcock et al., 2021). Transport pathways into the stratosphere for VSLS have been derived from observations of bominated VSLS (Br-VSLS; e.g., Sturges et al., 2000; Ashfold et al., 2012; Wales et al., 2018; Filus et al., 2020; Keber et al., 2020; Rotermund et al., 2021) or modeled specifically for Br-VSLS (e.g., Levine et al., 2007; Aschmann et al., 2009; Ashfold et al., 2012; Liang et al., 2014) which have mainly natural emission sources (Engel et al., 2018). However, the only Br-VSLS with a photochemical lifetime comparable to those of $CH_2Cl_2$ and $CHCl_3$ is $CH_2Br_2$ (150 days; WMO, 2018) which is mostly emitted by the oceans and, consequently, is differently distributed in the troposphere than the mainly anthropogenically (land-based) emitted $CH_2Cl_2$ and most Cl-VSLS (e.g., Engel et al., 2018). Thus, transport studies of Br-VSLS focus on transport into the stratosphere from likely different source regions than those of Cl-VSLS and their results might not necessarily be directly applicable to the transport into the stratosphere of $CH_2Cl_2$ and $CHCl_3$. In addition, in order to specifically study transport into the stratosphere via the ASM it is beneficial to observe VSLS with their strongest sources being located in the core region of the ASM. This is the case for $CH_2Cl_2$ while most Asian Br-VSLS sources are located only in adjacent regions of the ASM."

**5. "…two most efficient and fast transport pathways from (sub-)tropical source regions into the extratropical lower stratosphere (Ex-LS)…" (Line 7) This study has identified the ASMA as an efficient pathway into the extratropical stratosphere and the North American monsoon as a fast transport pathway into the Northern Hemisphere upper troposphere/lower stratosphere. However, from the observations presented in this paper, it is not clear how efficiently Cl-VSLS lofted into the upper troposphere by hurricanes are mixed into the stratosphere. This sentence could be simplified to: "…two transport pathways from (sub-)tropical source regions into the extratropical lower stratosphere and upper troposphere…". Further detail of the difference between the two pathways is provided later in the abstract.**

This is true. We changed the sentence as suggested.

**6. Line 132: Please state why earlier flights are not included in this study. Also, altitude and pressure ranges are not given for the measurements. Were any filtering criteria used to limit observations to the UTLS?**

We have reconsidered the wording of Line 132 and added the full ranges of the used data set regarding potential temperature, altitude, and pressure. No explicit filtering of the data set was used to confine the presented observations to the UTLS region. However, due to a malfunctioning water trap of the HAGAR-V instrument, we could perform $CH_2Cl_2$ and $CHCl_3$ measurements only at ambient water vapor levels below about $100\,ppm$ (measuring mainly at around $6\,ppm$), thus "naturally" confining the observations to the upper part of the troposphere and above. This is briefly explained in Section 2.2.1 to which we now refer in the introduction to the used data set (Sect. 2.1). We also added to Section 2.2.1 why we were able to measure only during the last ten flights of the WISE campaign. The respective parts of the manuscript are modified as follows (sections 2.1 and 2.2.1, respectively):

"In this study we present UTLS measurements between a potential temperature of $315\,K$ and $404\,K$ (i.e., $7.4 - 14.5\,km$ altitude; $388 - 130\,hPa$ pressure) of the last ten WISE flights, i.e., from 28 September to 21 October 2017 (Figure 1). Due to technical issues of the instrument, $CH_2Cl_2$ and $CHCl_3$ measurements below the given range and during earlier flights of the WISE campaign were not performed (cf. Section 2.2.1)."

"However, during WISE the dehydration system of HAGAR-V was malfunctioning.  For the last 10 WISE flights that system was bypassed and the MS module measured only at low ambient water vapor levels (mainly

at $H_2O < 100$ ppm; median: 5.6 ppm), i.e., in the UTLS region.  thus yielding measurements during about 90% of a typical  flight's duration (i.e., about 7.6 h per flight). MS measurements of WISE
145 flights before 28 September could not be used for analysis due to the malfunctioning sample dehydration unit."

**7. Section 3.1.3 would read more clearly if the "Analysis of transport pathways" is discussed before "Case study: convective uplift by hurricane Maria". As written, what is meant by fast transport time**

150 We agree and have rearranged the respective paragraphs and figures. Note, that this is not directly marked in the track changes version of the manuscript to keep changes within the text highlighted.

**8. Line 370: Please explain why the comparison to August and September ground-based observations is justified based on the transport time since maximum $\Delta\Theta_{18h}$ and not transport time since the boundary layer? Similarly, is the uplift**
155 **described on line 373 recent or fast uplift of CH2Cl2-poor air (since max $\Delta\Theta_{18h}$ or since the boundary layer, respectively)?**

This is a good point and by providing the different calculated transport times (Table 2; cf. response to specific comment "1.") this should be clear now. We additionally included the information about the transport time since the boundary layer to the
160 sentence (see quote below). For $CH_2Cl_2$-poor air parcels with maximum convection above Central America the difference between median transport times from the location of maximum convection and from the model boundary layer is only about one week. Thus 'recent transport' can also be considered 'fast transport'. We agree that this was not clear before we added Table 2 to the paper. In regard to specific comment "8.", the text of Sect. 3.1.3 - "Locations of maximum diabatic ascent rate and transport times" was modified as follows:

165

"The transport times to the UTLS since the ascent above Central America mainly range between 2 − 5 weeks   (cf. Table 2). The main uplift of $CH_2Cl_2$-poor air above Central America thus falls in the time period of late August and throughout the entire September. With transport times from the BL being only about one week longer (Table 2), this result supports the comparison of $CH_2Cl_2$-poor air with the seasonal minimum $CH_2Cl_2$
170 mixing ratios observed by AGAGE at Barbados"

**9. In the discussion section, line 504, the authors state "... clearly benefits the use of CH2Cl2 observations to derive details about the different transport mechanisms and pathways...". It is an interesting and novel way to frame the paper, and if the authors would like to highlight that aspect, it should be introduced at the end of Section 1. However, in**
175 **the discussion please note that CH2Cl2 and CH2Br2 have similar atmospheric lifetimes, and regionally varying surface mixing ratios, not just the OH seasonal cycle, can influence the low CH2Cl2 observed in the UTLS. The use of the CHCl3:CH2Cl2 ratio as a diagnostic tracer could also be better highlighted.**

These are good points. With our revision of the introduction regarding the specific comment number "4." we also highlighted
180 the beneficial use of a VSL tracer like $CH_2Cl_2$ with its main sources located in the core region of the ASM to analyze transport into the stratosphere via the ASM.

We also changed the wording regarding the advantage of $CH_2Cl_2$ over the Br-VSLS analyzed by Rotermund et al. (2021) due to its very strong Asian emission sources and very low mixing ratios in other regions of strong convection. Thereby we
185 highlight the more general fact that we could derive the two transport pathways so clearly due to the strong contrast of tropospheric $CH_2Cl_2$ mixing ratios in the regions of significant convection. The possible reasons for the low $CH_2Cl_2$ mixing ratios are already discussed in Section 3.1 (see our response to the specific comment nr. "2."). However, we wanted to keep the phrasing "[...] a longer lifetime [...]" when comparing $CH_2Cl_2$ to the Br-VSLS analyzed by Rotermund et al. (2021) without specifically commenting on the comparable atmospheric lifetime of $CH_2Br_2$ (this is already mentioned in the introduction, cf.
190 answer to specific comment nr. "4.") because of two reasons: (1) our statement is correct ($CH_2Cl_2$ and $CH_2Br_2$ have similar

atmospheric lifetimes, but on average that of $CH_2Cl_2$ is up to one month longer; WMO, 2018); (2) we write that not a single but the combination of attributes (lifetime and source distribution) is the reason why $CH_2Cl_2$ is the preferred tracer for the analysis of details on the transport pathways into the Ex-UTLS in NH late summer. The respective part of the discussion is now modified as follows:

195

"Nevertheless, compared to the very short-lived bromine species analyzed by Rotermund et al. (2021), the combination of a longer lifetime, highly significant Asian emission sources, and a strong seasonal cycle and very low mixing ratios in other regions of strong convection clearly benefits the use of $CH_2Cl_2$ observations to derive details about the different transport mechanisms and pathways from the source region into the NH summertime UTLS. In addition, using the $CHCl_3$:$CH_2Cl_2$ ratio

200 to support the analysis of air mass origin is a unique and helpful tool in the analysis of transport pathways."

**10. The discussion of the flight on 1 October (lines 542 – 556) could be moved to the hurricane case study section. Having this text at the end of Section 4 somewhat distracts from the big picture highlights given. Please note on line 553 and in similar text that while NAMA was observed to be a fast transport pathway to the UTLS, a low fraction of stratospheric**

205 **air originates from central and western ITCZ (Figure 6).**

We agree and moved this paragraph and Figure 14 (now Fig. 11) to the hurricane case study section which itself has also been moved further below in the text (cf. specific comment number "7."). Note, that the moving of figures and paragraphs is not directly marked in the track changes version of the manuscript to keep changes within the text highlighted.

210

When discussing the transport pathway into the Ex-LS via the NAMA we now added the information about the relatively low fractions of central and western ITCZ air mass origin in stratospheric air as well as the note that we actually observed only 25 % of the air parcels transported by the NAMA within the Ex-LS. We modified the text at three different locations (Section 3.1.3 - Analysis of transport pathways, Section 4 - Discussion (now Section 3.1.3 - Case study: convective uplift by hurricane

215 Maria), and Section 5 - Conclusion) as follows:

"Another aspect adding to the different transport times is the longer transport pathway from Asia because simulations indicated air masses were always observed to reach reached the location of measurement always from the west. Nevertheless, air parcels observed in the Ex-LS are impacted more strongly by air masses transported via the ASMA than via the NAMA (cf. figures 6

220 and 14)."

"We have thereby shown that tropical surface mixing ratios of VSLS from the region of Central America and the Atlantic Ocean can be efficiently transported into the Ex-LS during the late North American monsoon season. For instance, this is of particular importance for brominated short-lived substances (e.g., $CH_2Br_2$ and $CHBr_3$) that have a high ODP and some of

225 their largest emission sources located in tropical oceans (e.g., Hepach et al., 2015; Rotermund et al., 2021). However, it has to be noted that only 25 % of air parcels transported by this pathway were observed in the Ex-LS and stratospheric air masses showed relatively low fractions of air originating in the region of central and western ITCZ compared to those originating in southern and eastern Asia (cf. Figure 14)."

230 Eventually, these air masses mixed into the LS by Rossby wave breaking and influenced the chemical composition of the NH Ex-LS $10 - 20$ K below the air masses dominated by transport via the ASMA. However, only 25 % of air parcels transported via this pathway were observed above the thermal TP with transport times from the BL to the location of measurement ranging from 5 to 9 weeks.

235 **Technical comments:**

**- "Regionally differing ... CHCl3 sources" (Lines 23 – 26): As written this sentence is confusing. Please clarify that a larger fraction of CHCl3 than CH2Cl2 is emitted by natural sources, and, consequently, a lower CHCl3:Cl2Cl2 is found in air parcels that originate from regions with significant anthropogenic influences than regions with weaker**

240     **anthropogenic emissions.**

We changed the wording of the mentioned sentence in the abstract as follows:

"
245
 CH$_2$Cl$_2$ and CHCl$_3$ have similar atmospheric sinks and lifetimes but the fraction
of biogenic emissions is clearly higher for CHCl$_3$ than for the mainly anthropogenically emitted CH$_2$Cl$_2$, consequently
lower CHCl$_3$:CH$_2$Cl$_2$ ratios are expected in air parcels showing a higher impact of anthropogenic emissions. The observed
CHCl$_3$:CH$_2$Cl$_2$ ratio suggests clearly stronger anthropogenic emissions in the region of southern and eastern Asia compared
250     to those in the region of Central America and the tropical Atlantic."

**- "Due to the sparseness of Cl-VSLS measurements in the stratosphere" (Line 36): This statement is true, but please
reference some past studies that measured Cl-VSLS in the stratosphere, such as (Laube et al., 2008; Schauffler et al.,
1993, 2003) in addition to some of the references already present in the paper.**

255

References to other studies presenting Cl-VSLS measurements in the stratosphere are now given: "Owing to the sparseness
of Cl-VSLS measurements in the stratosphere (e.g., Schauffler et al., 1993, 2003; Laube et al., 2008; Park et al., 2010; Adcock et al., 2021)
the impact of changes in Cl-VSLS surface emissions on their distribution in the stratosphere has yet to be fully characterized
on an observational basis."

260

**- Paragraph beginning on Line 54: It's important for your analysis of the CHCl3:CH2Cl2 ratio that the reader under-
stands CH2Cl2 is largely emitted by anthropogenic sources, while a larger fraction of CHCl3 is from natural sources.
This should be directly stated for the reader here. Overall, the detail given for the different emissions sources for the
two VSLS could be more concise and focused so that this point does not get missed.**

265

We agree and the phrase " Nevertheless, on a global scale CHCl$_3$ has a significant fraction of biogenic emission sources in
contrast to CH$_2$Cl$_2$ which is almost exclusively emitted by anthropogenic sources." was added to the mentioned paragraph to
highlight this difference between the two analyzed species.

270     In addition and in agreement with a comment of Referee #2 we shortened the information given in the introduction to focus
more on things directly relevant to the paper.

**- Line 77: It may be helpful to state the atmospheric lifetimes of CH2Cl2 and CHCl3 at the beginning of this paragraph
so that they are not lost in the discussion.**

275

We agree and the mentioned paragraph is now rearranged accordingly:

"For CH$_2$Cl$_2$ Hossaini et al. (2019) suggest an average tropospheric lifetime of 168 days (about 6 months) and a stratospheric
lifetime of $1-2$ years (outside the poles) was estimated by Hossaini et al. (2017). The main atmospheric sink of both CH$_2$Cl$_2$
280     and CHCl$_3$ is the reaction with hydroxyl radicals (OH) in the troposphere. Both species have similar reaction rates with OH
implying similar photochemical lifetimes for both Cl-VSLS (Hsu and DeMore, 1994). Time series of background mixing ra-
tios of both species are anticorrelated to the seasonal cycle of OH (Cox et al., 2003). In the NH seasonal anthropogenic use
of products releasing CHCl$_3$ to the atmosphere (e.g., landfill and chlorination of water) have been observed to have a small
local impact on the seasonality of CHCl$_3$ (Gentner et al., 2010). In addition, the global distribution of OH shows signifi-
285     cant regional differences (Spivakovsky et al., 2000; Hanisco et al., 2001; Lelieveld et al., 2016). Therefore, also the pho-
tochemical lifetimes of CH$_2$Cl$_2$ and CHCl$_3$ are regionally different.

"

**- Line 104: If there are modeling studies that explored source regions of Cl-VSLS in addition to Claxton et al. (2019), they should be referenced here.**

This paragraph highlights the regional dependency of the Cl-VSLS's ODP which is, to the best of our knowledge, presented only by Claxton et al. (2019).

**- Lines 255 - 260: Some of the wording in this paragraph is hard to follow. A useful point that is missing here is N2O, due to its long tropospheric lifetime is well mixed in the troposphere, while CH2Cl2 is not.**

We considered the comment and added the proposed additional information to the text as follows:

"With a photochemical lifetime of 123 years (Ko et al., 2013) $N_2O$ is well mixed in the troposphere and has a much longer lifetime than $CH_2Cl_2$ which exhibits strongly varying mixing ratios throughout the boundary layer (e.g., Simmonds et al., 2006)."

**- "measurements of AGAGE" (Line 262): Here and elsewhere in the paper, please correct to read "ground-based measurements of CH2Cl2 from the AGAGE network"**

The text is now changed to the proposed phrasing at all convenient places in the manuscript. For details, we refer to the track change version of the manuscript.

**- Line 299 – 304: The description of the different criteria is hard to follow. Please carefully reword. Also, readers may have to refer back to criteria (1) and (2), it could be helpful to format them as separate bullets.**

We agree and reconsidered the wording and structure of the paragraph modifying it as follows:

"Further, to work out differences of air mass origin between $CH_2Cl_2$-rich and -poor air,  the median fraction of a surface origin tracer in $CH_2Cl_2$-rich air parcels is compared to that in $CH_2Cl_2$-poor air parcels. To combine regions of air mass origin with a particularly high relative impact on either $CH_2Cl_2$-rich or -poor air, the ratio of these median surface origin tracer fractions in $CH_2Cl_2$-rich and -poor air is analyzed. Surface origin tracers with particularly high relative median fractions in either $CH_2Cl_2$-rich or -poor air are combined following these two criteria:

(1) considered are only surface origin tracers with median fractions $\geq 1\%$ in $CH_2Cl_2$-rich or -poor air parcels, and

(2) the  ratio of a median surface origin tracer fraction ($CH_2Cl_2$-rich/$CH_2Cl_2$-poor air respectively $CH_2Cl_2$-poor/$CH_2Cl_2$-rich air) must be $> 1.8$.

**- Figure 7: Could the transport time to the boundary layer be shown in a similar figure? The lines given in Figure 10 are hard to distinguish.**

Yes, we added Figure B1 to the appendix and refer to it in Section 3.1.3 as follows:

" The locations of trajectory end points at the model boundary layer color coded with transport time are given in the appendix (Figure B1)."

**- Line 354: Locations of and transport times since maximum diabatic ascent rate**

We have now included also transport times from the boundary layer (Table 2; see reply to specific comment number "1.") and will thus keep the title of the subsection as it is.

**- Line 389: Please state how many of these air masses were collected above the thermal tropopause.**

It were 5 out of 27. The sentence was modified as follows:

"This analysis directly links 27 WISE measurements (five were observed above the thermal TP) to the convection of hurricane Maria with transport times since convection ranging between one week and one month."

**- "positively correlate with transport time" (Line 391): What is shown in Figure 8 is the time since max convection, not the overall transport time. Please clarify the discussion in this paragraph.**

We focus on the transport since the main convection by hurricane Maria because this gives a better impression on the time scale for air parcels lofted by a major hurricane to reach the Ex-UTLS and we analyze impacts on these air parcels induced in the UTLS. However, all trajectories linked to air parcels lifted up by hurricane Maria reach the model boundary layer in September and most do so above the Atlantic Ocean, i.e., at the time of seasonal surface minimum $CH_2Cl_2$ mixing ratios and in a region of weak $CH_2Cl_2$ emission sources. We added the respective information to the text and included a note to the relationship of $CH_2Cl_2$ and transport time since the boundary layer as follows:

"Interestingly, $CH_2Cl_2$ mixing ratios of measurements linked to hurricane Maria positively correlate with transport time since maximum convection ($R_{Pearson} = 0.85$; Figure 10, top right. Note, these $CH_2Cl_2$ mixing ratios also correlate with transport time since the model BL but with a lower $R_{Pearson} = 0.64$ and transport times between 9 and 48 days. However, here we focus on the transport since convection by hurricane Maria to derive impacts on the air parcels induced by processes in the UTLS region). Those air samples related to short transport times contain the lowest $CH_2Cl_2$ mixing ratios at $N_2O > 325$ ppb measured during WISE (Figure 10, top left). According to the back-trajectories, most of the air parcels lifted up by hurricane Maria left the model boundary layer above the tropical Atlantic in September where $CH_2Cl_2$ sources are small (Figure 10, bottom). In addition, in that region the seasonal minimum of $CH_2Cl_2$ mixing ratios is in September (cf. Section 3.1)."

**- Lines 411 and 434: Please clearly state that you are only performing back trajectories for observations collected above the thermal tropopause, otherwise here and elsewhere should read "into the UTLS"**

We analyze the trajectories for observations in the UTLS region and corrected the wording accordingly.

**- Line 447: State for the readers that a broader range of surface mixing ratios of CH2Cl2 (~10 – 70 ppt) are reported from surface stations than for CHCl3 (~5 – 15 ppt) (https://agage.mit.edu/data/agage-data).**

We have considered this statement carefully and concluded that it is not necessarily helpful to the readers. The much clearer split of the $CH_2Cl_2$-$N_2O$ relationship than that of the $CHCl_3$-$N_2O$ relationship is likely not based on the different global absolute surface mixing ratio ranges of the two Cl-VSLS. For air parcels measured during WISE we identified significant regions of air mass origin only around the summertime ITCZ (cf. Figure 2). The surface mixing ratio ranges within these regions between July and September could provide a supporting argument to the discussion. However, using the monthly averaged surface mixing ratios ignores the daily variations at the given location. Including the standard deviation or any other measure of the mean surface mixing ratio scatter range within the relevant regions would actually provide a useful argument for why we can clearly resolve the two branches in the $CH_2Cl_2$-$N_2O$ relationship but not so clearly in the $CHCl_3$-$N_2O$ relationship ($CHCl_3$ has stronger relative variations of mixing ratios within the source regions than $CH_2Cl_2$). However, the sparseness of ground-based Cl-VSLS observations in these regions prevents us from using such an argument. In any case, we think it is not helpful if we provide the global range of monthly averaged mixing ratios of the two Cl-VSLS here.

**- Line 476: Remind the reader here or earlier in the section that a larger fraction of CHCl3 emissions than CH2Cl2 are from natural sources and the two compounds have similar atmospheric lifetimes.**

385

This is a good suggestion and it was included in the text of Section 3.2 as follows:

"The $CHCl_3$-$CH_2Cl_2$ correlation slope thus flattens with increasing entry of air masses originating from southern and eastern Asia.  Knowing that both species have similar sinks and photochemical lifetimes but $CHCl_3$ has a larger fraction of emissions

390 from biogenic sources than $CH_2Cl_2$, this suggests larger $CHCl_3$:$CH_2Cl_2$ emission ratios in the region of the central and western ITCZ region (with presumably mostly biogenic sources) than in southern and eastern Asia (where anthropogenic sources likely dominate)."

**- Line 408: The ASMA pathway was also identified by Adcock et al. (2021) for CH2Cl2.**

395

We partly agree on this point. In this paragraph we highlight the transport pathway from Asia via the ASM including eastward outflow of the ASMA into the Ex-LS derived from observations above western Europe/Atlantic. Adcock et al. (2021) present a transport pathway into the LS based on measurements within the ASMA (StratoClim 2017) and observations of westward ASMA outflow (without back-trajectory analysis) above eastern Europe and the Mediterranean (StratoClim 2016). We do not

400 argue the point that the observations by Adcock et al. (2021) are an important contribution to observational studies regarding Cl-VSLS transport into the LS by the ASMA (and we refer to the paper elsewhere in our manuscript) but quoting their paper at line 508 would be inaccurate to the content of this paragraph.

[revised manuscript text omitted]

---

## Author Comment (AC2)

Response to the comments of Referee #2 on the paper

*In situ observations of $CH_2Cl_2$ and $CHCl_3$ show efficient transport pathways for very short-lived species into the lower stratosphere via the Asian and North American summer monsoons*

by V. Lauther et al.

We thank Referee #2 for the thorough reading of and the very helpful remarks on our manuscript. The comments have been considered carefully while revising the draft and resulting modifications are addressed point-py-point in the following (Referee's comments are cited in bold face):

**General comments:**

**Introduction p2-3: To me there seemed to be a lot of detail here about the overall significance of the CL-VSLS, and about sources. This is important, of course, but it does not all seem to be essential to this paper. The key points, for this paper, are the general geographical characterisation for the source regions of the different species being considered. The reader would be able to focus more effectively on the major points of this paper if some of the detail was removed.**

We agree. A similar remark was made by Reviewer #1 and we reduced the information given in the introduction to focus on the most essential details about Cl-VSLS. For details, we refer to the track change version of the manuscript.

**p10-11: I think that it is a bit confusing to refer to the 'lower branch of the correlation' for the CH2Cl2-N2O relationship. It looks as though there are large number of parcels for which the N2O values are around 330ppb but the CH2Cl2 values are distributed across the range 30-45ppt. I think that in some ways it weakens your case if you call something a correlation when by conventional measures the correlation is rather weak. I am not disputing the fact that there are two 'families' in the plot at high N2O values – but can you find a more neutral term to describe them?**

We agree that our use of the term "correlation" is rather based on lab jargon than on mathematics. We now use the general and more precise wording of a "$CH_2Cl_2$-$N_2O$ relationship" instead of "$CH_2Cl_2$-$N_2O$ correlation" if "correlation" is not specifically meant. We thus refer now to the "lower branch of the $CH_2Cl_2$-$N_2O$ relationship" to describe the values of particularly low $CH_2Cl_2$ mixing ratios at high $N_2O$ mixing ratios (similar for the upper branch). These modifications are made throughout the manuscript and for details we refer to the track change version of it.

**p12: A similar comment applies to this approach of fitting a 'mean correlation curve' and using that as a basis for separating the upper and lower branches. The quadratic fit might be intended to seem quantitative – but there is really no reason to believe that the extrapolated values have any concrete relevance to the concentrations of CH2Cl2 measured in high N2O concentration parcels – the split between the two categories is essentially being made on the basis of the appearance of the plot. It seems very reasonable to make the split – but if the authors (or the readers) felt that it was more justifiable on the basis of a quadratic fit and an extrapolation then I would say that they were confused.**

It is true, the split between the two categories of data was made on the basis of the appearance of the $CH_2Cl_2$-$N_2O$ relationship and we do not claim otherwise. In order to decide on the "geometry" of the split we used the 'mean correlation curve' as a best-guess proxy of the annual mean $CH_2Cl_2$-$N_2O$ relationship. Other methods can be thought of to realize a similar split of the data set. We changed the extrapolation of the shown 'mean correlation curve' in Figure 5 to a dashed line to reduce a possible misunderstanding about the origin of this curve.

**p21: At this stage in the paper you use the term 'convective transport' quite frequently and in association with trajectories/CLaMS. I think that you should be a bit clearer about what transport is included in trajectories/CLaMS – which I believe is simply that in the ERA-I velocity fields, i.e. there is no inclusion of convective transport by parametrisation. I suspect that ERA-I velocities tend to be rapidly upward in regions of large-scale convection – and that serves as some kind of representation of convective transport, but it is unlikely that upward velocities are quantitatively correct. Certainly this sort of interpretation has been made by many authors who have used trajectory-based approaches, including myself, and I would not quarrel with it – but I do think that it needs to be clearly stated. The uncertainty perhaps becomes a bit more serious when considering tropical cyclones. How well is vertical transport in tropical cyclones represented by something like ERA-I? One imagines that maximum vertical velocities are significantly underestimated – but it could be, for example, that vertical transport is distributed over too large a region. You may not be able to resolve this uncertainty, but I think that you should at least say that it exists.**

Many thanks for this helpful comment and we confirm that in CLaMS as used here there is no inclusion of convective transport by parametrisation. We added the following text to Sect. 2.3.

" In CLaMS, the diabatic approach was applied using the diabatic heating rate as the vertical velocity with contributions from radiative heating including the effects of clouds, latent heat release, mixing, and diffusion (for details, see Ploeger et al., 2010). CLaMS employs a hybrid vertical coordinate ($\zeta$) which, in this study, transforms from a strictly isentropic coordinate ($\Theta$) to a pressure-based orography-following coordinate system ($\sigma$ coordinates) below a threshold of approximately 300 hPa(Pommrich et al., 2014). In both three-dimensional simulations as well as in trajectory calculations, the upward transport in CLaMS is driven by ERA-Interim reanalysis data in which changes are implemented to improve deep and mid-level convection compared to previous reanalysis data (Dee et al., 2011). However small-scale rapid uplift in convective cores is not included, therefore small-scale convection is most likely underestimated in CLaMS simulations driven by ERA-Interim. Nevertheless, upward transport in larger convective systems such as tropical cyclones is represented in CLaMS trajectory calculations driven by ERA-Interim (Li et al., 2017, 2020)."

CLaMS trajectory calculations using ECMWF's next-generation reanalysis ERA5 (Hersbach et al., 2020), demonstrate that diabatic trajectory calculations using ERA5 show much faster and stronger vertical transport than ERA-Interim primarily because of ERA5's higher spatial and temporal resolution, which likely resolves convective events more accurately (Li et al., 2020). Nonetheless, Li et al. (2020) demonstrate that the large scale convective uplift as represented in ERA-Interim does represent well the large scale uplift in convective systems.

**Detailed comments:**

**l126: 'last accessed' info not needed in citation?**

The last accessed info is now removed in the main text.

**l164: 'essential' would be better than 'mandatory'. ('Mandatory' means 'required by some rule or regulation'.)**

Good point, we changed "mandatory" to "essential".

**l210-215: I found these sentences confusing. What is the distinction between 'pure CLaMS back trajectories', 'pure CLaMS trajectories' and 'back trajectories'? There seem to be two separate issues here – 'forward vs backward' and 'with CLaMS mixing vs without CLaMS mixing'.**

We agree that this section was written a bit confusing and changed the text as follows:

"To support the interpretation of airborne measurements we use global three-dimensional simulations of the Chemical La-grangian Model of the Stratosphere (CLaMS; McKenna, 2002a,b; Pommrich et al., 2014) as well as  CLaMS back-trajectory calculations.  CLaMS back-trajectory calculations consider only the advective (reversible) transport, neglecting (irreversible) mixing processes entirely (e.g., Vogel et al., 2019; Hanumanthu et al., 2020)."

**l221: 'lapsrate' should be 'lapse rate'.**

Thanks, it is corrected in the text now.

**l239-240: This seems to repeat some information on back trajectories what has been given early (which potentially causes more confusion – 'is this intended to be subtly different to what has been said previously').**

No, it should not be different. We agree and removed in the revision in Section 2.3.2 the confusing statements. For details, we refer to the track change version of the manuscript.

**l241: 'The spatial uncertainty of calculated back-trajectories increases with time because mixing processes occurring during transport are neglected' – the uncertainty doesn't just result from neglect mixing processes – the nature of chaotic advection is such that e.g. small errors in velocity fields convert into increasingly large errors in particle posi-tion. The idea that one could accurately calculate a trajectory for, e.g. 50 days, simply doesn't make sense (whether or not one accounts for mixing). The key point for back-trajectory calculations on this sort of time scale is that ensembles of particles are used – so one is essentially calculation probabilities of location of origin rather than 'the' location of origin'.**

We agree and changed the respective text in Section 2.3.2 as follows:

"In general, trajectory calculations have limitations caused by trajectory dispersion increasing with the trajectory length, therefore ensembles of trajectories (of about 100 to 200 trajectories) are used here."

**l250: 'Thereby mainly air parcels of ... branch are from below the thermal TP' – do you mean this – i.e. that most of the parcels from below the thermal tropopause are from the lower branch, or do you mean something slightly different, that most of the parcels from the lower branch are from below the thermal tropopause – in which case the 'mainly' should before 'from below'.**

Actually both versions are true. However, we wrote the sentence as intended thus no corrections are made here.

**Figure 15: You use the term 'convective updraft' in this Figure. There seems to be potentially an unfortunate confusion with the use of 'updrafts' and 'downdrafts' as description of rather small-scale features (perhaps 1km) of convective clouds. 'Convective transport' might be a better term (and a better fit to the fact that, as noted above, your model cal-culations are incorporating some kind of global re-analysis scale representation of large-scale transport by convective systems.**

Many thanks for this interesting remark! The labeling in Figure 15 is updated now as suggested.

**References**

[revised manuscript text omitted]

---

## Author Response (AR2)

Reply to the editor's comments on the paper

"In situ observations of CH$_2$Cl$_2$ and CHCl$_3$ show efficient transport pathways for very short-lived species into the lower stratosphere via the Asian and North American summer monsoons"

by V. Lauther et al.

Dear Farahnaz Khosrawi,

we thank you for the helpful remarks and for accepting our manuscript for publication! Your final comments have been considered as follows (Editor's comments are cited in bold face):

**P12, L268: Consider rephrasing sentence. It should rather read: "The most frequent convection up to potential temperature levels of the order of 360 K........"**

We have changed the sentence as suggested.

**P14, Fig 5 caption: Also consider here rephrasing. It should rather read: "......measurements considered as being CH2Cl2- rich air with mixing ratios 5 ppt higher than ......."**

We have changed the text as suggested. Also in Fig 5 caption, we have similarly changed the sentence describing the coloration of CH$_2$Cl$_2$-poor air.

**P14, L288-289: add " as" twice: " are considered as CH2Cl2 rich air:.....are considered as CH2Cl2-poor air.**

We have changed the wording of the text as suggested.

**P22, L449: Also consider here rephrasing sentence. Not clear and sounds weird.**

You are right, this sentence could be more precise. We have changed the wording as follows:

"This analysis directly links 27 WISE measurements (five were observed above the thermal TP) to the convection of hurricane Maria with transport times from the location of max. $\Delta\Theta_{18h}$ to the location of observation ranging between one week and one month."

**P23, L454: something missing here in the sentence? Please check.**

After consideration we only find that "...since convection by hurricane Maria and not since the boundary layer to derive impacts..." could be included to point out again what is NOT done in the following analysis. However, this is only a side note and we intended to keep the text as short as possible. Thus we do not further adapt the sentence because we don't see it to cause any misinterpretation.

**P23, L458: mixing ratios us in September -> mixing ratio found in September**

The sentence is changed as suggested.

**P23, L466: add "of air" or "pathway" after "fast transport"?**

We have added "pathway" after "fast transport" at the respective location in the manuscript.

**P24, Fig10 caption: in two occasions consider changing "since uplift" to "since the air that has been uplifted"**

We don't think that the suggested change of wording conserves the intended meaning of the sentence but it is true, these two sentences could be more precise. We have specified the sentences to "... transport time since the air parcel uplift by hurricane Maria ..."

**P25, L476: ....that tropical air lifted up ->.... that tropical air has been lifted up......**

We have changed the sentence as follows: "This implies that tropical air that has been lifted up by hurricanes ..."

**P26, Fig 11 caption: same here as for P25, L476.**

We similarly have changed the text in Fig 11 caption.

**P26, Fig 11 caption: large volume mixed? Volume of what? Air?**

Good point! We included the following to the sentence: "... a large volume of tropical air mixed into ..."

**P29, L531: mainly due -> are mainly due**

We have changed the text as suggested.

**P30, Fig 14 caption: the location of the full symbol data points ....... -> rather " the full symbol data points mark locations above the tropical TP."**

We agree that the wording of this sentence is not very accurate. We have changed it as follows: "... the location of full symbols mark data points is measurements located above the thermal TP"

**P32, L605: in the region at about -> at potential temperature levels of about 380 K**

We have changed the sentence as suggested.

*In situ observations of CH$_2$Cl$_2$ and CHCl$_3$ show efficient transport pathways for very short-lived species into the lower stratosphere via the Asian and North American summer monsoons*

by V. Lauther et al.

We thank Referee #1 for the thorough reading of and the very helpful remarks on our manuscript. The comments have been considered carefully while revising the draft and resulting modifications are addressed point-py-point in the following (Referee's comments are cited in bold face):

**Specific comments:**

**1. This distinction between tropospheric and stratospheric air is clear in the figures but does get lost in some of the text. For stratospherically relevant numbers reported, these should be calculated with only observations collected above the thermal tropopause. If that is already being done, it is not clearly communicated in the main text. For example, on line 19 in the abstract and in the main text it is not clear if the reported "1-5 weeks into the Ex-LS" was calculated only with observations collected above the thermal tropopause.**

This is a very good remark! We agree and have now specified the given transport times at the respective locations in the paper. In addition, we have added an overview table with median transport times derived from back-trajectories of different lengths (from the model boundary layer and from the location of maximum convection) for all observations in the UTLS and either only for those in the UT or only for those in the LS. The used data set in the new table (Table 2) is also filtered to include only measurements of CH$_2$Cl$_2$-poor air linked to the NAM as well as measurements of CH$_2$Cl$_2$-rich air linked to the ASM. We consistently use median transport times now and provide ranges of transport times derived from the 25. and 75. percentile of the respective data set. For details, we refer to the track change version of the manuscript.

In addition, we noticed that the separation between measurements in the UT and the LS shown in the figures was based on ECMWF's ERA5 reanalysis data (Hersbach et al., 2020). There are but a few differences regarding the discrimination between UT and LS of individual data points when using ERA5 instead of ERA-Interim data. However, to be consistent, we changed the displayed separation between UT and LS in all relevant figures (3, 12 (now 13), 13 (now 14), B1 (now C1); no changes were necessary in Fig. 9 (now 10)) to the calculations based on ERA-Interim.

**2. It is misleading to place N2O and month on the same axis in Figure 4. Measurements of CH2Cl2 at N2O mixing ratios of 325 ppb should not necessarily be compared to surface observations in June; some of the low mixing ratios of CH2Cl2 shown here are likely due to photochemical processing in the stratosphere, not seasonal surface trends.**

This is true and we thought about that while carefully choosing the wording of this section (i.e. "Although this simplified view ignores the impact of mixing processes and chemical reduction of CH$_2$Cl$_2$ it qualitatively explains the lower branch of the correlation curve for air parcels younger than a few months."). However, we think Figure 4 is an interesting qualitative comparison between two different data sets providing the reader a good first impression of a plausible explanation for one part of the rather complex CH$_2$Cl$_2$-N$_2$O relationship. Nevertheless, we agree that Figure 4 should be interpreted with care and not be used out of context. In the caption of Figure 4 we thus refer to the similarities between the CH$_2$Cl$_2$-N$_2$O relationship and the ground-based CH$_2$Cl$_2$ observations from the AGAGE network focusing only on the lower branch of the correlation — implying a comparison of the correlation with AGAGE data from about July to September. This range for a plausible comparison is also stated in the main text ("The observed decrease of low CH$_2$Cl$_2$ mixing ratios for increasing N$_2$O mixing ratios (from older to younger air) agrees well with the decreasing tropical monthly averaged CH$_2$Cl$_2$ mixing ratios from about July to September

2017, as observed by AGAGE."). However, we agree that the figure might be misleading to some readers and therefore we have now included vertical lines highlighting the comparison between July and September to clarify the plausible range of the comparison. In addition, we have specified the respective text in the Figure 4 caption as follows:

"The gradient of the $CH_2Cl_2$-$N_2O$ relationship's lower branch (low $CH_2Cl_2$ mixing ratios between vertical lines) qualitatively fits the temporal variation of the ground-based $CH_2Cl_2$ measurements."

**It should also be noted that biases between NOAA and Advanced Global Atmospheric Gas Experiment (AGAGE) records of CH2Cl2, particularly in the northern tropic station shown in Figure 4, have been identified, suggesting either calibration errors or longitudinal gradients in mixing ratios of CH2Cl2 (Engel et al., 2018).**

**Additionally, the CH2Cl2-poor air is connected to the OH-driven seasonality based on ground-based observations at one AGAGE station without discussing other explanations. However, since CH2Cl2 is not well mixed in the troposphere, a contributing factor could be the uplift of air with less anthropogenic influence than measured at the ground-based station. This option is alluded to on line 402, but it is not clearly introduced as a separate mechanism for low CH2Cl2 mixing ratios**

These are good points and we have included them in the text of Section 3.1 as follows:

"It is thus very likely that the lower branch of the $CH_2Cl_2$-$N_2O$ relationship is caused by the tropical Atlantic $CH_2Cl_2$ surface seasonality. However, the low $CH_2Cl_2$ mixing ratios observed during WISE could also be impacted by air uplifted from regions less influenced by $CH_2Cl_2$ sources than the observations from the AGAGE network at Barbados. Ground-based observations of $CH_2Cl_2$ surface mixing ratios from the AGAGE and the NOAA network show strong regional differences particularly in the NH tropics. However, it is unclear if these differences are caused by calibration biases or are of a natural origin (Engel et al., 2018)."

**3. Variations of "first study" statements appear throughout the paper. I would caution against that phrasing since it seems unnecessary for the importance of the paper and can be untrue if one of the qualifications in these statements are removed. From a literature search, at least Adcock et al. (2021) provides back-trajectory analysis and identifies source regions of measurements of CH2Cl2 in the UTLS. Additionally, Rotermund et al. (2021), which is properly referenced in the discussion section, should also be acknowledged in the introduction as a study that employed similar methods to investigate source regions of brominated VSLS during the WISE campaign.**

We agree and have removed the respective statements of "first study" from the paper. Also Adcock et al. (2021) is discussed in the paper. For details, we refer to the track change version of the manuscript.

Rotermund et al. (2021) is now acknowledged in the introduction of the paper in the following way:

A study by Rotermund et al. (2021) employed similar methods to identify source regions and the impact on the Ex-LS of Br-VSLS using measurements from the same aircraft campaign as the measurements used in the present paper and is compared to our results in Section 4.

**4. It would be helpful to either place the results of the present study in the context of similar studies conducted with observations of other VSLS (e.g., Aschmann et al., 2009; Ashfold et al., 2012; Levine et al., 2007; Liang et al., 2014), or explain in the introduction why the results of those studies have limited applications for studying the transport of Cl-VSLS.**

The results of similar transport studies focused on other VSLS (usually Br-VSLS with mainly natural, oceanic sources) cannot necessarily be used to describe the transport of Cl-VSLS (mainly anthropogenic, land-based sources) because of their different

source distribution in the troposphere. Therefore it is necessary to describe transport pathways into the stratosphere for VSLS (particularly for those with strongly varying mixing ratios in the troposphere) specifically for the respective VSLS or group of VSLS with similar characteristics (e.g., photochemical lifetime and source distribution). To point this out, we have now added the following explanations to the introduction of the paper:

"Observational evidence for Cl-VSLS being transported into the stratosphere is extremely rare  (e.g., Schauffler et al., 1993; Woodbridge et al., 1995; Schauffler et al., 2003; Laube et al., 2008; Park et al., 2010; Adcock et al., 2021). Transport pathways into the stratosphere for VSLS have been derived from observations of bominated VSLS (Br-VSLS; e.g., Sturges et al., 2000; Ashfold et al., 2012; Wales et al., 2018; Filus et al., 2020; Keber et al., 2020; Rotermund et al., 2021) or modeled specifically for Br-VSLS (e.g., Levine et al., 2007; Aschmann et al., 2009; Ashfold et al., 2012; Liang et al., 2014) which have mainly natural emission sources (Engel et al., 2018). However, the only Br-VSLS with a photochemical lifetime comparable to those of $CH_2Cl_2$ and $CHCl_3$ is $CH_2Br_2$ (150 days; WMO, 2018) which is mostly emitted by the oceans and, consequently, is differently distributed in the troposphere than the mainly anthropogenically (land-based) emitted $CH_2Cl_2$ and most Cl-VSLS (e.g., Engel et al., 2018). Thus, transport studies of Br-VSLS focus on transport into the stratosphere from likely different source regions than those of Cl-VSLS and their results might not necessarily be directly applicable to the transport into the stratosphere of $CH_2Cl_2$ and $CHCl_3$. In addition, in order to specifically study transport into the stratosphere via the ASM it is beneficial to observe VSLS with their strongest sources being located in the core region of the ASM. This is the case for $CH_2Cl_2$ while most Asian Br-VSLS sources are located only in adjacent regions of the ASM."

**5. "...two most efficient and fast transport pathways from (sub-)tropical source regions into the extratropical lower stratosphere (Ex-LS)..." (Line 7) This study has identified the ASMA as an efficient pathway into the extratropical stratosphere and the North American monsoon as a fast transport pathway into the Northern Hemisphere upper troposphere/lower stratosphere. However, from the observations presented in this paper, it is not clear how efficiently Cl-VSLS lofted into the upper troposphere by hurricanes are mixed into the stratosphere. This sentence could be simplified to: "...two transport pathways from (sub-)tropical source regions into the extratropical lower stratosphere and upper troposphere...". Further detail of the difference between the two pathways is provided later in the abstract.**

This is true. We changed the sentence as suggested.

**6. Line 132: Please state why earlier flights are not included in this study. Also, altitude and pressure ranges are not given for the measurements. Were any filtering criteria used to limit observations to the UTLS?**

We have reconsidered the wording of Line 132 and added the full ranges of the used data set regarding potential temperature, altitude, and pressure. No explicit filtering of the data set was used to confine the presented observations to the UTLS region. However, due to a malfunctioning water trap of the HAGAR-V instrument, we could perform $CH_2Cl_2$ and $CHCl_3$ measurements only at ambient water vapor levels below about 100 ppm (measuring mainly at around 6 ppm), thus "naturally" confining the observations to the upper part of the troposphere and above. This is briefly explained in Section 2.2.1 to which we now refer in the introduction to the used data set (Sect. 2.1). We also added to Section 2.2.1 why we were able to measure only during the last ten flights of the WISE campaign. The respective parts of the manuscript are modified as follows (sections 2.1 and 2.2.1, respectively):

"In this study we present UTLS measurements between a potential temperature of $315\,\mathrm{K}$ and $404\,\mathrm{K}$ (i.e., $7.4 - 14.5\,\mathrm{km}$ altitude; $388 - 130\,\mathrm{hPa}$ pressure) of the last ten WISE flights, i.e., from 28 September to 21 October 2017 (Figure 1). Due to technical issues of the instrument, $CH_2Cl_2$ and $CHCl_3$ measurements below the given range and during earlier flights of the WISE campaign were not performed (cf. Section 2.2.1)."

"However, during WISE the dehydration system of HAGAR-V was malfunctioning.  For the last 10 WISE flights that system was bypassed and the MS module measured only at low ambient water vapor levels (mainly

220   at $H_2O < 100$ ppm; median: 5.6 ppm), i.e., in the UTLS region.  thus yielding measurements during about 90% of a typical  flight's duration (i.e., about 7.6 h per flight). MS measurements of WISE flights before 28 September could not be used for analysis due to the malfunctioning sample dehydration unit."

**7. Section 3.1.3 would read more clearly if the "Analysis of transport pathways" is discussed before "Case study: con-**
225   **vective uplift by hurricane Maria". As written, what is meant by fast transport time**

We agree and have rearranged the respective paragraphs and figures. Note, that this is not directly marked in the track changes version of the manuscript to keep changes within the text highlighted.

230   **8. Line 370: Please explain why the comparison to August and September ground-based observations is justified based on the transport time since maximum $\Delta\Theta_{18h}$ and not transport time since the boundary layer? Similarly, is the uplift described on line 373 recent or fast uplift of CH2Cl2-poor air (since max $\Delta\Theta_{18h}$ or since the boundary layer, respectively)?**

235   This is a good point and by providing the different calculated transport times (Table 2; cf. response to specific comment "1.") this should be clear now. We additionally included the information about the transport time since the boundary layer to the sentence (see quote below). For $CH_2Cl_2$-poor air parcels with maximum convection above Central America the difference between median transport times from the location of maximum convection and from the model boundary layer is only about one week. Thus 'recent transport' can also be considered 'fast transport'. We agree that this was not clear before we added
240   Table 2 to the paper. In regard to specific comment "8.", the text of Sect. 3.1.3 - "Locations of maximum diabatic ascent rate and transport times" was modified as follows:

"The transport times to the UTLS since the ascent above Central America mainly range between 2 − 5 weeks  (cf. Table 2). The main uplift of $CH_2Cl_2$-poor air above Central America
245   thus falls in the time period of late August and throughout the entire September. With transport times from the BL being only about one week longer (Table 2), this result supports the comparison of $CH_2Cl_2$-poor air with the seasonal minimum $CH_2Cl_2$ mixing ratios observed by AGAGE at Barbados"

**9. In the discussion section, line 504, the authors state "…clearly benefits the use of CH2Cl2 observations to derive**
250   **details about the different transport mechanisms and pathways…". It is an interesting and novel way to frame the paper, and if the authors would like to highlight that aspect, it should be introduced at the end of Section 1. However, in the discussion please note that CH2Cl2 and CH2Br2 have similar atmospheric lifetimes, and regionally varying surface mixing ratios, not just the OH seasonal cycle, can influence the low CH2Cl2 observed in the UTLS. The use of the CHCl3:CH2Cl2 ratio as a diagnostic tracer could also be better highlighted.**

255

These are good points. With our revision of the introduction regarding the specific comment number "4." we also highlighted the beneficial use of a VSL tracer like $CH_2Cl_2$ with its main sources located in the core region of the ASM to analyze transport into the stratosphere via the ASM.

260   We also changed the wording regarding the advantage of $CH_2Cl_2$ over the Br-VSLS analyzed by Rotermund et al. (2021) due to its very strong Asian emission sources and very low mixing ratios in other regions of strong convection. Thereby we highlight the more general fact that we could derive the two transport pathways so clearly due to the strong contrast of tropospheric $CH_2Cl_2$ mixing ratios in the regions of significant convection. The possible reasons for the low $CH_2Cl_2$ mixing ratios are already discussed in Section 3.1 (see our response to the specific comment nr. "2."). However, we wanted to keep the
265   phrasing "[...] a longer lifetime [...]" when comparing $CH_2Cl_2$ to the Br-VSLS analyzed by Rotermund et al. (2021) without specifically commenting on the comparable atmospheric lifetime of $CH_2Br_2$ (this is already mentioned in the introduction, cf. answer to specific comment nr. "4.") because of two reasons: (1) our statement is correct ($CH_2Cl_2$ and $CH_2Br_2$ have similar

atmospheric lifetimes, but on average that of $CH_2Cl_2$ is up to one month longer; WMO, 2018); (2) we write that not a single but the combination of attributes (lifetime and source distribution) is the reason why $CH_2Cl_2$ is the preferred tracer for the analysis of details on the transport pathways into the Ex-UTLS in NH late summer. The respective part of the discussion is now modified as follows:

"Nevertheless, compared to the very short-lived bromine species analyzed by Rotermund et al. (2021), the combination of a longer lifetime, highly significant Asian emission sources, and a strong seasonal cycle and very low mixing ratios in other regions of strong convection clearly benefits the use of $CH_2Cl_2$ observations to derive details about the different transport mechanisms and pathways from the source region into the NH summertime UTLS. In addition, using the $CHCl_3$:$CH_2Cl_2$ ratio to support the analysis of air mass origin is a unique and helpful tool in the analysis of transport pathways."

**10. The discussion of the flight on 1 October (lines 542 – 556) could be moved to the hurricane case study section. Having this text at the end of Section 4 somewhat distracts from the big picture highlights given. Please note on line 553 and in similar text that while NAMA was observed to be a fast transport pathway to the UTLS, a low fraction of stratospheric air originates from central and western ITCZ (Figure 6).**

We agree and moved this paragraph and Figure 14 (now Fig. 11) to the hurricane case study section which itself has also been moved further below in the text (cf. specific comment number "7."). Note, that the moving of figures and paragraphs is not directly marked in the track changes version of the manuscript to keep changes within the text highlighted.

When discussing the transport pathway into the Ex-LS via the NAMA we now added the information about the relatively low fractions of central and western ITCZ air mass origin in stratospheric air as well as the note that we actually observed only 25 % of the air parcels transported by the NAMA within the Ex-LS. We modified the text at three different locations (Section 3.1.3 - Analysis of transport pathways, Section 4 - Discussion (now Section 3.1.3 - Case study: convective uplift by hurricane Maria), and Section 5 - Conclusion) as follows:

"Another aspect adding to the different transport times is the longer transport pathway from Asia because simulations indicated air masses were always observed to reach reached the location of measurement always from the west. Nevertheless, air parcels observed in the Ex-LS are impacted more strongly by air masses transported via the ASMA than via the NAMA (cf. figures 6 and 14)."

"We have thereby shown that tropical surface mixing ratios of VSLS from the region of Central America and the Atlantic Ocean can be efficiently transported into the Ex-LS during the late North American monsoon season. For instance, this is of particular importance for brominated short-lived substances (e.g., $CH_2Br_2$ and $CHBr_3$) that have a high ODP and some of their largest emission sources located in tropical oceans (e.g., Hepach et al., 2015; Rotermund et al., 2021). However, it has to be noted that only 25 % of air parcels transported by this pathway were observed in the Ex-LS and stratospheric air masses showed relatively low fractions of air originating in the region of central and western ITCZ compared to those originating in southern and eastern Asia (cf. Figure 14)."

Eventually, these air masses mixed into the LS by Rossby wave breaking and influenced the chemical composition of the NH Ex-LS $10 - 20$ K below the air masses dominated by transport via the ASMA. However, only 25 % of air parcels transported via this pathway were observed above the thermal TP with transport times from the BL to the location of measurement ranging from 5 to 9 weeks.

**Technical comments:**

**- "Regionally differing ... CHCl3 sources" (Lines 23 – 26): As written this sentence is confusing. Please clarify that a larger fraction of CHCl3 than CH2Cl2 is emitted by natural sources, and, consequently, a lower CHCl3:Cl2Cl2 is found in air parcels that originate from regions with significant anthropogenic influences than regions with weaker**

**anthropogenic emissions.**

We changed the wording of the mentioned sentence in the abstract as follows:

" CH₂Cl₂ and CHCl₃ have similar atmospheric sinks and lifetimes but the fraction of biogenic emissions is clearly higher for CHCl₃ than for the mainly anthropogenically emitted CH₂Cl₂, consequently lower CHCl₃:CH₂Cl₂ ratios are expected in air parcels showing a higher impact of anthropogenic emissions. The observed CHCl₃:CH₂Cl₂ ratio suggests clearly stronger anthropogenic emissions in the region of southern and eastern Asia compared to those in the region of Central America and the tropical Atlantic."

**- "Due to the sparseness of Cl-VSLS measurements in the stratosphere" (Line 36): This statement is true, but please reference some past studies that measured Cl-VSLS in the stratosphere, such as (Laube et al., 2008; Schauffler et al., 1993, 2003) in addition to some of the references already present in the paper.**

References to other studies presenting Cl-VSLS measurements in the stratosphere are now given: "Owing to the sparseness of Cl-VSLS measurements in the stratosphere (e.g., Schauffler et al., 1993, 2003; Laube et al., 2008; Park et al., 2010; Adcock et al., 2021 the impact of changes in Cl-VSLS surface emissions on their distribution in the stratosphere has yet to be fully characterized on an observational basis."

**- Paragraph beginning on Line 54: It's important for your analysis of the CHCl3:CH2Cl2 ratio that the reader understands CH2Cl2 is largely emitted by anthropogenic sources, while a larger fraction of CHCl3 is from natural sources. This should be directly stated for the reader here. Overall, the detail given for the different emissions sources for the two VSLS could be more concise and focused so that this point does not get missed.**

We agree and the phrase " Nevertheless, on a global scale CHCl₃ has a significant fraction of biogenic emission sources in contrast to CH₂Cl₂ which is almost exclusively emitted by anthropogenic sources." was added to the mentioned paragraph to highlight this difference between the two analyzed species.

In addition and in agreement with a comment of Referee #2 we shortened the information given in the introduction to focus more on things directly relevant to the paper.

**- Line 77: It may be helpful to state the atmospheric lifetimes of CH2Cl2 and CHCl3 at the beginning of this paragraph so that they are not lost in the discussion.**

We agree and the mentioned paragraph is now rearranged accordingly:

"For CH₂Cl₂ Hossaini et al. (2019) suggest an average tropospheric lifetime of 168 days (about 6 months) and a stratospheric lifetime of 1 − 2 years (outside the poles) was estimated by Hossaini et al. (2017). The main atmospheric sink of both CH₂Cl₂ and CHCl₃ is the reaction with hydroxyl radicals (OH) in the troposphere. Both species have similar reaction rates with OH implying similar photochemical lifetimes for both Cl-VSLS (Hsu and DeMore, 1994). Time series of background mixing ratios of both species are anticorrelated to the seasonal cycle of OH (Cox et al., 2003). In the NH seasonal anthropogenic use of products releasing CHCl₃ to the atmosphere (e.g., landfill and chlorination of water) have been observed to have a small local impact on the seasonality of CHCl₃ (Gentner et al., 2010). In addition, the global distribution of OH shows significant regional differences (Spivakovsky et al., 2000; Hanisco et al., 2001; Lelieveld et al., 2016). Therefore, also the photochemical lifetimes of CH₂Cl₂ and CHCl₃ are regionally different.

"

**- Line 104: If there are modeling studies that explored source regions of Cl-VSLS in addition to Claxton et al. (2019), they should be referenced here.**

This paragraph highlights the regional dependency of the Cl-VSLS's ODP which is, to the best of our knowledge, presented only by Claxton et al. (2019).

**- Lines 255 - 260: Some of the wording in this paragraph is hard to follow. A useful point that is missing here is N2O, due to its long tropospheric lifetime is well mixed in the troposphere, while CH2Cl2 is not.**

We considered the comment and added the proposed additional information to the text as follows:

"With a photochemical lifetime of 123 years (Ko et al., 2013) $N_2O$ is well mixed in the troposphere and has a much longer life-time than $CH_2Cl_2$ which exhibits strongly varying mixing ratios throughout the boundary layer (e.g., Simmonds et al., 2006)."

**- "measurements of AGAGE" (Line 262): Here and elsewhere in the paper, please correct to read "ground-based measurements of CH2Cl2 from the AGAGE network"**

The text is now changed to the proposed phrasing at all convenient places in the manuscript. For details, we refer to the track change version of the manuscript.

**- Line 299 – 304: The description of the different criteria is hard to follow. Please carefully reword. Also, readers may have to refer back to criteria (1) and (2), it could be helpful to format them as separate bullets.**

We agree and reconsidered the wording and structure of the paragraph modifying it as follows:

"Further, to work out differences of air mass origin between $CH_2Cl_2$-rich and -poor air,  the median fraction of a surface origin tracer in $CH_2Cl_2$-rich air parcels is compared to that in $CH_2Cl_2$-poor air parcels. To combine regions of air mass origin with a particularly high relative impact on either $CH_2Cl_2$-rich or -poor air, the ratio of these median surface origin tracer fractions in $CH_2Cl_2$-rich and -poor air is analyzed. Surface origin tracers with particularly high relative median fractions in either $CH_2Cl_2$-rich or -poor air are combined following these two criteria:

(1) considered are only surface origin tracers with median fractions $\geq 1\,\%$ in $CH_2Cl_2$-rich or -poor air parcels, and

(2) the  ratio of a median surface origin tracer fraction ($CH_2Cl_2$-rich/$CH_2Cl_2$-poor air respectively $CH_2Cl_2$-poor/$CH_2Cl_2$-rich air) must be $> 1.8$.

**- Figure 7: Could the transport time to the boundary layer be shown in a similar figure? The lines given in Figure 10 are hard to distinguish.**

Yes, we added Figure B1 to the appendix and refer to it in Section 3.1.3 as follows:

" The locations of trajectory end points at the model boundary layer color coded with transport time are given in the appendix (Figure B1)."

**- Line 354: Locations of and transport times since maximum diabatic ascent rate**

We have now included also transport times from the boundary layer (Table 2; see reply to specific comment number "1.") and will thus keep the title of the subsection as it is.

**- Line 389: Please state how many of these air masses were collected above the thermal tropopause.**

It were 5 out of 27. The sentence was modified as follows:

"This analysis directly links 27 WISE measurements (five were observed above the thermal TP) to the convection of hurricane Maria with transport times since convection ranging between one week and one month."

**- "positively correlate with transport time" (Line 391): What is shown in Figure 8 is the time since max convection, not the overall transport time. Please clarify the discussion in this paragraph.**

We focus on the transport since the main convection by hurricane Maria because this gives a better impression on the time scale for air parcels lofted by a major hurricane to reach the Ex-UTLS and we analyze impacts on these air parcels induced in the UTLS. However, all trajectories linked to air parcels lifted up by hurricane Maria reach the model boundary layer in September and most do so above the Atlantic Ocean, i.e., at the time of seasonal surface minimum $CH_2Cl_2$ mixing ratios and in a region of weak $CH_2Cl_2$ emission sources. We added the respective information to the text and included a note to the relationship of $CH_2Cl_2$ and transport time since the boundary layer as follows:

"Interestingly, $CH_2Cl_2$ mixing ratios of measurements linked to hurricane Maria positively correlate with transport time since maximum convection ($R_{Pearson} = 0.85$; Figure 10, top right. Note, these $CH_2Cl_2$ mixing ratios also correlate with transport time since the model BL but with a lower $R_{Pearson} = 0.64$ and transport times between 9 and 48 days. However, here we focus on the transport since convection by hurricane Maria to derive impacts on the air parcels induced by processes in the UTLS region). Those air samples related to short transport times contain the lowest $CH_2Cl_2$ mixing ratios at $N_2O > 325$ ppb measured during WISE (Figure 10, top left). According to the back-trajectories, most of the air parcels lifted up by hurricane Maria left the model boundary layer above the tropical Atlantic in September where $CH_2Cl_2$ sources are small (Figure 10, bottom). In addition, in that region the seasonal minimum of $CH_2Cl_2$ mixing ratios is in September (cf. Section 3.1)."

**- Lines 411 and 434: Please clearly state that you are only performing back trajectories for observations collected above the thermal tropopause, otherwise here and elsewhere should read "into the UTLS"**

We analyze the trajectories for observations in the UTLS region and corrected the wording accordingly.

**- Line 447: State for the readers that a broader range of surface mixing ratios of CH2Cl2 (~10 – 70 ppt) are reported from surface stations than for CHCl3 (~5 – 15 ppt) (https://agage.mit.edu/data/agage-data).**

We have considered this statement carefully and concluded that it is not necessarily helpful to the readers. The much clearer split of the $CH_2Cl_2$-$N_2O$ relationship than that of the $CHCl_3$-$N_2O$ relationship is likely not based on the different global absolute surface mixing ratio ranges of the two Cl-VSLS. For air parcels measured during WISE we identified significant regions of air mass origin only around the summertime ITCZ (cf. Figure 2). The surface mixing ratio ranges within these regions between July and September could provide a supporting argument to the discussion. However, using the monthly averaged surface mixing ratios ignores the daily variations at the given location. Including the standard deviation or any other measure of the mean surface mixing ratio scatter range within the relevant regions would actually provide a useful argument for why we can clearly resolve the two branches in the $CH_2Cl_2$-$N_2O$ relationship but not so clearly in the $CHCl_3$-$N_2O$ relationship ($CHCl_3$ has stronger relative variations of mixing ratios within the source regions than $CH_2Cl_2$). However, the sparseness of ground-based Cl-VSLS observations in these regions prevents us from using such an argument. In any case, we think it is not helpful if we provide the global range of monthly averaged mixing ratios of the two Cl-VSLS here.

**- Line 476: Remind the reader here or earlier in the section that a larger fraction of CHCl3 emissions than CH2Cl2 are from natural sources and the two compounds have similar atmospheric lifetimes.**

This is a good suggestion and it was included in the text of Section 3.2 as follows:

"The $CHCl_3$-$CH_2Cl_2$ correlation slope thus flattens with increasing entry of air masses originating from southern and eastern Asia.  Knowing that both species have similar sinks and photochemical lifetimes but $CHCl_3$ has a larger fraction of emissions from biogenic sources than $CH_2Cl_2$, this suggests larger $CHCl_3$:$CH_2Cl_2$ emission ratios in the region of the central and western ITCZ region (with presumably mostly biogenic sources) than in southern and eastern Asia (where anthropogenic sources likely dominate)."

**- Line 408: The ASMA pathway was also identified by Adcock et al. (2021) for CH2Cl2.**

We partly agree on this point. In this paragraph we highlight the transport pathway from Asia via the ASM including eastward outflow of the ASMA into the Ex-LS derived from observations above western Europe/Atlantic. Adcock et al. (2021) present a transport pathway into the LS based on measurements within the ASMA (StratoClim 2017) and observations of westward ASMA outflow (without back-trajectory analysis) above eastern Europe and the Mediterranean (StratoClim 2016). We do not argue the point that the observations by Adcock et al. (2021) are an important contribution to observational studies regarding Cl-VSLS transport into the LS by the ASMA (and we refer to the paper elsewhere in our manuscript) but quoting their paper at line 508 would be inaccurate to the content of this paragraph.

Response to the comments of Referee #2 on the paper

*In situ observations of CH$_2$Cl$_2$ and CHCl$_3$ show efficient transport pathways for very short-lived species into the lower stratosphere via the Asian and North American summer monsoons*

by V. Lauther et al.

We thank Referee #2 for the thorough reading of and the very helpful remarks on our manuscript. The comments have been considered carefully while revising the draft and resulting modifications are addressed point-py-point in the following (Referee's comments are cited in bold face):

**General comments:**

**Introduction p2-3: To me there seemed to be a lot of detail here about the overall significance of the CL-VSLS, and about sources. This is important, of course, but it does not all seem to be essential to this paper. The key points, for this paper, are the general geographical characterisation for the source regions of the different species being considered. The reader would be able to focus more effectively on the major points of this paper if some of the detail was removed.**

We agree. A similar remark was made by Reviewer #1 and we reduced the information given in the introduction to focus on the most essential details about Cl-VSLS. For details, we refer to the track change version of the manuscript.

**p10-11: I think that it is a bit confusing to refer to the 'lower branch of the correlation' for the CH2Cl2-N2O relationship. It looks as though there are large number of parcels for which the N2O values are around 330ppb but the CH2Cl2 values are distributed across the range 30-45ppt. I think that in some ways it weakens your case if you call something a correlation when by conventional measures the correlation is rather weak. I am not disputing the fact that there are two 'families' in the plot at high N2O values – but can you find a more neutral term to describe them?**

We agree that our use of the term "correlation" is rather based on lab jargon than on mathematics. We now use the general and more precise wording of a "CH$_2$Cl$_2$-N$_2$O relationship" instead of "CH$_2$Cl$_2$-N$_2$O correlation" if "correlation" is not specifically meant. We thus refer now to the "lower branch of the CH$_2$Cl$_2$-N$_2$O relationship" to describe the values of particularly low CH$_2$Cl$_2$ mixing ratios at high N$_2$O mixing ratios (similar for the upper branch). These modifications are made throughout the manuscript and for details we refer to the track change version of it.

**p12: A similar comment applies to this approach of fitting a 'mean correlation curve' and using that as a basis for separating the upper and lower branches. The quadratic fit might be intended to seem quantitative – but there is really no reason to believe that the extrapolated values have any concrete relevance to the concentrations of CH2Cl2 measured in high N2O concentration parcels – the split between the two categories is essentially being made on the basis of the appearance of the plot. It seems very reasonable to make the split – but if the authors (or the readers) felt that it was more justifiable on the basis of a quadratic fit and an extrapolation then I would say that they were confused.**

It is true, the split between the two categories of data was made on the basis of the appearance of the CH$_2$Cl$_2$-N$_2$O relationship and we do not claim otherwise. In order to decide on the "geometry" of the split we used the 'mean correlation curve' as a best-guess proxy of the annual mean CH$_2$Cl$_2$-N$_2$O relationship. Other methods can be thought of to realize a similar split of the data set. We changed the extrapolation of the shown 'mean correlation curve' in Figure 5 to a dashed line to reduce a possible misunderstanding about the origin of this curve.

**p21: At this stage in the paper you use the term 'convective transport' quite frequently and in association with trajectories/CLaMS. I think that you should be a bit clearer about what transport is included in trajectories/CLaMS – which I believe is simply that in the ERA-I velocity fields, i.e. there is no inclusion of convective transport by parametrisation. I suspect that ERA-I velocities tend to be rapidly upward in regions of large-scale convection – and that serves as some kind of representation of convective transport, but it is unlikely that upward velocities are quantitatively correct. Certainly this sort of interpretation has been made by many authors who have used trajectory-based approaches, including myself, and I would not quarrel with it – but I do think that it needs to be clearly stated. The uncertainty perhaps becomes a bit more serious when considering tropical cyclones. How well is vertical transport in tropical cyclones represented by something like ERA-I? One imagines that maximum vertical velocities are significantly underestimated – but it could be, for example, that vertical transport is distributed over too large a region. You may not be able to resolve this uncertainty, but I think that you should at least say that it exists.**

Many thanks for this helpful comment and we confirm that in CLaMS as used here there is no inclusion of convective transport by parametrisation. We added the following text to Sect. 2.3.

" In CLaMS, the diabatic approach was applied using the diabatic heating rate as the vertical velocity with contributions from radiative heating including the effects of clouds, latent heat release, mixing, and diffusion (for details, see Ploeger et al., 2010). CLaMS employs a hybrid vertical coordinate ($\zeta$) which, in this study, transforms from a strictly isentropic coordinate ($\Theta$) to a pressure-based orography-following coordinate system ($\sigma$ coordinates) below a threshold of approximately 300 hPa(Pommrich et al., 2014). In both three-dimensional simulations as well as in trajectory calculations, the upward transport in CLaMS is driven by ERA-Interim reanalysis data in which changes are implemented to improve deep and mid-level convection compared to previous reanalysis data (Dee et al., 2011). However small-scale rapid uplift in convective cores is not included, therefore small-scale convection is most likely underestimated in CLaMS simulations driven by ERA-Interim. Nevertheless, upward transport in larger convective systems such as tropical cyclones is represented in CLaMS trajectory calculations driven by ERA-Interim (Li et al., 2017, 2020)."

CLaMS trajectory calculations using ECMWF's next-generation reanalysis ERA5 (Hersbach et al., 2020), demonstrate that diabatic trajectory calculations using ERA5 show much faster and stronger vertical transport than ERA-Interim primarily because of ERA5's higher spatial and temporal resolution, which likely resolves convective events more accurately (Li et al., 2020). Nonetheless, Li et al. (2020) demonstrate that the large scale convective uplift as represented in ERA-Interim does represent well the large scale uplift in convective systems.

**Detailed comments:**

**l126: 'last accessed' info not needed in citation?**

The last accessed info is now removed in the main text.

**l164: 'essential' would be better than 'mandatory'. ('Mandatory' means 'required by some rule or regulation'.)**

Good point, we changed "mandatory" to "essential".

**l210-215: I found these sentences confusing. What is the distinction between 'pure CLaMS back trajectories', 'pure CLaMS trajectories' and 'back trajectories'? There seem to be two separate issues here – 'forward vs backward' and 'with CLaMS mixing vs without CLaMS mixing'.**

We agree that this section was written a bit confusing and changed the text as follows:

"To support the interpretation of airborne measurements we use global three-dimensional simulations of the Chemical La-grangian Model of the Stratosphere (CLaMS; McKenna, 2002a,b; Pommrich et al., 2014) as well as  CLaMS back-trajectory calculations.  CLaMS back-trajectory calculations consider only the advective (reversible) transport, neglecting
575  (irreversible) mixing processes entirely (e.g., Vogel et al., 2019; Hanumanthu et al., 2020)."

**l221: 'lapsrate' should be 'lapse rate'.**

Thanks, it is corrected in the text now.

580

**l239-240: This seems to repeat some information on back trajectories what has been given early (which potentially causes more confusion – 'is this intended to be subtly different to what has been said previously').**

No, it should not be different. We agree and removed in the revision in Section 2.3.2 the confusing statements. For details, we
585  refer to the track change version of the manuscript.

**l241: 'The spatial uncertainty of calculated back-trajectories increases with time because mixing processes occurring during transport are neglected' – the uncertainty doesn't just result from neglect mixing processes – the nature of chaotic advection is such that e.g. small errors in velocity fields convert into increasingly large errors in particle posi-
590  tion. The idea that one could accurately calculate a trajectory for, e.g. 50 days, simply doesn't make sense (whether or not one accounts for mixing). The key point for back-trajectory calculations on this sort of time scale is that ensembles of particles are used – so one is essentially calculation probabilities of location of origin rather than 'the' location of origin'.**

We agree and changed the respective text in Section 2.3.2 as follows:

595

"In general, trajectory calculations have limitations caused by trajectory dispersion increasing with the trajectory length, therefore ensembles of trajectories (of about 100 to 200 trajectories) are used
600  here."

**l250: 'Thereby mainly air parcels of ... branch are from below the thermal TP' – do you mean this – i.e. that most of the parcels from below the thermal tropopause are from the lower branch, or do you mean something slightly different, that most of the parcels from the lower branch are from below the thermal tropopause – in which case the 'mainly'
605  should before 'from below'.**

Actually both versions are true. However, we wrote the sentence as intended thus no corrections are made here.

**Figure 15: You use the term 'convective updraft' in this Figure. There seems to be potentially an unfortunate confusion
610  with the use of 'updrafts' and 'downdrafts' as description of rather small-scale features (perhaps 1km) of convective clouds. 'Convective transport' might be a better term (and a better fit to the fact that, as noted above, your model cal-culations are incorporating some kind of global re-analysis scale representation of large-scale transport by convective systems.**

615  Many thanks for this interesting remark! The labeling in Figure 15 is updated now as suggested.

[revised manuscript text omitted]

**2.3.2 Back-trajectory calculations**

895 In order to investigate the transport pathways corresponding to the WISE measurements analyzed here, the trajectory module of CLaMS was used to calculate back-trajectories.  The back-trajectories are initialized at

the time and location of the center of the respective MS sample integration time window and end at the first contact with the model boundary layer (below $2-3\,$km above surface). In general, the maximum length of a trajectory is confined to 120 days, however most of the trajectories reach the model boundary layer much earlier.

In general, trajectory calculations have limitations caused by trajectory dispersion increasing with the trajectory length, therefore ensembles of trajectories (of about 100 to 200 trajectories) are used here. The maximum trajectory length of 120 days was chosen to match a large part of the time frame of the three-dimensional CLaMS simulation but the average length of the used back-trajectories is 50 days. We will show (in sections 3.1.2 and 3.1.3) that the results of the three-dimensional CLaMS simulation in which mixing of air parcels is included agrees very well with the results of the back-trajectory analysis.

**3 Results**

**3.1 $CH_2Cl_2$-$N_2O$ relationship during WISE**

The analysis presented in this paper is mainly based on the $CH_2Cl_2$-$N_2O$ relationship observed during WISE (Figure 3). With a photochemical lifetime of 123 years (Ko et al., 2013) $N_2O$ is well mixed in the troposphere and has a much longer lifetime than $CH_2Cl_2$ which exhibits strongly varying mixing ratios throughout the boundary layer (e.g., Simmonds et al., 2006). As expected, the $CH_2Cl_2$-$N_2O$ relationship is relatively compact for data points with low $N_2O$ mixing ratios (i.e., $N_2O < 325\,$ppb, relatively old, mixed and processed air). Towards younger air masses ($N_2O > 325\,$ppb) there is a distinct split of the compact relationship into two branches. In the stratosphere, the upper branch of the $CH_2Cl_2$-$N_2O$ relationship shows up to $150\,\%$ enhanced $CH_2Cl_2$ mixing ratios compared to data of the lower branch at the same $N_2O$ mixing ratios. For $N_2O > 328.5\,$ppb, data points with low $CH_2Cl_2$ mixing ratios even decrease with increasing $N_2O$ (Figure 3, inlay). In general, the majority of measurements was obtained in the stratosphere above the thermal tropopause (TP) with an increasing number of observations below the thermal TP for increasing $N_2O$ mixing ratios. Thereby mainly air parcels of the lower branch of the $CH_2Cl_2$-$N_2O$ relationship  are from below the thermal TP.

[Figure]

**Figure 3.** CH$_2$Cl$_2$-N$_2$O relationship color coded by flight date. The embedded figure shows a detailed magnification of decreasing CH$_2$Cl$_2$ mixing ratios with increasing N$_2$O within the lower branch of the CH$_2$Cl$_2$-N$_2$O relationship. Air parcels below the thermal tropopause are marked as open circles, air parcels above by closed circles.

The most frequent convection up to potential temperature levels of the order of 360 K is expected to originate in the tropics. Therefore, tropical monthly averaged ground-based  measurements of CH$_2$Cl$_2$ from the AGAGE network at Ragged Point, Barbados (Figure 4; Prinn et al., 2018) were analyzed. These AGAGE observations suggest an explanation for the lower branch of the CH$_2$Cl$_2$-N$_2$O relationship observed during WISE. The mainly OH induced CH$_2$Cl$_2$ seasonality results in minimum tropical CH$_2$Cl$_2$ surface mixing ratios in September 2017. This September minimum is comparable to WISE data of low CH$_2$Cl$_2$ mixing ratios in the UTLS region in October (Figure 4, at N$_2$O $\approx$ 330.9 ppb) assuming a transport time from Earth's surface to the UTLS region of a few weeks. The observed decrease of low CH$_2$Cl$_2$ mixing ratios for increasing N$_2$O mixing ratios (from older to younger air) agrees well with the decreasing tropical monthly averaged CH$_2$Cl$_2$ mixing ratios from about July to September 2017, as observed by AGAGE. Extratropical NH ground-based  observations from the AGAGE network yield significantly higher CH$_2$Cl$_2$ mixing ratios than those in the tropics. It is thus very likely that the lower branch of the CH$_2$Cl$_2$-N$_2$O relationship is caused by the tropical Atlantic CH$_2$Cl$_2$ surface seasonality. However, the low CH$_2$Cl$_2$ mixing ratios observed during WISE could also be impacted by air uplifted from regions less influenced by

935 CH$_2$Cl$_2$ sources than the observations from the AGAGE network at Barbados. Ground-based observations of CH$_2$Cl$_2$ surface mixing ratios from the AGAGE and the NOAA network show strong regional differences particularly in the NH tropics. However, it is unclear if these differences are caused by calibration biases or are of a natural origin (Engel et al., 2018).

[Figure]

**Figure 4.** Monthly averaged ground-based  measurements of CH$_2$Cl$_2$ from the AGAGE network at Ragged Point, Barbados (13° N Prinn et al., 2018) overlayed by a detailed plot of the CH$_2$Cl$_2$-N$_2$O relationship observed during WISE. The AGAGE CH$_2$Cl$_2$ time series (shown in red) shows CH$_2$Cl$_2$'
[revised manuscript text omitted]